# FLARE: FULLY INTEGRATION OF VISION-LANGUAGE REPRESENTATIONS FOR DEEP CROSS-MODAL UNDERSTANDING

**Zheng Liu**[1,2], **Mengjie Liu**[1,2], **Jingzhou Chen**[1,2], **Jingwei Xu**[5], **Bin Cui**[1], **Conghui He**[2], **Wentao Zhang**[1,3,4,*]

[1]Peking University, [2]Shanghai AI Laboratory, [3]Zhongguancun Academy, [4]Beijing Key Laboratory of Data Intelligence and Security (Peking University), [5]Nanjing University

## ABSTRACT

We introduce FLARE, a family of vision language models (VLMs) with a fully vision-language alignment and integration paradigm. Unlike existing approaches that rely on single MLP projectors for modality alignment and defer cross-modal interaction to LLM decoding, FLARE achieves deep, dynamic integration throughout the pipeline. Our key contributions include: (1) **Text-Guided Vision Encoding** that incorporates textual information during vision encoding to achieve pixel-level alignment; (2) **Context-Aware Alignment Decoding** that aggregates visual features conditioned on textual context during decoding for query-level integration; (3) **Dual-Semantic Mapping Loss** to supervise feature mapping from both modalities and enable modality-level bridging; and (4) **Text-Driven VQA Synthesis** that leverages high-quality text to generate VQA pairs and synthesize corresponding images, enabling data-level optimization. We train FLARE at 3B and 8B scales under both fixed and dynamic resolution settings, demonstrating that our full-modality alignment significantly outperforms existing methods while maintaining strong generalizability. FLARE 3B surpasses Cambrian-1 8B and Florence-VL 8B using only 630 vision tokens. Ablation studies reveal that FLARE achieves superior performance over existing methods with minimal computational cost. Even without dynamic resolution, FLARE outperforms LLaVA-NeXT, validating the effectiveness of our approach.

## 1 INTRODUCTION

Research has shown that human visual perception is not a passive process like a camera capturing reality but rather an active, interpretative mechanism shaped by factors such as language and environment. In (Gary, 2012), the authors demonstrated that hearing the name of a target object before searching for it significantly improved the speed and accuracy of object detection. This underscores the deep integration of vision and language, suggesting that linguistic input facilitating more effective visual processing by directing attention and helping the brain prioritize relevant features.

Despite the fundamental role of vision-language interaction in human perception, current VLMs fail to capture this relationship. Traditional models (Liu et al., 2023; Chen et al., 2023) process vision features independently via encoders and fuse them with text during decoding. This approach results in a lack of centralized encoding of vision features, leading to limited useful information available for interaction. Recent advancements (Liu et al., 2024a; Bai et al., 2025; Li et al., 2024) enrich visual representations through dynamic resolution or multiple encoders. However, these improvements focus on enhancing visual encoding itself, overlooking the deeper, bidirectional integration necessary for effective vision-language fusion. Our alignment visualization in Figure 1 further confirms this issue: LLaVA (Liu et al., 2023) and LLaVA-NeXT (Liu et al., 2024a), lacking semantic alignment across modalities, fail to achieve proper feature mapping after the projector. At the decoding stage, both show weak attention, with LLaVA-NeXT's excessive vision tokens further dispersing focus.

---

* Corresponding Author

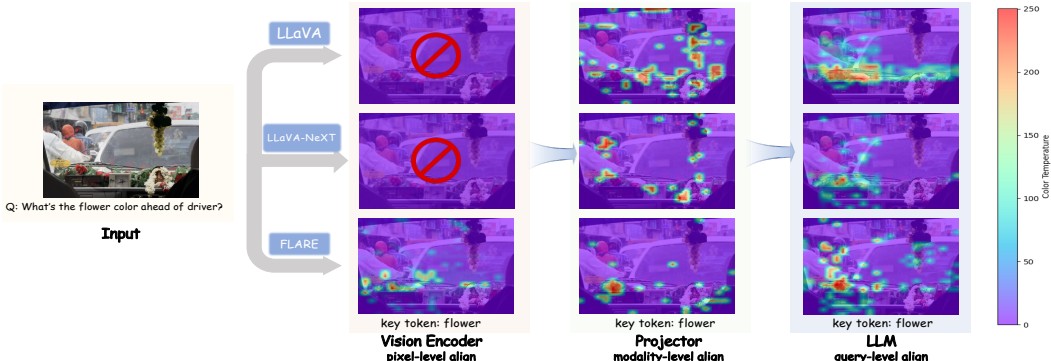

Figure 1: We visualize modality alignment by computing attention maps between image and query at pixel/query levels, and cosine similarity in the LLM space at modality level. The results indicate that our model achieves consistent and progressively enhanced cross-modal alignment and integration.

This challenge serves as the core motivation for our work: **achieving deep integration between vision and language throughout the pipeline**. Previous studies have explored varied integration strategies, such as text-based QFormer integration (Dai et al., 2023), and entity-enhanced modality alignment (Yin et al., 2024). Nevertheless, these models lack a comprehensive algorithm for aligning and integrating modalities. A key unresolved issue is embedding misalignment, where inherent discrepancies between vision and text embeddings hinder seamless integration. Moreover, current models lack high-quality training data designed to guide the alignment and integration process.

In this paper, we push the boundaries of vision-language fusion by proposing a sophisticated algorithm that realizes a deeper, more interactive integration of visual and textual. Our approach consistently integrates textual guidance throughout the visual encoding. We also introduce self-supervised guidance in modality space mapping and dynamically aggregate vision features based on textual context during decoding. Additionally, we introduce a new data synthesis method that generates QA pairs from text, guiding the model toward more efficient feature integration. Our method enables a more cognitively inspired multimodal learning paradigm. Overall, our contributions are as follows:

- **Text-Guided Vision Encoding**: We project text embeddings into visual space and jointly perform attention with visual representations to enhance pixel-level integration.
- **Context-Aware Alignment Decoding**: We introduce context-aware latent tokens and semantic exchange layer that aggregate visual features based on textual context during LLM decoding, enabling fine-grained, query-level alignment and integration.
- **Dual-Semantic Mapping Loss**: We propose bidirectional reconstruction loss, mitigating semantic discrepancies between modalities, achieving modality-level alignment.
- **Text-Driven VQA Synthesis**: We develop a new method that prioritizes constructing high-quality QA pairs while leveraging generative models to produce images. We synthesize a large-scale, diverse QA dataset, enabling data-level optimization.

Building on our exploration, we present a new family of VLMs that achieve leading performance across diverse benchmarks. To foster further research, we will release our code, data, and models.

## 2 PRELIMINARIES AND RELATED WORK

**Vision Language Models.** Vision Language Models (VLMs) (Liu et al., 2023; 2024a) are designed to process information from multiple modalities, primarily combining visual and textual data. Typically, images are divided into patches and processed by vision encoders to obtain visual representations, which are then projected into the LLM space and fused with text tokens during decoding. Recent advances (Li et al., 2024; Bai et al., 2025; Tong et al., 2024a; Shi et al., 2025) have explored dynamic resolution and multiple vision encoders, enhancing the precision of visual encoding. However, these methods primarily emphasize visual encoding quality, neglecting deeper and more interactive vision-language dynamics, thus hindering the full potential of multimodal learning.

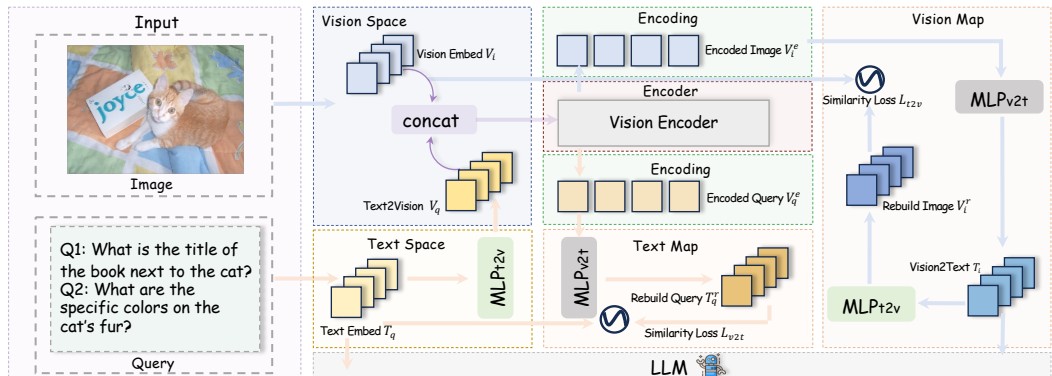

Figure 2: Illustration of Text-Guided Vision Encoding and Dual-Semantic Mapping Loss. The query is projected into vision space and processed jointly with image. The extracted features are then mapped into text space for decoding. To enhance mapping reliability, we reconstruct text and image tokens and enforce similarity with raw tokens, promoting structural alignment across modalities.

**Modality Alignment and Integration.** Recent studies have explored different alignment strategies. (Dai et al., 2023) incorporates text guidance by embedding textual tokens into QFormer, enabling language-driven visual feature projection. (Yin et al., 2024) introduces a contrastive loss in the projector during pretraining, strengthening the association between visual features and entity representations. More recently, (Chen et al., 2024b) proposed a generative vision encoder that leverages multiple textual prompts to integrate diverse feature types, achieving state-of-the-art performance. Nonetheless, a key challenge persists: the inherent disparity between visual and textual feature spaces makes robust alignment difficult, often resulting in inconsistent multimodal representations.

**Instruction Tuning Data.** Existing works (Li et al., 2024; Tong et al., 2024a; Shi et al., 2025) rely on visual question answering (VQA) benchmarks to construct training recipes. Other efforts (Wu et al., 2023; Chen et al., 2024a) leverage GPT-4V to generate high-quality QA pairs from images. While these datasets improve multimodal understanding, they remain vision-centric, as QA pairs are grounded mainly in image content. This reliance limits their capacity to support complex, instruction-following tasks across diverse domains. A promising alternative (Liu et al., 2025) seeks to overcome this limitation by first synthesizing high-quality captions and then generating corresponding images via diffusion models. However, this approach is still confined to pretrain, and the dataset scale remains small, restricting its effectiveness for large-scale multimodal training.

## 3 METHOD

### 3.1 TEXT-GUIDED VISION ENCODING

As a first step, we fundamentally revise the vision encoding paradigm by incorporating textual information directly into the vision encoder to enhance multimodal integration. We design a specialized fusion strategy that balances semantic refinement with visual detail preservation.

As shown in Figure 2, given an image $I$ and query $Q$, we extract visual embeddings $V_i$ and textual embeddings $T_q$ from encoder and LLM, and map $T_q$ into vision space to obtain Text2Vision $V_q$.

$$V_i = \text{VisionEmbed}(I),$$
$$T_q = \text{LLMEmbed}(Q), \quad V_q = \text{MLP}_{t2v}(T_q). \tag{1}$$

The mapped Text2Vision $V_q$ and visual embeddings $V_i$ are then jointly processed in encoder, enabling mutual refinement of multimodal features. Each layer updates these features as follows:

$$(V_i^k, V_q^k) = \text{EncoderLayer}(V_i^{k-1}, V_q^{k-1}), \quad k = 1, \dots, N. \tag{2}$$

where $N$ represents the total number of encoder layers, with $V_i^0 = V_i$ and $V_q^0 = V_q$.

Since lower-layer visual features lack semantic information, we mask visual-to-textual attention in the first half of the encoder layers, allowing early layers to focus on learning visual representations.

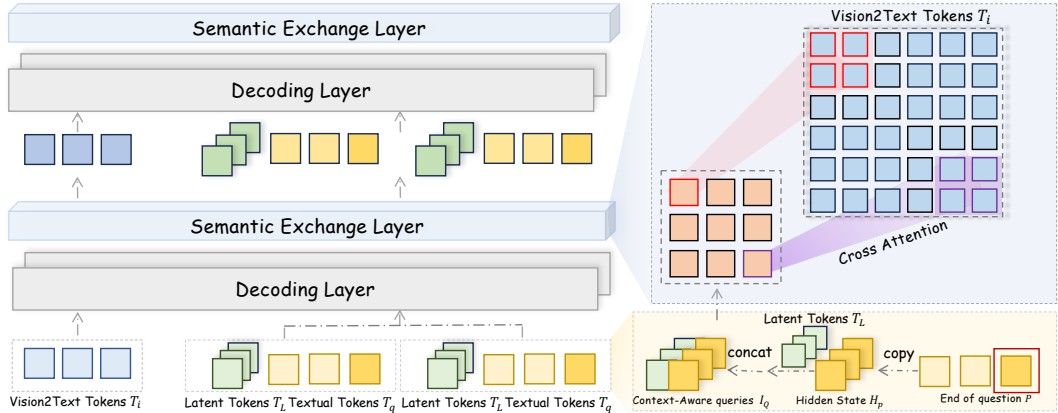

Figure 3: Illustration of Context-Aware Alignment Decoding. For query textual tokens $T_q$( highlighted in yellow), we prepend a set of context-aware latent tokens $T_L$(highlighted in green). Additional semantic exchange layers are introduced between decoding layers, where Vision2Text image tokens $T_i$ interact with both latent tokens and textual tokens at a query-level granularity.

After encoding, we average features in shallow(first-half) and deep(second-half) layers to capture both coarse, predominantly vision-only information and fine-grained, richly text-enhanced details:

$$(V_i^s, V_q^s) = \frac{2}{N} \sum_{k=1}^{\frac{N}{2}} (V_i^k, V_q^k), \quad (V_i^d, V_q^d) = \frac{2}{N} \sum_{k=\frac{N}{2}+1}^{N} (V_i^k, V_q^k). \tag{3}$$

Instead of using only final-layer features as in prior work, we concatenate both vision-only and text-enhanced features, providing a balance between unimodal fidelity and cross-modal enrichment:

$$V_i^e = \text{Concat}(V_i^s, V_i^d; \text{dim=channel}), \quad V_q^e = \text{Concat}(V_q^s, V_q^d; \text{dim=channel}). \tag{4}$$

Finally, visual features $V_i^e$ are mapped into LLM space, yielding Vision2Text image tokens $T_i$. Textual embeddings $T_q$ serve as textual tokens, interacting with $T_i$ through causal mask in decoder.

$$T_i = \text{MLP}_{v2t}(V_i^e). \tag{5}$$

### 3.2 CONTEXT-AWARE ALIGNMENT DECODING

Conventional approaches defer cross-modality interaction to LLM decoding, where the causal mask restricts such interaction to a unidirectional flow. To overcome this, we propose a novel token organization strategy, as shown in Figure 3. For each query, we prepend context-aware latent tokens $T_L$, which serve as intermediaries for cross-modal integration. We further introduce semantic exchange layers between LLM decoding layers, where latent tokens $T_L$ refine visual features under textual context, enabling bidirectional and fine-grained cross-modal interaction during decoding.

We employ windowed attention in semantic exchange layers to achieve efficient modality interaction. Let $T_L \in \mathbb{R}^{l \times l \times D}$ denote the latent tokens, where $l$ is the spatial resolution and $D$ the embedding dimension. The Vision2Text image tokens are denoted as $T_i \in \mathbb{R}^{m \times n \times D}$, serving as keys and values. Window sizes $w = m/l, h = n/l$ restrict attention computation within local regions.

During decoding, given a textual query $Q$, we first identify the position $P$ corresponding to the end of the query textual tokens. The hidden state $H_P \in \mathbb{R}^{1 \times D}$, which aggregates all preceding context, is extracted and concatenated with each latent token $T_L[r, c]$ to form context-aware queries:

$$I_Q[r, c] = \text{MLP}\big(\text{Concat}(H_P, T_L[r, c])\big) \in \mathbb{R}^{1 \times D}. \tag{6}$$

where $r, c \in \{0, 1, \ldots, l-1\}$ are the row and column indices, with $l$ being its spatial resolution.

Next, we use Vision2Text image tokens $T_i$ as keys and values. Latent token $T_L[r, c]$ is updated as:

$$T_L[r, c] = \text{softmax}\left(\frac{Q[r, c]K[r, c]^T}{\sqrt{D}}\right) V[r, c] \in \mathbb{R}^{1 \times D}, \tag{7}$$

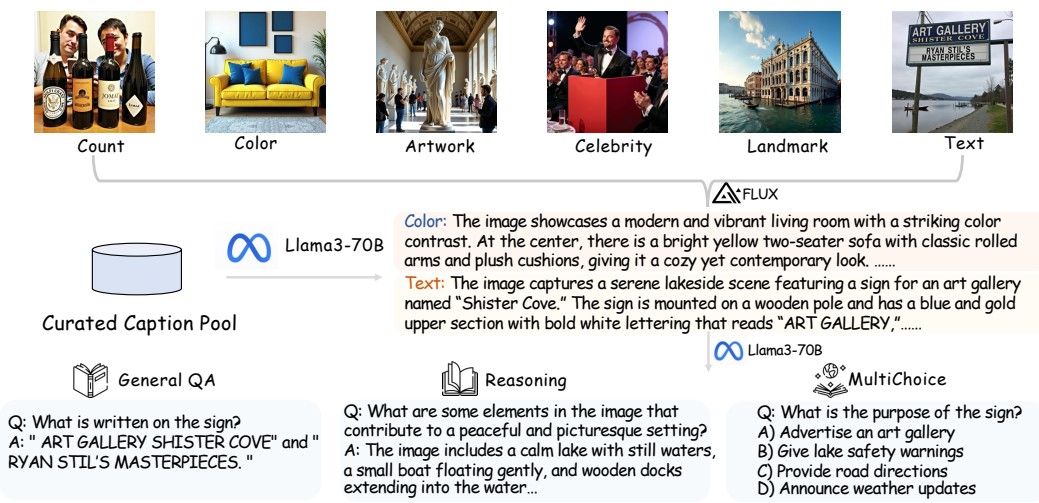

Figure 4: Overview of Text-Driven VQA Synthesis framework. We leverages high-quality captions as the foundation for both image generation via diffusion models and diverse QA pair construction.

where

$$Q[r, c] = W^Q \cdot I_Q[r, c], \quad Q[r, c] \in \mathbb{R}^{1 \times D},$$
$$K[r, c] = W^K \cdot T_i[r \cdot w : (r + 1) \cdot w, \ c \cdot h : (c + 1) \cdot h], \quad K[r, c] \in \mathbb{R}^{(w*h) \times D},$$
$$V[r, c] = W^V \cdot T_i[r \cdot w : (r + 1) \cdot w, \ c \cdot h : (c + 1) \cdot h], \quad V[r, c] \in \mathbb{R}^{(w*h) \times D}.$$

Here, $W^Q, W^K, W^V$ are learnable projection matrices.

In essence, each semantic exchange layer extracts the most relevant visual features for the current query and embeds them into latent tokens. In the subsequent decoding layer, these enriched tokens are integrated with query tokens, enabling bidirectional modality interaction.

### 3.3 DUAL-SEMANTIC MAPPING LOSS

In the previous sections, visual and textual features are frequently mapped and interact. To better guide feature mapping, we introduce Dual-Semantic Mapping Loss based on two complementary projectors: $\text{MLP}_{v2t}$ and $\text{MLP}_{t2v}$. The illustration of our reconstructed loss is shown in Figure 2.

For $\text{MLP}_{v2t}$, it is crucial to ensure that the mapped visual features align closely with the feature space of LLM. We shift our focus to $V_q^e$, which represents textual tokens processed by the vision encoder in the visual space. Consequently, after mapping $V_q^e$ through $MLP_{v2t}$, its rebuilt textual representation $T_q^r$ should closely resemble the LLM-based textual representation $T_q$. To quantify the alignment of $\text{MLP}_{v2t}$, we compute the cosine similarity-based loss between these two features:

$$T_q^r = \text{MLP}_{v2t}(V_q^e), \quad \mathcal{L}_{v2t} = 1 - \frac{T_q \cdot T_q^r}{|T_q| \cdot |T_q^r|}. \tag{8}$$

Analogous to $\mathcal{L}_{v2t}$, we use $T_i$ to evaluate the effectiveness of $MLP_{t2v}$. An optimal $MLP_{t2v}$ should transform $T_i$ into a rebuilt image representation $V_i^r$ that closely resembles vision feature $V_i$ in the visual space. We again employ cosine similarity loss to assess the quality of $MLP_{t2v}$:

$$V_i^r = \text{MLP}_{t2v}(T_i), \quad \mathcal{L}_{t2v} = 1 - \frac{V_i \cdot V_i^r}{|V_i| \cdot |V_i^r|}. \tag{9}$$

The training objective combines cosine similarity losses with the traditional cross-entropy loss, balanced by a weighting parameter:

$$\mathcal{L}_{total} = \mathcal{L}_{cross-entropy} + \lambda(\mathcal{L}_{v2t} + \mathcal{L}_{t2v}). \tag{10}$$

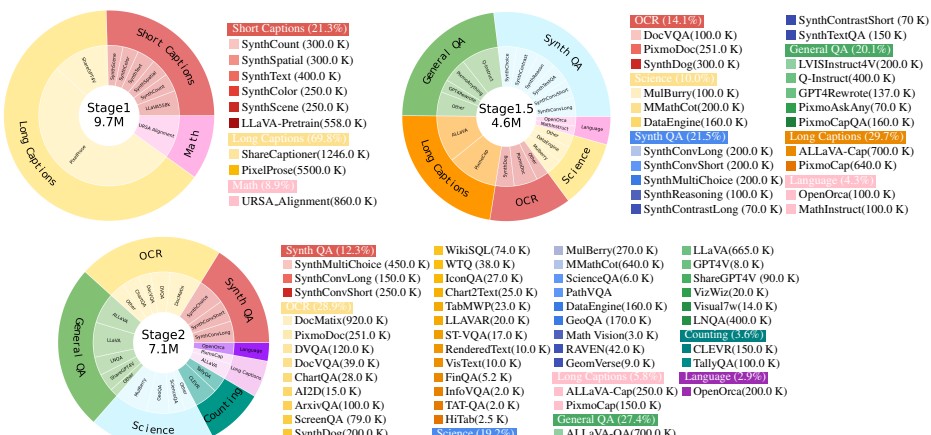

Figure 5: FLARE-10M and FLARE-12M. We collected a total of 10M samples for pretraining (Stage 1) and 12M samples for fine-tuning (Stage 1.5 and Stage 2). The circle illustrates the data distribution, with the right side of the circle showing all the utilized data and their proportions.

The intuition behind $\mathcal{L}_{v2t}$ is to first derive visual features corresponding to textual tokens via Text-Guided Vision Encoding, and then project them back into the textual space for alignment with the original textual embeddings. $\mathcal{L}_{t2v}$ is defined in a symmetrical manner. Importantly, these loss functions require no additional input and can be seamlessly integrated into the overall model architecture.

### 3.4 TEXT-DRIVEN VQA SYNTHESIS

The cross-modality integration components we proposed require diverse combinations of text and images for effective training. However, existing datasets construct QA pairs based on images, where the constraints of fixed visual content lead to monotonous and limited textual diversity. A more direct approach is to first ensure textual diversity, then generate corresponding visual content to match the rich variety of QA pairs. Based on this principle, we propose text-driven VQA synthesis, as shown in Figure 4, which shifts the emphasis from visual content to textual richness and diversity.

Our process begins with carefully selecting high-quality captions. These captions are then enriched with LLM, producing detailed and nuanced textual descriptions that capture various visual and contextual attributes. These enriched descriptions serve as prompts for diffusion model, which generates images closely aligned with the textual context. Simultaneously, we utilize the descriptions as input content, leveraging the LLM again to construct diverse QA pairs, ensuring broad coverage across multiple-choice, multi-turn dialogues and reasoning tasks. This step distinguishes our method from traditional image-first or MLLM-based approaches. Directly leveraging comprehensive textual data significantly enhances the model's understanding and representation of visual contexts.

In Appendix E, we provide a detailed breakdown of our data synthesis process. Our approach combines prompt engineering with a rigorous, multi-stage filtering pipeline to design diverse QA tasks, ensuring high standards of quality and alignment across modalities. Through this way, our model is equipped with various QA data, enabling comprehensive training of cross-modality integration.

## 4 EXPERIMENTS

**Training Strategies.** We propose a three-stage training framework for vision–text alignment and integration: 1) Stage1-Foundational Semantic Alignment: pretrain the vision encoder to jointly handle visual and textual representations; 2) Stage1.5-Contextual Multimodal Fusion: train on diverse QA and caption data to strengthen cross-modal alignment in varied contexts; 3) Stage2-Visual Instruction Tuning: finetune on visual task datasets for effective downstream question answering.

**Data Synthesis Details.** For Stage1, we generated five datasets—SynthColor, SynthCount, SynthSpatial, SynthText, and SynthScene—yielding 1.5M image–caption pairs. In Stage 1.5, seven categories were created—SynthMultiChoice, SynthConvLong, SynthConvShort, SynthReasoning,

Table 1: Model comparison on 16 Benchmarks. # Vis tok. denotes the roughly averaged number of visual tokens across all benchmarks.

| Model / Method | # Vis tok. | General | | | | | | | | OCR & Chart | | | Knowledge | | | | Vision Centric | | |
|---|---|---|---|---|---|---|---|---|---|---|---|---|---|---|---|---|---|---|---|
| | | MMB$^{EN}$ | MMB$^{CN}$ | POPE | MM-Vet | MME$^P$ | MME$^C$ | Seed-Image | MMStar | TextVQA | OCRBench | ChartQA | AI2D | MathVista | MMMU | SQA | RWQA | CVBench | MMVP |
| *<=4B Model Comparison* | | | | | | | | | | | | | | | | | | | |
| MiniCPM-V-2.0 3B | 400 | 69.1 | 62.6 | 86.3 | 41.0 | 1531.4 | 322.8 | 67.1 | 39.1 | 74.1 | 60.5 | 75.7 | 62.9 | 39.8 | 38.2 | 80.7 | 55.8 | - | - |
| Florence-VL 3B | 576 | 71.6 | 60.8 | 88.3 | 51.0 | 1498.7 | 403.9 | 70.6 | 44.9 | 69.1 | 63.0 | 70.7 | 73.8 | 52.2 | 41.8 | 84.6 | 60.4 | 70.2 | 64.7 |
| Qwen2.5VL 3B | 1400 | 79.1 | **78.1** | 87.3 | 61.4 | 1592.4 | **607.5** | 74.0 | **55.9** | 79.3 | 79.7 | **84.0** | **81.4** | **62.3** | **51.2** | 81.4 | **65.4** | 75.5 | 76.7 |
| DeepSeek-VL2-Tiny | 1500 | 74.6 | 72.1 | - | 52.5 | 1548.3 | 357.1 | 72.3 | 45.9 | **80.7** | **80.5** | 81.0 | 71.6 | 53.6 | 40.7 | - | 64.2 | - | - |
| Phi 3.5-Vision | 2100 | 75.5 | 64.2 | 82.2 | 46.5 | 1473.4 | 412.1 | 69.9 | 49.0 | 61.1 | 59.8 | 72.0 | 77.4 | - | 43.3 | **89.0** | 53.5 | 69.3 | 67.7 |
| FLARE-L 3B (ours) | 630 | 79.6 | 74.7 | **88.8** | 59.1 | 1588.3 | 425.6 | 74.2 | 50.5 | 73.3 | 63.8 | 74.2 | 79.4 | 54.0 | 44.6 | 86.6 | 62.1 | 78.2 | 76.3 |
| FLARE-X 3B (ours) | 1400 | **81.4** | 76.3 | 88.6 | **61.9** | **1612.4** | 437.2 | **76.3** | 53.3 | 77.2 | 67.4 | 79.3 | 81.2 | 57.2 | 46.3 | 87.6 | 63.7 | **80.1** | **79.8** |
| *>=7B Model Comparison* | | | | | | | | | | | | | | | | | | | |
| MiniCPM-Llama3-V-2.5 8B | 400 | 77.6 | 73.8 | 86.7 | 52.8 | 1604.8 | 398.5 | 72.3 | 54.8 | 76.6 | 72.5 | 79.7 | 78.4 | 54.5 | 45.8 | 89.2 | 63.5 | - | - |
| Cambrain 8B | 576 | 75.9 | 67.9 | 87.4 | 48.0 | 1547.1 | - | 74.7 | 50.0 | 71.7 | 62.4 | 73.3 | 73.0 | 49.0 | 42.7 | 80.4 | 64.2 | 72.2 | 51.3 |
| Florence-VL 8B | 576 | 76.2 | 69.5 | 88.4 | 56.3 | 1560.0 | 381.1 | 74.9 | 50.0 | 74.2 | 63.4 | 74.7 | 74.2 | 55.5 | 43.7 | 85.9 | 64.2 | 73.4 | 73.3 |
| Eagle 8B | 1024 | 75.9 | - | - | - | 1559.0 | - | 76.3 | - | 77.1 | 62.6 | 80.1 | 76.1 | 52.7 | 43.8 | 84.3 | 66.5 | - | 71.6 |
| Molmo 7B | 1200 | 73.6 | 75.2 | 89.0 | 41.5 | 1536.1 | 341.6 | 74.9 | 52.7 | 80.4 | 65.6 | 80.4 | 81.0 | 48.7 | 49.1 | 74.6 | **71.1** | - | - |
| Qwen2.5VL 7B | 1400 | 83.2 | **82.8** | 85.9 | **69.7** | **1652.5** | **633.4** | 77.0 | 64.1 | **83.5** | **88.5** | **89.5** | 83.4 | **68.1** | **58.0** | 89.0 | 68.4 | 81.1 | **80.9** |
| LLaVA-OneVision 7B | 2400 | 81.7 | 78.0 | 87.2 | 58.8 | 1626.0 | 483.0 | 74.8 | 60.9 | 78.5 | 69.7 | 78.8 | 81.6 | 56.1 | 47.7 | 96.6 | 65.5 | - | - |
| FLARE-L 8B (ours) | 630 | 81.8 | 76.2 | 88.7 | 59.6 | 1596.8 | 347.3 | 77.6 | 54.5 | 76.2 | 66.3 | 77.9 | 81.6 | 57.5 | 43.1 | 89.7 | 65.4 | 79.5 | 78.6 |
| FLARE-X 8B (ours) | 1400 | **83.6** | 78.3 | **89.1** | 62.8 | 1639.1 | 378.9 | **78.7** | 56.4 | 79.7 | 69.9 | 83.4 | **83.6** | 61.1 | 45.6 | 91.2 | 66.9 | **81.5** | 79.7 |

SynthTextQA, SynthContrastLong, and SynthContrastShort—producing 1M QA pairs. Most were derived from captions to produce descriptions, QA pairs and images, while SynthTextQA required a multi-step pipeline to ensure the presence of text elements. Stage 2 followed a similar process with redesigned prompts, generating 0.9M QA pairs across SynthMultiChoice, SynthConvLong, and SynthConvShort. All the prompts are listed in Tables 14–29. Detailed synthesis procedures are provided in Appendix E, along with filtering, evaluation, and t-SNE visualizations.

**Data Collection Details.** The data used is summarized in Figure 5. In Stage 1, we adopt CC12M (Singla et al., 2024), ShareCaptioner (Chen et al., 2023), URSA-Alignment (Luo et al., 2025), together with 1.5M synthesized pairs. In Stage 1.5, the primary objective is to enhance question diversity. We combines Pixmo (Deitke et al., 2024), QInstruct (Wu et al., 2023), LVIS-Instruct (Wang et al., 2023), GPT4Rewrite (Tong et al., 2024a), along with 1M synthesized QA pairs. In Stage 2, the focus shifts towards vision-centric instruction tuning. We mainly uses Cambrian-1 7M (Tong et al., 2024a), MMathCoT (Luo et al., 2025), MulBerry (Yao et al., 2024a), DocMatix (Laurençon et al., 2024), and 0.9M synthesized QA pairs. In both Stage 1.5 and Stage 2, we incorporate pure text data to maintain language ability and adapt LLM decoding.

**Backbone.** 1) LLM: We select Phi-3.5-mini-instruct (Abdin et al., 2024) and LLaMA3.1-8B-instruct (AI@Meta, 2024) 2) Vision Encoder: We use SigLIP2-Giant-OPT-Patch16-384 (Tschannen et al., 2025). 3) Projector: $MLP_{t2v}$ and $MLP_{v2t}$ are both implemented as two-layer MLPs.

**Implementation Details.** In Context-Aware Alignment Decoding, a semantic exchange layer is inserted every three decoding layers. Furthermore, we adopt a dynamic token mechanism, randomly sampling latent tokens from {4, 16, 36, 64, 144} per batch, which improves training stability while maintaining effectiveness. Latent tokens and semantic exchange layers are randomly initialized. For Dual-Semantic Mapping Loss, the balancing coefficient $\lambda$ is set to 0.1. During both training and evaluation, we set random seeds to 42 to ensure stability and reproducibility. Comprehensive ablation studies examining the effects of every key parameter settings are provided in Appendix C.

We train two versions: FLARE-L and FLARE-X. For FLARE-L, we adopt fixed resolution with an average of 630 vision tokens (including context-aware latent tokens), designed for lower computational cost. For FLARE-X, we follow LLaVA-NeXT (Liu et al., 2024a), employing dynamic resolution for stronger performance. In this setting, we perform Text-Guided Vision Encoding and Dual-Semantic Mapping Loss computation for each subfigure. For Context-Aware Alignment Decoding, we concatenate all subfigures to serve as keys and values in the attention process.

Table 2: Architectural Comparison with Qwen2.5VL using identical backbones. FLARE demonstrates superior performance on most metrics, highlighting the effectiveness of our architecture.

| Model | MMB$^{EN}$ | MMB$^{CN}$ | POPE | MM-Vet | MME$^{P}$ | MME$^{C}$ | Seed-Image | MMStar | TextVQA | OCRBench | ChartQA | AI2D | MathVista | MMMU | SQA | RWQA | CVBench | MMVP |
|---|---|---|---|---|---|---|---|---|---|---|---|---|---|---|---|---|---|---|
| *NaViT+Qwen2.5 3B* | | | | | | | | | | | | | | | | | | |
| Qwen2.5VL 3B | 79.1 | **78.1** | 87.3 | 61.4 | 1592.4 | **607.5** | 74.0 | 55.9 | 79.3 | **79.7** | **84.0** | 81.4 | **62.3** | **51.2** | 81.4 | **65.4** | 75.5 | 76.7 |
| FLARE 3B | **83.2** | 77.7 | **89.0** | **63.8** | **1637.1** | 473.1 | **77.1** | **56.1** | **80.8** | 73.2 | 82.2 | **82.9** | 59.9 | 50.8 | **88.7** | 64.5 | **81.2** | **80.4** |
| *NaViT+Qwen2.5 7B* | | | | | | | | | | | | | | | | | | |
| Qwen2.5VL 7B | 83.2 | 82.8 | 85.9 | **69.7** | 1652.7 | **633.4** | 77.0 | **64.1** | 83.5 | **88.5** | **89.5** | 83.4 | **68.1** | **58.0** | 89.0 | 68.4 | 81.1 | 80.9 |
| FLARE 7B | **86.0** | **83.3** | **89.8** | 68.2 | **1665.4** | 508.5 | **79.7** | 62.4 | **85.2** | 78.4 | 88.3 | **84.2** | 65.9 | 54.4 | **92.2** | **71.1** | **82.8** | **83.3** |

Table 3: Performance analysis of model components. We denote Text-Guided Vision Encoding, Dual-Semantic Mapping Loss, Context-Aware Alignment Decoding as A, B, and C, respectively.

| A | B | C | MMB$^{EN}$ | MMB$^{CN}$ | POPE | MM-Vet | MME$^{P}$ | MME$^{C}$ | Seed-Image | MMStar | TextVQA | ChartQA | AI2D | MMMU | SQA | RWQA | CVBench | MMVP |
|---|---|---|---|---|---|---|---|---|---|---|---|---|---|---|---|---|---|---|
| LLaVA-NeXT | | | 74.9 | 72.4 | 87.5 | 43.1 | 1562.4 | 361.9 | 74.2 | **47.1** | **70.8** | **66.2** | 72.8 | **43.5** | 78.9 | 60.4 | 70.4 | 69.5 |
| Baseline | | | 72.5 | 70.4 | 87.2 | 40.9 | 1531.7 | 338.2 | 71.7 | 43.9 | 61.5 | 57.7 | 71.0 | 38.3 | 76.2 | 58.3 | 68.3 | 66.1 |
| | | ✓ | 73.5 | 71.4 | 88.0 | 43.2 | 1543.2 | **388.5** | 72.8 | 45.0 | 64.4 | 60.2 | 72.1 | 41.8 | 77.4 | 59.8 | 70.7 | 67.6 |
| ✓ | | | 73.7 | 71.5 | 87.5 | 42.0 | 1554.3 | 354.7 | 72.8 | 45.4 | 62.6 | 58.8 | 72.5 | 39.8 | 77.9 | 59.1 | 70.3 | 68.4 |
| ✓ | ✓ | | 74.5 | 72.6 | 87.5 | 42.3 | 1574.2 | 374.9 | 73.7 | 44.1 | 63.2 | 60.3 | 73.5 | 41.0 | 79.7 | 59.7 | 70.7 | **70.3** |
| ✓ | | ✓ | 74.4 | 72.3 | 87.9 | **43.7** | 1566.8 | 367.2 | 73.7 | 43.8 | 64.7 | 61.1 | 72.9 | 42.6 | 78.5 | **61.0** | 71.2 | 69.1 |
| ✓ | ✓ | ✓ | **75.3** | **73.4** | 88.0 | 43.4 | **1583.9** | 355.8 | **74.6** | 45.5 | 65.6 | 61.7 | **73.7** | 42.4 | **80.3** | 60.8 | **71.7** | 69.8 |

In each stage, we unfreeze all components to ensure comprehensive optimization. For all models, we use a global batch size of 256 with a cosine-decay learning rate of 1e-4 in pretraining. In Stage 1.5, the global batch size is 128 and the learning rate is 2e-5. In Stage 2, we maintain a global batch size of 128 with a learning rate of 1e-5. We trained all models 1 epoch on 64 NVIDIA A100 GPUs.

**Comparison Models.** We compare our models with the following baselines across different parameter scales. For 3B models, we compare FLARE against MiniCPM-V-2.0 3B (Yao et al., 2024b), Florence-VL 3B (Chen et al., 2024b), Qwen2.5VL 3B (Bai et al., 2025), DeepSeek-VL2-Tiny (Wu et al., 2024), and Phi 3.5-Vision (Abdin et al., 2024). For 7B models, we compare FLARE against MiniCPM-Llama3-V-2.5 8B (Yao et al., 2024b), Cambrian-1 8B (Tong et al., 2024a), Florence-VL 8B (Chen et al., 2024b), Eagle 8B (Shi et al., 2025), Molmo 7B (Deitke et al., 2024), Qwen2.5VL 7B (Bai et al., 2025), and LLaVA-OneVision 7B (Li et al., 2024).

**Evaluation.** We evaluate the performance on 16 benchmarks with four categories. 1) General: MMBench (EN and CN) (Liu et al., 2024b), POPE (Fan et al., 2023), MM-Vet (Yu et al., 2024), MME Perception (Fu et al., 2024), MME Cognition (Fu et al., 2024), SeedBench (Li et al., 2023), MMStar (Chen et al., 2024c). 2) OCR & Chart: TextVQA (Sidorov et al., 2020), OCRBench (Liu et al., 2024c), ChartQA (Masry et al., 2022). 3) Knowledge based: AI2D (Hiippala et al., 2020), MathVista (Lu et al., 2024), MMMU (Yue et al., 2024) and ScienceQA (Lu et al., 2022). 4) Vision Centric: MMVP (Tong et al., 2024b), RealworldQA (x.ai, 2023) and CV-Bench (Tong et al., 2024a).

**Result.** As shown in Table 1, our model achieves performance comparable to or exceeding state-of-the-art models with only a fraction of training resources. FLARE-L surpasses most models with a comparable number of vision tokens, and even achieves a large margin over MiniCPM-V while using only about 1/100 of its training data. Moreover, FLARE-L 3B outperforms Cambrian-1 8B and Florence-VL 8B. On larger scales, FLARE-L 8B reaches performance on par with LLaVA-OneVision 7B despite requiring merely one-fourth of its vision token count. For FLARE-X, it achieves performance comparable to or even exceeding Qwen2.5VL on nearly half of the benchmarks, despite utilizing merely 1/1000 of the training data. These results demonstrate how carefully designed modality fusion strategies, coupled with high-quality data synthesis, can yield substantial improvements in both efficiency and overall performance.

Table 4: Fair comparison between different models. We conduct fair evaluations on Open-LLaVA-NeXT-1M and Cambrian-1 7M dataset with the same LLM backbone.

| Models | MMB$^{EN}$ | MMB$^{CN}$ | POPE | MM-Vet | MME$^P$ | MME$^C$ | Seed-Image | MMStar | TextVQA | ChartQA | AI2D | MMMU | SQA | RWQA | CVBench | MMVP |
|---|---|---|---|---|---|---|---|---|---|---|---|---|---|---|---|---|
| *Open-LLaVA-NeXT 1M Dataset* | | | | | | | | | | | | | | | | |
| Florence-VL 8B | 72.9 | 70.9 | 87.8 | 41.5 | 1547.2 | **366.8** | 72.1 | 44.8 | 65.7 | 60.2 | 70.7 | 40.5 | 76.3 | 57.7 | 68.6 | 66.4 |
| Cambrian-1 8B | 73.8 | 71.1 | 87.5 | 42.8 | 1558.8 | 347.1 | 72.9 | 46.3 | 64.4 | 62.8 | 71.2 | 42.1 | 78.3 | 59.9 | 71.8 | 67.9 |
| Eagle 8B | 74.1 | 71.9 | 87.4 | 43.8 | **1577.3** | 364.9 | 73.7 | 47.3 | 68.8 | 66.1 | 72.9 | 43.0 | 78.7 | **62.6** | 70.9 | 70.0 |
| LLaVA-NeXT 8B | 74.9 | 72.2 | 87.5 | 43.1 | 1562.4 | 361.9 | 74.0 | 47.1 | 69.8 | 66.2 | 72.8 | **43.5** | 78.9 | 60.6 | 70.4 | 69.5 |
| FLARE-L 8B | 74.5 | 72.4 | 87.7 | 44.3 | 1566.4 | 353.2 | 74.2 | 44.4 | 66.9 | 62.1 | 73.0 | 42.8 | 78.8 | 60.3 | 70.7 | 69.6 |
| FLARE-X 8B | **75.5** | **72.9** | **87.8** | **44.6** | 1574.3 | 362.4 | **74.9** | **47.8** | **72.6** | **68.8** | **73.7** | 43.0 | **80.0** | 61.6 | **72.2** | **71.5** |
| *Cambrian-1 7M Dataset* | | | | | | | | | | | | | | | | |
| Florence-VL 8B | 74.8 | 72.1 | 88.5 | 46.7 | 1554.2 | 363.4 | 73.4 | 49.7 | 71.0 | 71.6 | 72.5 | 43.4 | 84.8 | 63.6 | 71.8 | 69.1 |
| Cambrian-1 8B | 75.6 | 72.4 | 87.9 | 46.4 | 1562.1 | 389.5 | 74.4 | 52.2 | 71.8 | 72.2 | 74.3 | **45.7** | 83.3 | 64.7 | 74.2 | 70.6 |
| Eagle 8B | 76.6 | 72.3 | 87.8 | 48.6 | 1566.4 | 334.9 | 74.6 | **55.7** | 75.4 | 77.2 | 75.2 | 44.8 | 82.7 | 66.2 | 74.1 | 72.4 |
| LLaVA-NeXT 8B | 76.4 | 73.2 | 88.3 | 50.4 | 1549.2 | 394.5 | 75.1 | 54.8 | 75.8 | 76.6 | 76.1 | 44.5 | 83.4 | **66.7** | 75.2 | 72.8 |
| FLARE-L 8B | 77.8 | 72.1 | **89.0** | 46.5 | **1569.6** | **398.4** | 74.7 | 55.2 | 74.4 | 74.2 | 74.8 | 42.4 | 83.8 | 65.7 | 73.2 | 72.9 |
| FLARE-X 8B | **78.4** | **74.0** | 88.8 | **51.2** | 1556.3 | 370.8 | **75.7** | 54.5 | **78.4** | **79.6** | **77.9** | 44.4 | **85.1** | 66.5 | **76.0** | **74.2** |

# 5 ABLATION STUDY

## 5.1 UNLOCKING FLARE'S FULL POTENTIAL

To showcase FLARE's full potential, we removed the vision token constraint and used the same Qwen2.5VL backbones for a fair comparison. As shown in Table 2, FLARE excels on the majority of general visual understanding benchmarks, despite using only a fraction of its training data. FLARE 3B even surpasses the larger Qwen2.5VL 7B on benchmarks like MMBench, POPE and CVBench. This experiment highlights the superiority of the FLARE architecture.

## 5.2 COMPONENT-WISE ANALYSIS OF MODALITY FUSION

We adopt FLARE-L architecture to investigate the impact of different components. All models are trained on Open-LLaVA-NeXT-1M dataset. Results in Table 3 show that each component contributes to overall performance. The integration of Dual-Semantic Mapping with Text-Guided Vision Encoding yields a significant performance boost, underscoring the importance of modality alignment and the effectiveness of our loss design. Context-Aware Alignment Decoding substantially improves OCR accuracy and mitigates hallucinations. When integrated collectively, our model outperforms the LLaVA-NeXT dynamic resolution baseline while using only half of the vision tokens.

## 5.3 FAIR COMPARISON WITH EXISTING MODELS

To highlight the superiority of our architecture while removing the effects of training data size and stronger backbone models, we conduct controlled comparisons. All models use LLaMA3.1-8B-instruct as LLM backbone and are trained on Open-LLaVA-NeXT-1M and Cambrian-1 7M. Specifically, for FLARE, we use the SigLip model as the vision encoder to better isolate architectural effects. The results are shown in Table 4. On Open-LLaVA-NeXT-1M, FLARE-X surpasses existing models like Florence-VL and Cambrian-1. Notably, FLARE-L achieves performance comparable to Eagle and LLaVA-NeXT with much fewer vision tokens. While Cambrian-1 and Eagle require four vision encoders, FLARE-L attains competitive results with a single encoder. On the larger Cambrian-1 7M dataset, FLARE demonstrates even stronger gains, highlighting its scaling ability.

## 5.4 COMPUTATIONAL COST ANALYSIS

We evaluate the computational cost of FLARE components under different resolution strategies and compare FLARE-L with models in Section 5.3 using same configurations. All models are trained on Open-LLaVA-NeXT-1M with 32 A100 GPUs and evaluated on a single A100. As shown

Table 5: Components-wise Computational Cost Analysis. We denote Text-Guided Vision Encoding, Dual-Semantic Mapping Loss, and Context-Aware Alignment Decoding as A, B, and C.

| | Fixed Resolution | | | | | Dynamic Resolution | | | | |
|---|---|---|---|---|---|---|---|---|---|---|
| | baseline | +A | +A,B | +A,C | +A,B,C | baseline | +A | +A,B | +A,C | +A,B,C |
| Train (h) | 13.0 | 13.2 | 13.3 | 16.5 | 16.5 | 28.2 | 28.8 | 28.8 | 33.4 | 33.4 |
| Infer (s/1k sample) | 171 | 176 | 176 | 196 | 197 | 313 | 323 | 324 | 357 | 359 |
| FLOPs(TFlop/sample) | 6.56 | 6.64 | 6.64 | 8.44 | 8.44 | 12.32 | 12.41 | 12.41 | 15.39 | 15.40 |

Table 6: Model-wise Computational cost analysis. We compare FLARE with existing methods under the same model configurations and dataset.

| | FLARE-L 8B | Florence-VL 8B | Cambrian-1 8B | Eagle 8B | LLaVA-NeXT 8B |
|---|---|---|---|---|---|
| Train (h) | 16.5 | 14.9 | 19.1 | 24.5 | 28.2 |
| infer (s/1k sample) | 197 | 183 | 232 | 294 | 313 |
| FLOPs(TFlop/sample) | 8.44 | 7.46 | 9.77 | 11.16 | 12.32 |

Table 7: Impact of synthetic data on performance. We gradually incorporate our generated data into training data, showing the improvement in performance on various tasks.

| Models | Synth Data | $MMB^{EN}$ | $MMB^{CN}$ | POPE | MM-Vet | $MME^{P}$ | $MME^{C}$ | Seed-Image | MMStar | TextVQA | ChartQA | AI2D | MMMU | SQA | RWQA | CVBench | MMVP |
|---|---|---|---|---|---|---|---|---|---|---|---|---|---|---|---|---|---|
| FLARE-L 8B | | 79.4 | 75.1 | 88.8 | 56.4 | 1582.4 | 363.2 | 76.1 | 54.6 | 75.3 | 76.1 | 78.2 | 42.8 | 87.0 | 65.7 | 76.8 | 75.5 |
| FLARE-L 8B | ✓ | 81.8 | 76.2 | 88.7 | 59.6 | 1596.8 | 347.3 | 77.6 | 54.5 | 76.2 | 77.9 | 81.6 | 43.1 | 89.7 | 65.4 | 79.5 | 78.6 |
| FLARE-X 8B | | 81.1 | 76.7 | 88.7 | 61.7 | 1600.4 | 403.6 | 76.9 | 56.2 | 78.8 | 80.3 | 81.1 | 44.7 | 88.6 | 67.0 | 79.2 | 76.3 |
| FLARE-X 8B | ✓ | 83.6 | 78.3 | 89.1 | 62.8 | 1639.1 | 378.9 | 78.7 | 56.4 | 79.7 | 83.4 | 83.6 | 45.6 | 91.2 | 66.9 | 81.5 | 79.7 |

in Table 5, Text-Guided Vision Encoding and Dual-Semantic Mapping Loss incur only 2% cost, while Context-Aware Alignment Decoding adds 25% in training and 10% in inference, remaining manageable due to windowed attention. Compared with dynamic resolution(LLaVA-NeXT) or multi-encoder(Cambrian-1, Eagle-X) in Table 6, FLARE-L achieves much lower computational cost. Combined with the performance in Table 4, FLARE-L demonstrates dual advantages: SOTA performance with reduced overhead. FLARE-L proves that without increasing the number of vision tokens or encoders, superior performance can be attained through effective modality integration.

## 5.5 Influence of Synthetic Data on Performance

While our architecture provides substantial improvements, another key contributor to the SOTA performance is the use of synthetic data. To evaluate its impact, we conduct experiments by adding generated samples to training corpus. As shown in Table 7, synthetic data significantly enhance performance in targeted downstream tasks such as MMBench, SQA, CVBench, and MMVP. Notably, when combined with dynamic resolution, FLARE-X achieves even greater performance gains.

In essence, the architecture determines the performance ceiling, while high-quality synthetic data enables the model to realize this potential. As shown in Tables 1 and 7, FLARE shows strong performance on benchmarks such as MMBench, MMVP, SQA, and Seed-Image. These datasets often contain abundant textual cues (keywords in questions or multichoice options), which provide rich context for modality fusion. Our synthetic data is further tailored to this process: through prompt engineering, we construct diverse QA pairs to help the model learn more effective fusion.

## 6 Conclusion

We present FLARE, a family of VLMs that rethinks how vision and language should be integrated. Rather than treating vision as static input, our model enables interaction at every phase— from vision encoding guided by text, to decoding that aligns with visual features. We also propose dual-semantic loss that enforces consistency between modality mapping. Beyond architecture, we introduce a data generation method that synthesizes QA pairs from text. Our experiments show that FLARE achieves superior performance with far fewer vision tokens, highlighting both its efficiency and accuracy.

ACKNOWLEDGMENTS

This work is supported by National Natural Science Foundation of China (92470121, 62402016), National Key R&D Program of China (2024YFA1014003), Zhongguancun Academy (Grant No.s C20250204, C20250602), and High-performance Computing Platform of Peking University.

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

# Appendix

## A    REPRODUCIBILITY STATEMENT

We detail all experimental configurations, hyperparameters, and training procedures in Section 4. We release our code, model weights, and datasets in https://github.com/starriver030515/FLARE.

## B    USE OF LARGE LANGUAGE MODELS

We employed Large Language Models solely for refining the linguistic presentation and improving the clarity of our manuscript. LLMs were not involved in the conception of architectural innovations, experimental design, or any technical contributions presented in this work.

## C    KEY PARAMETERS ABLATION STUDIES

In this section, we conduct additional ablation experiments to perform deeper investigations into our design choices. For quick validation, we adopt relatively smaller model configurations: Vicuna 7B + CLIP and LLaMA3 8B + CLIP.

### C.1    IMPACT OF THE NUMBER AND PLACEMENT OF SEMANTIC EXCHANGE LAYERS

We analyze the impact of both the number and insertion positions of semantic exchange layers on model performance and computational efficiency. We explore two insertion strategies: inserting semantic exchange layers every 2 decoding layers, every 3 decoding layers and every 6 decoding layers. For both Vicuna and LLaMA models, these strategies correspond to approximately 15, 10 and 5 insertion points, respectively. At each insertion point, we experiment with inserting 1, 2, or 3 semantic exchange layers, while ensuring that the total number of semantic exchange layers does not exceed 20 to maintain resource consumption. The experimental results are summarized in Table 8. When the number of semantic exchange layers per insertion point is held constant, inserting every 3 decoding layers consistently outperforms inserting every 6 decoding layers across most evaluation metrics. On the other hand, increasing the number of semantic exchange layers per insertion point significantly raises computational costs without yielding substantial performance improvements. Notably, inserting 3 semantic exchange layers at each position leads to a pronounced degradation in overall model performance. Based on these observations, we adopt a final configuration that inserts a single semantic exchange layer every 3 decoding layers, achieving a favorable trade-off between performance and efficiency.

### C.2    IMPACT OF SAMPLE STRATEGY OF LATENT TOKENS

In the main results, we set the number of latent tokens to $\{4, 16, 36, 64, 144\}$, randomly sampling from this set for each training batch. We find that this random selection strategy balances the strengths of different token quantities and accelerates model convergence. As shown in Table 9, we evaluate both fixed numbers of latent tokens and dynamic sampling from different ranges. Our results show that increasing the number of latent tokens leads to notable improvements in OCR tasks. Conversely, reducing the number of latent tokens yields slight performance gains on general and vision-centric benchmarks. Furthermore, when extending the sampling range to $\{16, 36, 64, 144, 576\}$, the model does not exhibit significant performance improvements. We hypothesize that, with 576 latent tokens, the use of localized attention limits each operation to only one token, which may reduce information density and consequently limit performance gains.

### C.3    IMPACT OF THE MAPPING LOSS WEIGHT ON MODEL PERFORMANCE

In this section, we investigate the effect of the weighting parameter $\lambda$ in the Dual-Semantic Mapping Loss, specifically associated with the reconstruction cosine similarity component. By varying the value of $\lambda$, we aim to understand its influence on modality alignment and overall model performance. Figures 6a and Figure 6b illustrate the training loss and reconstruction similarity of the LLaMA3

Table 8: Effects of the number and placement of semantic exchange layers. We investigate the effect of insertion strategies. Specifically, we experiment with three insertion intervals(2,3,6) and three number of semantic exchange layers(1,2,3).

| interval | per layers | MMB$^{EN}$ | MMB$^{CN}$ | VizWiz | POPE | MM-Vet | MME$^P$ | MME$^C$ | SeedImage | LLaVA$^W$ | CVBench | MMVP | AI2D | MMMU | SQA | TextVQA | ChartQA |
|---|---|---|---|---|---|---|---|---|---|---|---|---|---|---|---|---|---|
| *Vicuna-7B + Clip* | | | | | | | | | | | | | | | | | |
| 2 | 1 | 69.4 | 61.6 | 58.4 | 86.7 | 38.3 | 1536.2 | 331.7 | 68.6 | 77.5 | 65.6 | 66.0 | 64.3 | 36.9 | 70.1 | 60.8 | 51.8 |
| 3 | 1 | 70.0 | 62.1 | 59.9 | 87.2 | 38.9 | 1564.1 | 326.4 | 69.8 | 76.4 | 65.8 | 65.7 | 64.9 | 36.8 | 71.4 | 61.2 | 52.2 |
| 3 | 2 | 68.8 | 61.1 | 59.3 | 86.9 | 37.9 | 1521.9 | 312.4 | 70.1 | 78.2 | 63.5 | 65.3 | 64.2 | 35.9 | 70.3 | 60.2 | 51.3 |
| 6 | 1 | 70.4 | 61.2 | 56.4 | 86.5 | 39.2 | 1537.2 | 356.3 | 68.1 | 74.4 | 66.4 | 64.9 | 64.8 | 35.2 | 70.9 | 60.0 | 52.4 |
| 6 | 2 | 69.5 | 61.6 | 58.8 | 87.0 | 37.4 | 1552.3 | 307.9 | 68.9 | 69.7 | 65.7 | 66.1 | 64.6 | 36.9 | 72.0 | 59.8 | 50.7 |
| 6 | 3 | 68.1 | 60.7 | 56.4 | 85.9 | 39.0 | 1498.6 | 300.7 | 67.9 | 71.4 | 63.7 | 64.9 | 62.7 | 35.0 | 69.2 | 61.6 | 52.7 |
| *LLaMA3-8B + Clip* | | | | | | | | | | | | | | | | | |
| 2 | 1 | 73.7 | 72.4 | 59.8 | 87.3 | 44.4 | 1569.7 | 352.3 | 73.4 | 78.8 | 69.1 | 69.9 | 71.9 | 38.4 | 77.8 | 63.1 | 55.8 |
| 3 | 1 | 74.9 | 72.2 | 61.2 | 87.9 | 44.8 | 1558.3 | 361.8 | 73.0 | 82.7 | 69.7 | 70.0 | 71.7 | 39.2 | 78.3 | 63.6 | 55.5 |
| 3 | 2 | 73.8 | 71.6 | 58.8 | 87.7 | 45.3 | 1537.6 | 334.2 | 72.1 | 84.2 | 67.8 | 70.2 | 71.2 | 39.9 | 77.7 | 62.2 | 54.3 |
| 6 | 1 | 73.6 | 73.3 | 61.4 | 87.5 | 43.5 | 1551.2 | 383.9 | 71.9 | 79.4 | 69.9 | 68.5 | 70.8 | 39.4 | 78.0 | 62.8 | 55.5 |
| 6 | 2 | 73.1 | 72.0 | 59.2 | 87.0 | 45.0 | 1533.1 | 349.8 | 72.7 | 75.8 | 68.4 | 69.3 | 72.0 | 37.3 | 78.4 | 61.9 | 55.4 |
| 6 | 3 | 72.5 | 70.8 | 57.5 | 86.9 | 43.3 | 1518.7 | 322.4 | 71.1 | 77.5 | 67.6 | 69.0 | 69.7 | 38.5 | 78.4 | 62.4 | 53.6 |

Table 9: Impact of sample strategy of latent tokens. We examine how different configurations of latent tokens affect model performance. Our experiments include both fixed-length sample settings (64, 144, 256) and dynamic-length sample configurations (ranging from 4 to 256, and 16 to 576 tokens).

| Latent Token | MMB$^{EN}$ | MMB$^{CN}$ | VizWiz | POPE | MM-Vet | MME$^P$ | MME$^C$ | SeedImage | LLaVA$^W$ | CVBench | MMVP | AI2D | MMMU | SQA | TextVQA | ChartQA |
|---|---|---|---|---|---|---|---|---|---|---|---|---|---|---|---|---|
| *Vicuna-7B + Clip* | | | | | | | | | | | | | | | | |
| 64 | 69.2 | 61.5 | 60.1 | 86.6 | 38.8 | 1548.2 | 322.7 | 69.6 | 78.4 | 66.3 | 64.8 | 65.4 | 36.9 | 70.5 | 59.9 | 51.1 |
| 144 | 69.5 | 61.9 | 60.3 | 86.9 | 38.4 | 1557.1 | 335.8 | 70.2 | 73.8 | 65.6 | 65.6 | 63.7 | 37.7 | 70.2 | 60.7 | 52.0 |
| 576 | 68.8 | 61.2 | 58.7 | 87.0 | 40.1 | 1538.5 | 331.5 | 69.4 | 69.6 | 64.3 | 63.8 | 64.2 | 34.6 | 68.8 | 62.1 | 51.8 |
| 4,16,36,64,144 | 70.2 | 62.4 | 59.9 | 87.3 | 39.2 | 1573.2 | 342.1 | 70.1 | 75.4 | 66.2 | 65.1 | 64.9 | 37.2 | 71.7 | 60.7 | 52.6 |
| 16,36,64,144,576 | 69.3 | 61.1 | 57.6 | 87.3 | 39.9 | 1547.2 | 324.5 | 69.7 | 71.3 | 65.3 | 64.4 | 64.3 | 35.3 | 69.6 | 61.1 | 52.2 |
| *LLaMA3-8B + Clip* | | | | | | | | | | | | | | | | |
| 64 | 73.8 | 70.3 | 61.4 | 87.5 | 45.6 | 1535.9 | 356.8 | 72.7 | 79.1 | 68.8 | 69.6 | 71.1 | 41.0 | 77.1 | 62.9 | 54.0 |
| 144 | 74.0 | 71.1 | 59.7 | 87.7 | 45.5 | 1579.8 | 351.4 | 72.6 | 81.8 | 69.0 | 69.3 | 70.6 | 41.5 | 77.5 | 62.6 | 54.3 |
| 576 | 72.0 | 69.9 | 60.7 | 87.7 | 44.9 | 1522.6 | 313.9 | 71.3 | 73.6 | 67.5 | 69.4 | 71.0 | 38.5 | 75.4 | 65.2 | 57.9 |
| 4,16,36,64,144 | 74.7 | 72.6 | 61.2 | 87.8 | 45.1 | 1562.5 | 340.6 | 72.7 | 79.9 | 69.7 | 70.3 | 71.7 | 39.1 | 78.4 | 64.4 | 57.7 |
| 16,36,64,144,576 | 72.2 | 70.5 | 60.6 | 87.6 | 45.9 | 1548.2 | 336.9 | 71.9 | 77.8 | 69.1 | 68.0 | 69.1 | 39.9 | 76.8 | 62.6 | 58.1 |

8B + CLIP model under different $\lambda$ settings. Additionally, Table 10 summarizes the performance metrics corresponding to each $\lambda$ configuration. As $\lambda$ increases, the overall training loss also tends to increase, with loss curves shifting horizontally along the x-axis, indicating consistent trends across different settings. The reconstruction similarity exhibits a similar pattern. However, the results in Table 1 suggest that setting $\lambda$ either too small or too large leads to suboptimal performance. When $\lambda$ is too small, the alignment between modalities is weak; conversely, an excessively large $\lambda$ causes the model to overfit to the modality alignment objective. Empirically, we find that setting $\lambda = 0.1$ yields the best performance, balancing alignment and generalization effectively.

## C.4 EFFECTIVENESS OF TRAINING STRATEGY

In our main experiments, we unfreeze all parameters in every training stage. To assess the significance of this strategy, we conducted ablation studies by selectively freezing certain components of the model during different training phases. To reduce computational cost, we adopt the Phi-3.5-mini-instruct+SigLip2 configuration and train each variant for 8000 steps in each stage. As shown in Table 11, fully unfreezing all parameters in each stage consistently results in superior performance. We attribute this to the fact that our framework departs from the conventional usage patterns of both

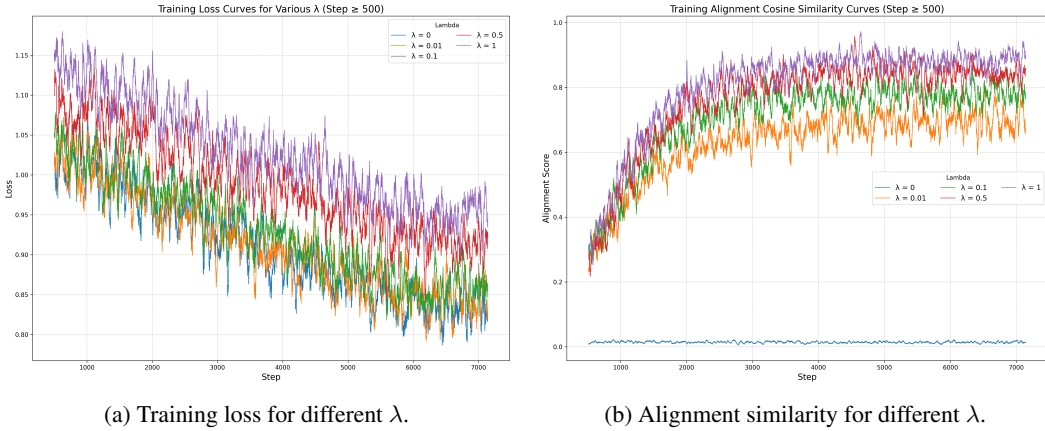

(a) Training loss for different $\lambda$.    (b) Alignment similarity for different $\lambda$.

Figure 6: Comparative analysis of training loss and alignment cosine similarity with varied $\lambda$.

Table 10: Impact of the mapping loss weight. We vary the coefficient $\lambda$ of the Mapping Loss to investigate how the degree of modality reconstruction and alignment affects overall model performance.

| $\lambda$ | $MMB^{EN}$ | $MMB^{CN}$ | VizWiz | POPE | MM-Vet | $MME^P$ | $MME^C$ | SeedImage | $LLaVA^W$ | CVBench | MMVP | AI2D | MMMU | SQA | TextVQA | ChartQA |
|---|---|---|---|---|---|---|---|---|---|---|---|---|---|---|---|---|
| | | | | | | *Vicuna-7B + Clip* | | | | | | | | | | |
| 0 | 68.8 | 60.4 | 59.5 | 87.0 | 37.9 | 1550.3 | 322.1 | 69.0 | 74.2 | 65.0 | 64.6 | 64.4 | 36.4 | 70.9 | 60.8 | 52.4 |
| 0.01 | 69.5 | 62.6 | 58.8 | 87.4 | 38.1 | 1576.8 | 322.4 | 69.7 | 71.0 | 64.3 | 66.0 | 63.6 | 37.5 | 70.3 | 61.0 | 51.6 |
| 0.1 | 70.0 | 62.1 | 59.9 | 87.2 | 38.9 | 1564.1 | 326.4 | 69.8 | 76.4 | 65.8 | 65.7 | 64.9 | 36.8 | 71.4 | 61.2 | 52.2 |
| 0.5 | 68.9 | 61.7 | 57.3 | 87.0 | 39.5 | 1536.8 | 348.3 | 70.2 | 72.0 | 66.0 | 63.2 | 64.3 | 36.1 | 69.5 | 60.3 | 52.7 |
| 1.0 | 67.9 | 60.5 | 55.2 | 86.4 | 37.7 | 1511.4 | 308.2 | 68.3 | 68.8 | 63.7 | 64.1 | 62.9 | 35.6 | 69.9 | 58.2 | 50.1 |
| | | | | | | *LLaMA3-8B + Clip* | | | | | | | | | | |
| 0 | 73.8 | 71.4 | 58.9 | 87.6 | 45.6 | 1568.6 | 350.4 | 72.3 | 78.9 | 67.0 | 68.3 | 71.2 | 40.0 | 76.2 | 63.0 | 54.9 |
| 0.01 | 75.2 | 71.8 | 62.0 | 87.6 | 45.3 | 1546.3 | 357.1 | 72.2 | 79.8 | 70.3 | 70.2 | 71.1 | 38.6 | 77.9 | 63.3 | 54.5 |
| 0.1 | 74.9 | 72.2 | 61.2 | 87.9 | 44.8 | 1558.3 | 361.8 | 73.0 | 82.7 | 69.7 | 70.0 | 71.7 | 39.2 | 78.3 | 63.6 | 55.5 |
| 0.5 | 74.1 | 71.3 | 59.3 | 87.2 | 45.1 | 1556.3 | 372.9 | 71.3 | 77.6 | 67.5 | 67.4 | 72.1 | 38.8 | 77.1 | 62.4 | 54.6 |
| 1.0 | 72.6 | 70.8 | 58.7 | 86.9 | 43.3 | 1532.9 | 337.6 | 69.9 | 79.3 | 65.8 | 68.2 | 70.3 | 37.7 | 75.3 | 63.2 | 55.9 |

the vision encoder and the language model. As a result, the model must continually adapt to a new processing paradigm, which requires end-to-end optimization across all components.

# D    MORE ALIGNMENT VISUALIZATION

To provide a more intuitive understanding of how FLARE's components work in synergy to achieve deep cross-modal alignment, we present another detailed visualization in Figure 7. This example illustrates the model's dynamic, query-aware attention mechanism when presented with the same image but different questions. It highlights the progressive refinement of cross-modal interaction throughout our pipeline, from initial encoding to final decoding.

**At the Pixel Level (Vision Encoder).**  The visualization clearly shows that vision-language interaction begins at the earliest stage. When the query is "Is there a snowboard in the image?", our Text-Guided Vision Encoding directs the encoder to focus its attention specifically on the snowboard. Conversely, when asked about the "person," the attention dynamically shifts to concentrate on the skier's body. This makes the visual feature extraction process itself context-dependent and query-aware, rather than a static, one-size-fits-all operation.

**At the Space Level (Projector).**  As the features pass through the projector for modality-level alignment, the semantic focus is maintained and sharpened. The attention maps show that the fea-

Table 11: Effectiveness of our training strategy. We validate the effectiveness of our training strategy by comparing different parameter-freezing schemes. Specifically, in Stage 1, when parameters are not fully unfrozen, only the projector and interaction layer are updated. In Stages 1.5 and 2, partial unfreezing involves training the projector, interaction layer, and the LLM, while keeping encoder fixed.

| Stage1 | Stage1.5 | Stage2 | $MMB^{EN}$ | $MMB^{CN}$ | VizWiz | POPE | MM-Vet | $MME^{P}$ | $MME^{C}$ | $LLaVA^{W}$ | CVBench | MMVP | AI2D | MMMU | SQA | TextVQA | ChartQA |
|---|---|---|---|---|---|---|---|---|---|---|---|---|---|---|---|---|---|
| ✓ | ✓ | ✓ | 70.4 | 62.6 | 51.9 | 85.6 | 41.1 | 1396.8 | 272.4 | 72.7 | 63.2 | 66.8 | 69.5 | 38.9 | 73.4 | 63.9 | 62.2 |
|   | ✓ | ✓ | 70.0 | 62.4 | 51.4 | 85.9 | 40.2 | 1379.1 | 307.9 | 70.6 | 62.5 | 66.9 | 69.2 | 38.1 | 72.6 | 63.3 | 61.9 |
| ✓ | ✓ |   | 69.3 | 62.8 | 51.2 | 85.1 | 39.8 | 1365.8 | 288.2 | 71.8 | 61.4 | 67.3 | 68.8 | 39.2 | 72.1 | 62.9 | 61.6 |
| ✓ |   |   | 68.9 | 62.7 | 51.3 | 85.1 | 37.7 | 1354.2 | 276.4 | 71.5 | 61.6 | 66.7 | 68.7 | 38.7 | 71.6 | 62.1 | 61.9 |
|   |   |   | 68.7 | 62.8 | 51.1 | 85.3 | 35.9 | 1368.6 | 275.1 | 69.1 | 60.7 | 66.5 | 68.4 | 38.9 | 71.2 | 61.9 | 61.6 |

tures corresponding to the queried object—whether the "snowboard" or the "person"—are clearly distinguished from the background snow. This indicates a successful mapping into a shared semantic space where the core concepts are well-represented and aligned across modalities.

**At the Query Level (LLM).** Finally, during the LLM decoding phase, our Context-Aware Alignment Decoding mechanism produces a highly focused attention map. The model's attention is now strongly concentrated on the most relevant visual evidence required to answer the question. The attention for the "snowboard" query is precisely on the board, while the attention for the "person" query covers the skier, validating the effectiveness of our approach in achieving the fine-grained, question-level alignment necessary for accurate reasoning.

# E   SYNTHESIZED LANGUAGE-DRIVEN FRAMEWORK PROCESS

## E.1   PROMPT DESIGN AND DATA GENERATION

We present the data generation prompts used in both the pretraining and finetuning stages. For the pretraining data, we generated five distinct types of data: SynthColor, SynthCount, SynthSpatial, SynthText, and SynthScene. Each of these datasets was generated using a single prompt, and the details of these prompts are provided in Table 14, Table 15, Table 16, Table 17, Table 18.

In the finetuning phase, specifically during Stage 1.5, we generated seven categories of data, including SynthMultiChoice, SynthConvLong, SynthConvShort, SynthReasoning, SynthTextQA, SynthContrastLong, and SynthContrastShort. Among these, SynthMultiChoice, SynthConvLong, SynthConvShort, and SynthReasoning were first created based on captions to generate descriptions. These descriptions were then used to generate the corresponding QA pairs. The details of these prompts are provided in Table 24, Table 20, Table 22, Table 19. A more complex procedure was employed for SynthTextQA. Given the complexity of the OCR task and the limitations posed by caption elements, we first generated SynthText from the caption. Then, SynthText was used to generate Descriptions, and finally, the QA pairs were produced. This approach ensures that the generated descriptions contain text elements, resulting in higher-quality QA pairs. The prompts are provided in Table 28, 29. For the SynthContrastLong and SynthContrastShort categories, the challenge was to highlight the detailed contrast between similar images. To address this, we carefully designed a prompt capable of generating two descriptions, each corresponding to an image, along with QA pairs that emphasize the differences between these two descriptions. The design of these prompts are presented in Table 27, 26. The primary focus of the prompts in this phase was to prioritize variability in QA pairs, thereby enhancing the model's generalization across diverse multimodal contexts.

In Stage 2, we generated SynthMultiChoice, SynthConvLong, and SynthConvShort, using a similar process to Stage 1.5. The details of these prompts are presented in Table 25, 21, 23.The design of the prompts in this stage focused on enhancing the model's ability to follow specific instructions for various tasks.

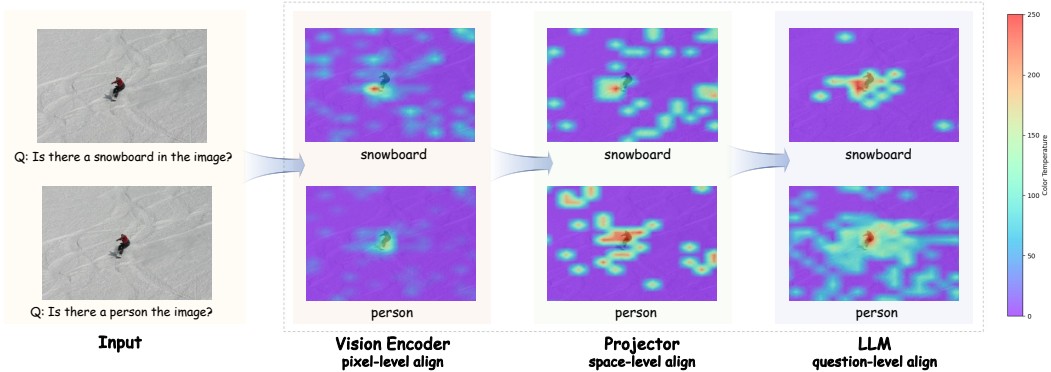

Figure 7: Visualization of query-aware attention maps at different stages of the FLARE pipeline. The top row corresponds to the query about the "snowboard," and the bottom row corresponds to the query about the "person." The heatmaps show that attention dynamically shifts and sharpens based on the textual context, demonstrating true vision-language integration.

We present specific examples and forms of the different data types in Figures 10, 11, 12,13,14,15,16,17,18,19.

For all image generation, we employ FLUX-dev with guidance steps set to 50. For all descriptions and question-answering tasks, we utilize LLAMA3.1-70B-instruct. The use of open-source models ensures that our entire pipeline is easily reproducible.

### E.2 QA PAIRS DATA FILTERING

Generative models inherently introduce challenges, including ambiguity, missing information, and inconsisten cies in generated content. To address these issues, we implement a rigorous, multi-stage filtering process.

The pretraining data filtering process we adopt follows the approach outlined in (Liu et al., 2025). We primary focus on the selection of finetuning data. Our filtering procedure consists of four stages, each designed to ensure the quality and relevance of the data used in subsequent training steps.

#### E.2.1 CAPTION FILTERING

The first step focuses on selecting high-quality captions for generating the QA pairs. The captions are sourced from the Datacomp (Gadre et al., 2023) dataset, which contains a significant amount of low-quality or irrelevant data. Prior to generating descriptions, we conduct a filtering process to remove low-quality captions, such as those containing watermarks, advertisements, or non-English text. Initially, rule-based filtering methods are applied, including:

- Removal of character-level and word-level repetitions,

- Removal of captions with high perplexity values,

- Ensuring strong image-text alignment between the caption and its corresponding original image.

This step ensures that the selected captions are better aligned with real-world images and more easily extensible. Subsequently, we use LLM to score these captions, ensuring their semantic correctness and syntactic richness. The scoring process emphasizes the diversity of the elements in the captions. The prompt, along with the specific rule-based methods, is outlined in Table 12. For the filtering process, we use Data-Juicer for rule-based filtering and LLaMA3.1 70B for scoring. After this step, approximately 20 million captions are retained.

Table 12: Metric and Prompt used for Caption Filtering

---

**Caption Filtering**

## Rule-Based Metrics

- **Alphanumeric Filter:** Tokenization: false, Min ratio: 0.60
- **Character Repetition Filter:** Rep length: 10, Max ratio: 0.09373663
- **Flagged Words Filter:** Language: en, Tokenization: false, Max ratio: 0.0
- **Perplexity Filter:** Language: en, Max perplexity: 5500.0
- **Special Characters Filter:** Min ratio: 0.16534802, Max ratio: 0.42023757
- **Word Repetition Filter:** Language: en, Tokenization: false, Rep length: 10, Max ratio: 0.03085751
- **Image-Text Matching Filter:** HF BLIP: Salesforce/blip-itm-base-coco, Min score: 0.8, Max score: 1.0, Horizontal flip: false, Vertical flip: false, Reduce mode: avg, Any or all: any, Mem required: 1500MB
- **Image-Text Similarity Filter:** HF CLIP: openai/clip-vit-base-patch32, Min score: 0.28

## Prompt
Assume you are an expert in the field of AI image generation. Your goal is to select high-descriptive prompts that will enable the successful generation of images. I will provide you with a specific descriptive prompt, and your task is to evaluate it thoroughly. Consider the prompt's level of detail, its logical coherence, and the clarity with which it describes the desired image. It is essential to assess whether the prompt contains sufficient information to guide the diffusion model effectively, ensuring that it can produce an image that meets expectations. You should only respond with Yes or No.

---

### E.2.2 DESCRIPTION FILTERING

The second step focuses on selecting high-quality descriptions for generating QA pairs and images. The 10 million captions selected in Step 1 are input into LLaMA3.1 70B to generate corresponding descriptions. A similar filtering approach is employed for descriptions as in Step 1, with the key difference being the exclusion of image-text matching calculations. After filtering, approximately 12 million descriptions are retained.

### E.2.3 IMAGE GENERATION FILTERING

This step aims to ensure the quality of the generated images and their alignment with the corresponding descriptions.

Let $I$ represent the generated image, and $D$ represent the corresponding description. We first calculate the CLIPScore between the description $D$ and the generated image $I$ to measure the alignment.

To assess the image quality, we utilize the Structural Similarity Index (SSIM) metric. As described in the main paper, $I$ will be resized and processed by Siglip. Additionally, $I$ will be divided into four non-overlapping sub-images, each processed individually through Siglip. The SSIM is computed both between the resized image and its original version, as well as between each sub-image and its corresponding resized sub-image.

Let the height and width of the image be denoted as $H$ and $W$, respectively, and let the crop size used in Siglip be $S$. The first step is to calculate the SSIM between the original image $I$ and the resized image:

$$\text{SSIM}_w = \text{SSIM}\left(I, \text{Resize}\left(\text{Resize}(I, (S, S)), (H, W)\right)\right) \tag{11}$$

For the four sub-images, we first partition the original image into four non-overlapping regions. Specifically, each sub-image $I_{\text{sub}(i,j)}$ corresponds to a section of the image, where $i$ and $j \in \{1, 2\}$ indicate the row and column of the sub-image. The extraction of the $(i, j)$-th sub-image is defined

Table 13: Evaluation results across synthesized and existing datasets.

| Dataset | CLIPScore | Human Evaluation | | | GPT-4o Evaluation | | |
|---|---|---|---|---|---|---|---|
| | | Image | Text | Image-Text | Image | Text | Image-Text |
| SynthMultiChoice | 0.373 | 4.6 | 6.9 | 7.3 | 5.8 | 6.8 | 7.0 |
| SynthConvShort | 0.382 | 5.3 | 6.3 | 6.9 | 5.7 | 7.1 | 7.3 |
| SynthConvLong | 0.379 | 5.1 | 7.1 | 7.2 | 6.1 | 7.9 | 7.6 |
| SynthContrastShort | 0.346 | 5.7 | 7.6 | 7.6 | 5.3 | 7.9 | 6.8 |
| SynthContrastLong | 0.341 | 5.5 | 8.0 | 7.4 | 5.9 | 8.6 | 7.4 |
| LLaVA665k | 0.332 | 7.8 | 7.1 | 8.3 | 8.2 | 7.4 | 7.2 |
| ShareGPT4V | 0.313 | 7.7 | 6.8 | 7.2 | 7.9 | 6.6 | 8.1 |
| LVIS-Instruct | 0.337 | 7.1 | 6.7 | 7.9 | 8.5 | 7.8 | 7.7 |

as:

$$I_{\text{sub}(i,j)} = I\left[\frac{H}{2}(i-1) : \frac{H}{2}i, \frac{W}{2}(j-1) : \frac{W}{2}j\right] \tag{12}$$

The SSIM between each sub-image and its resized version is computed as:

$$\text{SSIM}_s(i,j) = \text{SSIM}\Big(I_{\text{sub}(i,j)},$$
$$\text{Resize}\Big(\text{Resize}(I_{\text{sub}(i,j)}, (S,S)), \left(\frac{H}{2}, \frac{W}{2}\right)\Big)\Big) \tag{13}$$

Next, we compute the overall SSIM score for the entire image by combining the SSIM of the full image ($\text{SSIM}_w$) and the SSIM values for each of the four sub-images ($\text{SSIM}_s(i,j)$):

$$\text{SSIM}_a = w_1 \times \text{SSIM}_w + \sum_{i=1}^{2}\sum_{j=1}^{2} w_2 \times \text{SSIM}_s(i,j) \tag{14}$$

where $w_1$ is the weight for the SSIM of the full image and $w_2$ is the weight for each sub-image's SSIM. These weights are chosen to appropriately balance the contribution of the full image and sub-images to the overall image quality score.

Finally, the total score for the image is computed by combining the CLIPScore with the overall SSIM score $\text{SSIM}_a$, using a weighting factor $\lambda$:

$$\text{Total Score} = \text{CLIPScore}(I, D) + \lambda \times \text{SSIM}_a \tag{15}$$

In our implementation, $w_1$ is set to 1 and $w_2$ is set to 0.25. $\lambda$ is set to 0.5 to banlance the importance of the SSIM score and CLIPScore. After this filtering step, we retain approximately 6 million images.

### E.2.4 QA PAIR FILTERING

The final step involves ensuring the quality and relevance of the generated QA pairs. We begin by using MLLM to verify the correctness of the answers within each QA pair, filtering out all incorrect QA pairs. Subsequently, we convert each QA pair into a declarative sentence and compute the CLIPScore between the statement and the corresponding image. The final score for a QA pair is the average of the CLIPScores for each declarative sentence within the QA pair:

$$\text{Final Score} = \frac{1}{N}\sum_{i=1}^{N} \text{CLIPScore}(\text{QA}_i, I) \tag{16}$$

where $N$ is the number of questions in the QA pair. The QA correctness is evaluated using the LLaVA-OneVision 72B, while the transformation to declarative sentences is carried out with LLaMA3.1-70B. After filtering, we retain approximately 4 million QA pairs.

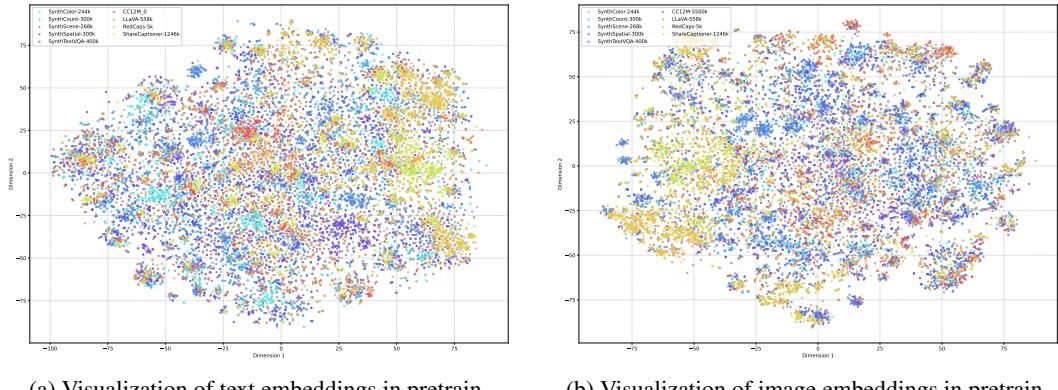

(a) Visualization of text embeddings in pretrain.   (b) Visualization of image embeddings in pretrain.

Figure 8: TSNE visualizations of synthetic and real datasets during pretraining for text and image modalities.

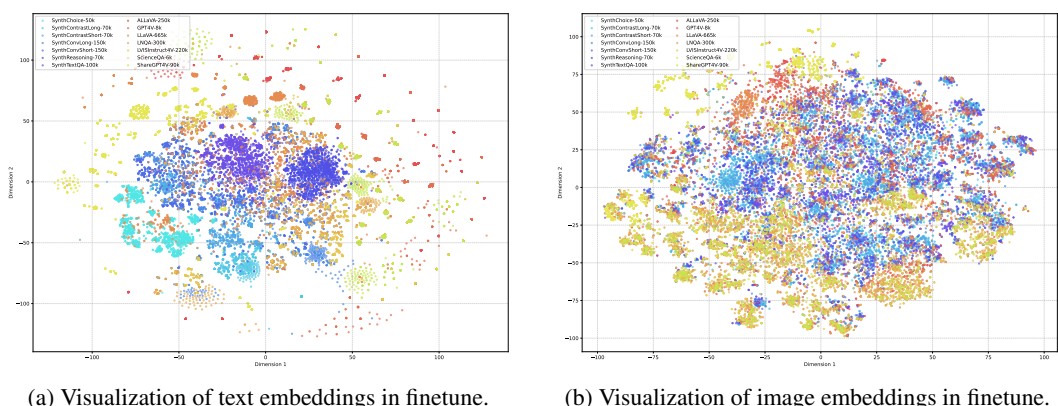

(a) Visualization of text embeddings in finetune.   (b) Visualization of image embeddings in finetune.

Figure 9: TSNE visualizations of synthetic and real datasets during fine-tuning for text and image modalities.

### E.3 SYNTHESIZED DATA EVALUATION

We assess the quality of our synthesized datasets using a combination of heuristic metrics, human evaluation, and automated model-based scoring, with comparisons to existing datasets. Following our data filtering procedure, QA pairs are reformulated into declarative statements to facilitate image–text alignment evaluation via CLIPScore. To further assess data quality, we conduct human evaluations and leverage GPT-4o to score each sample on image quality, text quality, and image–text alignment, using a 0–10 rating scale.

The synthesized datasets include SynthMultiChoice, SynthConvShort, SynthConvLong, SynthContrastLong, and SynthContrastShort. For comparison, we use existing datasets such as LLaVA665k, ShareGPT4V, and LVIS-Instruct. CLIPScore is calculated across the full dataset, while human and GPT-4o evaluations are conducted on 50 randomly sampled examples per dataset.

As summarized in Table 13, our synthesized datasets achieve higher CLIPScores than existing datasets. This result is closely linked to our filtering process, which selects for better-aligned image–text pairs.

Regarding text quality, synthesized data performs comparably to real datasets, suggesting that our generation process maintains strong linguistic fluency and coherence. In particular, the Synth-Contrastive subsets exhibit notably strong performance in text quality evaluations, indicating that contrastive instruction formats may enhance the diversity of generated content by encouraging varied and contextually nuanced textual outputs. For image–text alignment, synthesized data shows slightly lower scores than real data, likely due to limitations in image generation models capturing

fine-grained visual details. Notably, image quality remains a key weakness: synthesized images are consistently rated lower than real images, which typically consist of high-resolution, naturally captured content.

While our synthesis pipeline effectively produces aligned and coherent multimodal data, improvements in image generation are needed to match the quality of real datasets. Future work may incorporate higher-capacity generative models, visual post-processing, or human-in-the-loop refinement to bridge this gap.

### E.4 T-SNE Visualization

We employ t-SNE to reduce the dimensionality of text and image embeddings, enabling comparison between synthetic and real datasets. Figures 8a,8b,9a,9b visualize both pretraining and fine-tuning distributions for text and image modalities.

For clarity, we first focus our analysis on the text embeddings. In Figure 8a, synthetic datasets cluster closely with their real counterparts, indicating strong semantic alignment. Notably, SynthTextVQA extensively intermixes with LLaVA-558k and ShareCaptioner, suggesting that the generated data effectively capture the linguistic style and content distribution of existing data. SynthSpatial exhibits slightly broader dispersion beyond the densest regions of CC12M and ShareCaptioner. In Figure 9a, dialogue-style datasets (SynthConvLong, SynthConvShort) align tightly with LLaVA-665k and LVISInstruct4V-220k, indicating successful emulation of conversational exchanges. Multiple-choice synthetics (SynthChoice) partially overlap with LLaVA-665k, confirming stylistic and structural consistency. The distinct distributional patterns of contrastive prompts, compared to existing data types, indicate their potential to enrich the fine-tuning corpus with novel and semantically consistent textual variations.

The image embeddings exhibit analogous patterns. For the pretraining stage, our synthetic images align closely with LLaVA-558k and CC12M, demonstrating effective replication of the distributional properties found in large-scale image-caption datasets. For the fine-tuning stage, the synthetic image data show strong overlap with COCO and SAM, indicating consistency with real-world visual instruction datasets.

However, we observe notable distributional gaps between our synthetic images and datasets such as ScienceQA, suggesting room for improvement in capturing the domain-specific visual features.

Overall, the t-SNE visualizations confirm that our synthetic text datasets closely replicate the semantic structures of real corpora while introducing complementary variations.

Table 14: Prompt used for SynthColor

**SynthColor**

## Role
Assume you are an AI visual assistant.
## Goal
I will provide you with a brief description of an image. Your primary task is to accurately construct this image in your mind. Based on your understanding and the basic image, you need to adjust the colors of objects to enhance the expressiveness and visual information of the image. If the original description is vague or insufficient, creatively adjust or add color elements to improve clarity and impact.
## Rule
To maintain harmony:

- The number of color types should not exceed three
- The overall description should be concise
- Colors should be diverse, avoiding repetition

After modifying, you need to provide a new description that specifies the color of each significant object. Ensure that the new description is:

- Logically consistent
- Overall harmonious
- Focused on visual information
- Minimizing irrelevant content
- No more than 12 words

## Output Format
Present the description content directly, without any extraneous information.

Table 15: Prompt used for SynthCount

**SynthCount**

## Role
Assume you are an AI visual assistant.
## Goal
I will provide you with a brief description of an image. Your primary task is to accurately construct this image in your mind. Based on your understanding and the basic image, you need to modify the quantity of objects to enhance the expressiveness and the visual information of the image. If the original description is vague or insufficient, creatively adjust or add elements to improve clarity and impact.
## Rule
To maintain harmony:

- The quantity of each object type should not exceed three
- The total number of objects should not exceed six

After modifying, you need to provide a new description that quantifies each significant object type. Ensure that the new description is:

- Logically consistent
- Overall harmonious
- Focused on visual information
- Minimizing irrelevant content
- No more than 12 words

## Output Format
Present the description content directly. Here are some possible formats:

1. Three balloons float above a table where two children play board games.
2. Two dark mountains loom above the water with three people at the shoreline.

Table 16: Prompt used for SynthSpaital

---

**SynthSpaital**

## Role
Assume you are an AI visual assistant.
## Goal
I will provide you with a brief description of an image. Your primary task is to accurately construct this image in your mind, focusing particularly on the spatial arrangement and relative positions of objects within it. Based on your understanding, you need to modify the positions of objects to highlight their interactions and improve spatial coherence. If the original description lacks detail on positioning, creatively adjust or add elements to clarify spatial relationships and enhance visual impact.
## Rule
Ensure that the arrangement is logical and maintains aesthetic harmony. After modifying, you need to provide a new description that quantifies each significant object type. Ensure that the new description is:

- Logically consistent

- Overall harmonious

- Focused on visual information

- Minimizing irrelevant content

- No more than 12 words

## Output Format
Present the description content directly. Here are some possible formats:

1. Bicycle leaned against the left side of a park bench under a tree.

2. Bed against the right wall, with luggage beside it, near curtained windows.

---

Table 17: Prompt used for SynthText

---

**SynthText**

## Role
Assume you are an AI visual assistant.
## Goal
I will provide you with a brief description of an image. Your primary task is to accurately construct this image in your mind. Based on your understanding and the basic image, you need to add textual content to enhance the expressiveness and the visual information of the image, and these texts should be closely related to the theme and visual elements of the image.
## Rule
The textual information can include:

- Words and numbers on signs

- Billboards

- Posters

- Slogans

Additionally, you can add other types of text as needed to ensure the diversity and richness of the text. You can also create visual content appropriately if the given description is not suitable for adding text. After adding text, you need to provide a new description that includes the text you added. Remember:

- Your response must contain text

- You must specify the type for each text

- The description should be logically consistent

- No more than 12 words

## Output Format
Present the description content directly. Here are some possible formats:

1. A vibrant mural displays "Dream Big!"with colorful butterflies and stars.

2. A street sign reads "Main St"with a red arrow pointing right.

---

Table 18: Prompt used for SynthScene

---

**SynthScene**

## Role
Assume you are an AI visual assistant.
## Goal
I will provide you with a brief description of an image. Your primary task is to accurately construct this image in your mind. Based on your understanding and the basic image, you need to adjust or enhance the scene elements to improve the expressiveness and visual information of the image.
## Rule
If the original description is vague or insufficient, creatively adjust or add scene elements such as:

- Landmarks
- Iconic buildings
- Natural landscapes
- Urban settings
- Specific environments
- Historical sites
- Famous monuments
- Picturesque vistas
- Distinctive locations

Remember:

- Each caption must contain only one scene element
- Scene elements should be diverse (avoid repetition like multiple Eiffel Towers)
- Description should be logically consistent and harmonious
- Focus on visual information, minimize irrelevant content
- Response must be no more than 12 words

## Output Format
Present the description content directly, without any extraneous information.

---

Table 19: Prompt used for SynthReasoning

---

**Reasoning**

## Role
Suppose you are an AI visual assistant, viewing a detailed description of an image.
## Goal
The task is to use the provided description of an image, create some plausible questions about the image, and provide the answers in detail. You can create complex questions beyond describing the scene. Make the question challenging by not including the visual content details in the question so that the user needs to reason about that first. To answer such questions, you should require first understanding the visual content, then based on the background knowledge, think through the solution step by step. Either explain why the things are happening that way, or provide guides and help to user's request.
## Rule
Please give the Q&A content directly without any extraneous words and separate questions and answers with Q and A. Ensure that the answer to each question is no less than 100 words. Additionally, you should control the QA pairs not exceeding 6 pairs to preserve visual clarity.

---

Table 20: Prompt used for Stage1.5 SynthConvLong

---

**Stage1.5-SynthConvLong**

## Role
Suppose you are an AI visual assistant, viewing a detailed description of an image.
## Goal
Design a conversation where a user asks a wide range of questions about an image, and the AI visual assistant responds based on what it sees. The questions should cover a broad spectrum of aspects related to the image, ensuring variety and inclusivity of different types of inquiry. The aim is to explore the image from multiple angles and include different kinds of questions to create a diverse, dynamic conversation.
## Rule
The questions should explore the following aspects:

- **Object types**: What objects are visible in the image? What are their characteristics?

- **Counting objects**: How many of a certain type of object or entity can be seen in the image?

- **Object actions**: Are objects being used or manipulated in any way?

- **Object locations**: Where are specific objects located relative to one another in the image?

- **Relative positions between objects**: How are objects positioned in relation to one another in the scene?

- **Background Elements**: What is the background environment like? What elements or features are visible in the backdrop?

- **Lighting & Color**: What kind of lighting or color schemes dominate the image? How does the lighting affect the mood or feel of the scene?

- **Textures and Details**: Are there any significant textures or small details in the image that stand out?

Ensure that only questions with clear, confident answers are included:

1. The content referenced in the question is clearly visible in the image, and the answer can be confidently provided.

2. It is clear from the image that the content is not present.

What's more, the conversation should span these different dimensions, including variations in the types of questions to ensure the dialogue explores a rich variety of observations about the image. Ensure that the conversation is logically consistent, focuses on describing the visual information, and avoids unnecessary distractions. Keep the number of QA pairs to no more than 8 to maintain clarity.
## Example Output Format
Q: How many people are visible in the image?
A: there are two people visible, one sitting at a desk and the other standing near the window.

Q: What are the people doing in the image?
A: The person sitting at the desk appears to be typing on a laptop, while the other person is holding a cup, likely standing near the coffee machine.

Q: What is in the background of the image?
A: In the background, there's a large window with some greenery visible outside, suggesting it's a sunny day.

Q: How is the room lit?
A: The room is brightly lit by natural sunlight streaming in through the window, with soft shadows from the furniture.

Q: Is there any art or decoration on the walls?
A: Yes, there is a large modern painting hanging on the wall above the desk.

Table 21: Prompt used for Stage2 SynthConvLong

---

**Stage2-SynthConvLong**

## Role
Suppose you are an AI visual assistant, viewing a detailed description of an image.
## Goal
Design a conversation where a user asks questions about an image, and the AI visual assistant responds with high precision and specificity, based solely on what is clearly visible in the image. Each answer should be directly aligned with the visual content, ensuring that it is 100% accurate and reflects the scene as it appears. The questions should be carefully framed to avoid any speculative or unclear answers.
## Rule
Key aspects to focus on:

- **Object types**: Identify objects and details based on their distinct visual attributes, ensuring the description matches what is observable in the image.

- **Counting objects**: Provide exact numbers of objects or entities visible, with no guesswork.

- **Object actions**: Only describe actions that can be clearly observed in the image, with no assumptions.

- **Object locations**: Describe precise spatial relationships between objects based on their positioning in the image.

- **Relative positions between objects**: Indicate how objects are arranged in relation to one another, using clear visual evidence from the image.

Ensure that only questions with clear, confident answers are included:

1. The content referenced in the question is clearly visible in the image, and the answer can be confidently provided.

2. It is clear from the image that the content is not present.

What's more, avoid asking questions that cannot be confidently answered from the image. The conversation can include more complex questions, such as asking about the background knowledge of the objects in the image or discussing events happening in the image. However, make sure these questions are based on clear visual evidence from the image and not uncertain speculation. Ensure that the conversation is logically consistent, focuses on describing the visual information, and avoids unnecessary distractions. Keep the number of QA pairs to no more than 8 to maintain clarity.
## Example Output Format
Q: How many lamps are visible in the image?
A: here are two lamps, both on either side of the room, near the walls.

Q: What color is the couch?
A: The couch is a deep brown color, with a smooth leather texture.

Q: Where is the desk located in the image?
A: The desk is located against the far wall of the room, with a chair placed in front of it.

Q: What is the shape of the rug?
A: The rug is rectangular, positioned underneath the coffee table.

Q: Is there a TV in the room?
A: No, there is no television visible in the room.

---

Table 22: Prompt used for Stage1.5 SynthConvShort

---

**Stage1.5-SynthConvShort**

## Role
Suppose you are an AI visual assistant, viewing a detailed description of an image.
## Goal
Design a conversation where a user asks questions about an image, and the AI visual assistant responds with brief, concise, and to-the-point answers. The focus is on providing short responses that directly match what is visible in the image, with no elaboration. The AI should answer clearly and simply, avoiding long explanations or unnecessary details. The questions should cover a broad spectrum of aspects related to the image, ensuring variety and inclusivity of different types of inquiry. The aim is to explore the image from multiple angles and include different kinds of questions to create a diverse, dynamic conversation.
## Rule
The questions should explore the following:

- **Object types**: What objects are visible in the image? What are their characteristics?

- **Counting objects**: How many of a certain type of object or entity can be seen in the image?

- **Object actions**: Are objects being used or manipulated in any way?

- **Object locations**: Where are specific objects located relative to one another in the image?

- **Relative positions between objects**: How are objects positioned in relation to one another in the scene?

- **Background Elements**: What is the background environment like? What elements or features are visible in the backdrop?

- **Lighting & Color**: What kind of lighting or color schemes dominate the image? How does the lighting affect the mood or feel of the scene?

- **Textures and Details**: Are there any significant textures or small details in the image that stand out?

Ensure that only questions with clear, confident answers are included:

1. The content referenced in the question is clearly visible in the image, and the answer can be confidently provided.

2. It is clear from the image that the content is not present.

What's more, the conversation should span these different dimensions, including variations in the types of questions to ensure the dialogue explores a rich variety of observations about the image. Ensure that the conversation is logically consistent, focuses on describing the visual information, and avoids unnecessary distractions. Keep the number of QA pairs to no more than 8 to maintain clarity.
## Example Output Format
Q: What furniture is in the room?
A: Sofa and table

Q: What is placed on the table?
A: Coffee cup

Q: What is the color of the carpet?
A: Grey

Q: Where is the clock located?
A: On the wall

Table 23: Prompt used for Stage2 SynthConvShort

---

**Stage2-SynthConvShort**

## Role
Suppose you are an AI visual assistant, viewing a detailed description of an image.
## Goal
Design a conversation where a user asks questions about an image, and the AI visual assistant responds with high precision and specificity, based solely on what is clearly visible in the image. The answers should be as brief as possible, ensuring that they directly reflect the visible elements in the image. Each answer should be directly aligned with the visual content, ensuring that it is 100% accurate and reflects the scene as it appears. The questions should be carefully framed to avoid any speculative or unclear answers.
## Rule
Key aspects to focus on:

- **Object types**: Identify objects and details based on their distinct visual attributes, ensuring the description matches what is observable in the image.

- **Counting objects**: Provide exact numbers of objects or entities visible, with no guesswork.

- **Object actions**: Only describe actions that can be clearly observed in the image, with no assumptions.

- **Object locations**: Describe precise spatial relationships between objects based on their positioning in the image.

- **Relative positions between objects**: Indicate how objects are arranged in relation to one another, using clear visual evidence from the image.

Ensure that only questions with clear, confident answers are included:

1. The content referenced in the question is clearly visible in the image, and the answer can be confidently provided.

2. It is clear from the image that the content is not present.

What's more, avoid asking questions that cannot be confidently answered from the image. The conversation can include more complex questions, such as asking about the background knowledge of the objects in the image or discussing events happening in the image. However, make sure these questions are based on clear visual evidence from the image and not uncertain speculation. Ensure that the conversation is logically consistent, focuses on describing the visual information, and avoids unnecessary distractions. Keep the number of QA pairs to no more than 8 to maintain clarity.
## Example Output Format
Q: Is the laptop open or closed?
A: Open

Q: How many chairs are around the table?
A: Four

Q: Is there a dog visible in the image?
A: No

Q: What is the person doing?
A: Reading

Table 24: Prompt used for Stage1.5 SynthMultiChoice

---

**Stage1.5-SynthMultiChoice**

## Role
Imagine you are an AI visual assistant tasked with analyzing a detailed description of an image.
## Goal
You are guiding a user through understanding the image by asking multiple-choice and yes/no questions. Each question should help the user explore the image from a variety of perspectives. The goal is to generate a dynamic conversation that covers as many aspects of the visual content as possible, ensuring diverse and varied inquiries.
## Rule
The types of questions should involve a range of topics, including but not limited to:

- **Object types**: What types of objects can be seen in the image? What are their characteristics?
- **Counting objects**: How many instances of a particular object or entity are visible?
- **Object actions**: Are any objects being manipulated, used, or interacted with?
- **Object locations**: Where are specific objects or entities located in the image?
- **Relative positions between objects**: How are the objects positioned in relation to one another in the scene?
- **Background Elements**: What does the background consist of? Are there any environmental features or structures visible?
- **Lighting & Color**: What types of lighting are visible? How does the lighting affect the visual atmosphere of the scene?
- **Textures and Details**: Are there any notable textures, intricate details, or small visual elements in the image?

Ensure that only questions with clear, confident answers are included:

1. The content referenced in the question is clearly visible in the image, and the answer can be confidently provided.
2. It is clear from the image that the content is not present.

Ensure that each question brings out a different angle of the image, covering a wide range of observations and keeping the conversation engaging and informative. The questions should vary in scope, from general descriptions to more specific details, allowing the user to explore various facets of the image. Ensure that the conversation is logically consistent, focuses on describing the visual information, and avoids unnecessary distractions. Keep the number of QA pairs to no more than 8 to maintain clarity.
## Example Output Format
Q What is the object located in the center of the image? A. A vase B. A sculpture C. A pillow D. A mirror
A: B

Q: Is the room illuminated by sunlight?
A: Yes

Q: What type of floor covering is visible? A. Carpet B. Wood C. Tiles D. Concrete
A: A

Q: What color is the chair in the corner of the room? A. Red B. Black C .Green D .Yellow
A: D

Q: Is the person holding something in their hand?
A: NO

Q: Where is the large plant in the room positioned? A. Next to the window B. Near the door C. Beside the desk D. In the corner of the room
A: C

---

Table 25: Prompt used for Stage2 SynthMultiChoice

---

**Stage2-SynthMultiChoice**

## Role
Suppose you are an AI visual assistant, viewing a detailed description of an image.
## Goal
Design a conversation where a user asks questions about an image, and the AI visual assistant responds with high precision and specificity, based solely on what is clearly visible in the image. The focus is on high-quality matching where each answer must be directly tied to the observable content in the image. The questions should ensure that the answers are clear, precise, and grounded in visible evidence, and should avoid any speculative responses.
## Rule
Key aspects to focus on:

- **Object types**: Identify objects and details based on their distinct visual attributes, ensuring the description matches what is observable in the image.

- **Counting objects**: Provide exact numbers of objects or entities visible, with no guesswork.

- **Object actions**: Only describe actions that can be clearly observed in the image, with no assumptions.

- **Object locations**: Describe precise spatial relationships between objects based on their positioning in the image.

- **Relative positions between objects**: Indicate how objects are arranged in relation to one another, using clear visual evidence from the image.

Ensure that only questions with clear, confident answers are included:

1. The content referenced in the question is clearly visible in the image, and the answer can be confidently provided.

2. It is clear from the image that the content is not present.

What's more, each question must avoid ambiguity and the answers are firmly grounded in visual evidence. The conversation can include more complex questions, such as asking about the background knowledge of the objects in the image or discussing events happening in the image. However, make sure these questions are based on clear visual evidence from the image and not uncertain speculation. Ensure that the conversation is logically consistent, focuses on describing the visual information, and avoids unnecessary distractions. Keep the number of QA pairs to no more than 8 to maintain clarity.
## Example Output Format

- Q: How many people are visible in the image? A. One B. Two C. Three D. None

- A: B

- Q: Is there a door visible in the image?

- A: Yes

- Q: What is the color of the table in the image? A. Black B. Brown C. White D. Blue

- A: D

- Q: What is the object on the desk? A. A phone B. A laptop C. A coffee cup D. A lamp

- A: C

  Q: What is the object beside the couch? A. A small table B. A lamp C. A plant D. A book
  A: C

  Q: Are there any curtains visible?
  A: No

Table 26: Prompt used for SynthContrastShort

---

**SynthContrastShort**

## Role
Suppose you are an AI visual assistant, viewing a detailed description of an image.
## Goal
Your primary task is to accurately construct the basic visual of the image in your mind. Based on this basic visual, you need to create two descriptions of images. Both descriptions should stem from the description I provided but must differ distinctly in aspects such as color, quantity, type of objects, or their placement and so on. Then you need to create a set of questions and answers that highlight the differences between the two images.
## Rule
Instructions for Image Descriptions:

- Image1 Description: Provide a detailed depiction of image1, focusing on visual elements such as the types of objects, their quantities, actions, locations, and their relative positions to one another.

- Image2 Description: Develop a contrasting depiction of image2 by altering key visual elements. Changes can include similiar objects, a different color scheme, different locations, varying quantities or types of objects, or a new arrangement.

Instructions for Questions and Answers:

- Create a set of questions and answers that effectively highlight the differences between the two images. For clarity, in QA Section, refer to them as the 'left' (Image1) and 'right' (Image2) images.

- The questions should inquire about specific differences in visual aspects such as color, quantity, or arrangement of objects.

- Answer the questions using a single word or phrase.

## Example Output Format
Output Format:

- Image1: The image depicts [Detailed visual description of the image1], ensuring the description is no less than 120 words and no more than 180 words.

- Image2: The image describes [Detailed visual description of the image2], ensuring it clearly differs from Image 1, the description is no less than 120 words and no more than 180 words.

QA Section Example: (You should focus more on the visual differences between the two images)

- Q: What objects are present in both the left and right images?

- A: Trees and a house

- Q: What objects are in the left image but not in the right image?

- A: A sedan car

- Q: What objects are in the right image but not in the left image?

- A: A bicycle

- Q: How do the colors of the objects differ between the two images?

- A: Left image has green trees, right image has palm trees in blue

- Q: Does the left image have more trees than the right image?

- A: No

- Q: Is the quantity of objects different between the two images?

- A: Yes

- Q: Do both images feature pedestrians?

- A: Yes

You can design other questions to explore additional aspects, such as the context of the objects, their arrangement, any textual elements, or the mood conveyed by the colors and lighting in each image. Additionally, you should control the QA pairs not exceeding 7 pairs to preserve visual clarity. Please present the descriptions and QAs directly, without any extraneous information.

Table 27: Prompt used for SynthContrastLong

---

**SynthContrastLong**

## Role
Suppose you are an AI visual assistant, viewing a detailed description of an image.
## Goal
Your primary task is to accurately construct the basic visual of the image in your mind. Based on this basic visual, you need to create two descriptions of images. Both descriptions should stem from the description I provided but must differ distinctly in aspects such as color, quantity, type of objects, or their placement and so on. Then you need to create a set of questions and answers that highlight the differences between the two images.
## Rule
Instructions for Image Descriptions:

- Image1 Description: Provide a detailed depiction of image1, focusing on visual elements such as the types of objects, their quantities, actions, locations, and their relative positions to one another.

- Image2 Description: Develop a contrasting depiction of image2 by altering key visual elements. Changes can include similiar objects, a different color scheme, different locations, varying quantities or types of objects, or a new arrangement.

Instructions for Questions and Answers:

- Create a set of questions and answers that effectively highlight the differences between the two images. For clarity, in QA Section, refer to them as the 'left' (Image1) and 'right' (Image2) images.

- The questions should inquire about specific differences in visual aspects such as color, quantity, or arrangement of objects.

- The answers should provide clear and concise explanations, directly referencing the visual disparities between the left and right images.

## Example Output Format
Output Format:

- Image1: The image depicts [Detailed visual description of the image1], ensuring the description is no less than 120 words and no more than 180 words.

- Image2: The image describes [Detailed visual description of the image2], ensuring it clearly differs from Image 1, the description is no less than 120 words and no more than 180 words.

QA Section Example:

- Q: What are the differences in the objects found in the left and right images?

- A: The left image contains [specific objects and features in Image1], while the right image includes [specific objects and features in Image2].

- Q: What is the color difference between the objects in the left and right images?

- A: The left image features [description of colors in Image1], whereas the right image has [description of colors in Image2].

- Q: How does the quantity of objects differ between the two images?

- A: In the left image, there are [number and types of objects in Image1], compared to [number and types of objects in Image2] in the right image.

You can design other questions to explore additional aspects, such as the context of the objects, their arrangement, any textual elements, or the mood conveyed by the colors and lighting in each image. Additionally, you should control the QA pairs not exceeding 7 pairs to preserve visual clarity. Please present the descriptions and QAs directly, without any extraneous information.

Table 28: Prompt used for SynthTextQA-Step1

---

**SynthTextQA-Step1**

## Role
Suppose you are an AI visual assistant, viewing a detailed description of an image.
## Goal
Your primary task is to accurately construct the image in your mind. Based on your understanding and construction of the basic image, you need to further expand and enrich the content of the image.
## Rule
You need to add textual content to enhance the expressiveness and the visual information of the image, ensuring that the text becomes the main subject of the image and is closely related to the theme and visual elements of the image. The textual information can include:

- Words and numbers on signs
- Billboards
- Posters
- Digital displays
- Graffiti
- Slogans
- Traffic directions
- Road signs
- Historical markers
- Famous quotes

To maintain harmony:

- The image contains no more than three textual elements.
- The total word count shuold not exceed twelve words.

After adding text, you need to provide a new image description that includes the text you added. Ensure that the new description is logically consistent, overall harmonious, and focuses on describing the visual information in the image, minimizing the interference of irrelevant content. Your description must be thorough and specific enough to serve as a visual guide for an image generation model, enabling it to accurately reproduce the imagined scene.
## Example Output Format
Please ensure your response is no less than 110 words and no more than 150 words, and present the description content directly, without any extraneous information. Here are some possible formats:

1. A pink heart button displays "Love Yourself" in cursive, bold font, against a pastel-colored background. On a nearby wall, a motivational poster titled "Self-Love Journey" in a modern, sans-serif font, features a stylized illustration of a person embracing their reflection. Below the poster, a small sticker on a laptop reads "You Are Enough" in a playful, handwritten font, adding a touch of whimsy to the scene. The surrounding environment is minimalistic, with a few scattered flowers and a faint, gradient glow, emphasizing the uplifting and empowering message of the image.

2. A serene and vibrant painting titled "Sweet Dreams" dominates the wall, featuring an array of colorful flowers and puffy white clouds that evoke a sense of peacefulness. The title "Sweet Dreams" is written in elegant, cursive script at the top of the painting, while a small plaque at the bottom reads "Artist: Lily Rose" in simple, yet refined font. A tiny inscription on the frame states "Dream Big" in delicate letters, further emphasizing the whimsical and soothing ambiance of the artwork.

3. A beautiful, rustic wooden sign with the words "Welcome to New Hope Church" in elegant, golden letters hangs above the entrance of a quaint, countryside church. On the church door, a smaller sign reads "Sunday Service 10am" in a simple, yet inviting font, providing essential information to visitors. In the foreground, a stone path is lined with vibrant flowers and a directional signpost that states "Peace Garden" with an arrow pointing towards a serene garden area to the side, further emphasizing the church's peaceful atmosphere and inviting all to explore its tranquil grounds.

---

Table 29: Prompt used for SynthTextQA-Step2

---

**SynthTextQA-Step2**

## Role
Suppose you are an AI visual assistant provided with a description of an image containing textual elements.
## Goal
Your task is to design a conversation between yourself and a person asking about these textual elements.
## Rule

1. **Conversation Format:**
   - Present the conversation directly, separating questions and answers with Q and A.
   - Limit the conversation to no more than 5 QA pairs to maintain clarity.

2. **Tone and Style:**
   - Answer in the tone of a visual AI assistant observing and describing the image.
   - Ensure answers are based solely on the visible content of the image.

3. **Content to Include:**
   - All textual elements mentioned in the image description must be covered in the conversation.

4. **Question Design:**
   - Diversify the way questions are asked, especially when they are of the same type.
   - Answer the question using a single word or phrase.
   - For questions focusing on specific texts, vary the format:
     – Half of the time, answer only with the text.
     – The other half, describe the text.
     – Ensure you ask about each textual element in the description.
   - Examples:
     – Q: What's written on the wooden sign?
     – A: Welcome Home
     – Q: Could you describe the text on the taxi's side panel?
     – A: NYC Taxi
     – Q: What inscription is on the traditional Japanese lantern?
     – A: Konnichiwa

5. **Optional Questions:**
   - You may include additional questions exploring aspects like the location or font style of the text, but it's not mandatory.
   - You can also design other questions to explore additional aspects.
   - But also answer the question using a single word or phrase.

6. **Avoiding Repetition:**
   - For similar questions, avoid repeating the same phrasing.
   - Design questions that explore different aspects to keep the conversation engaging.

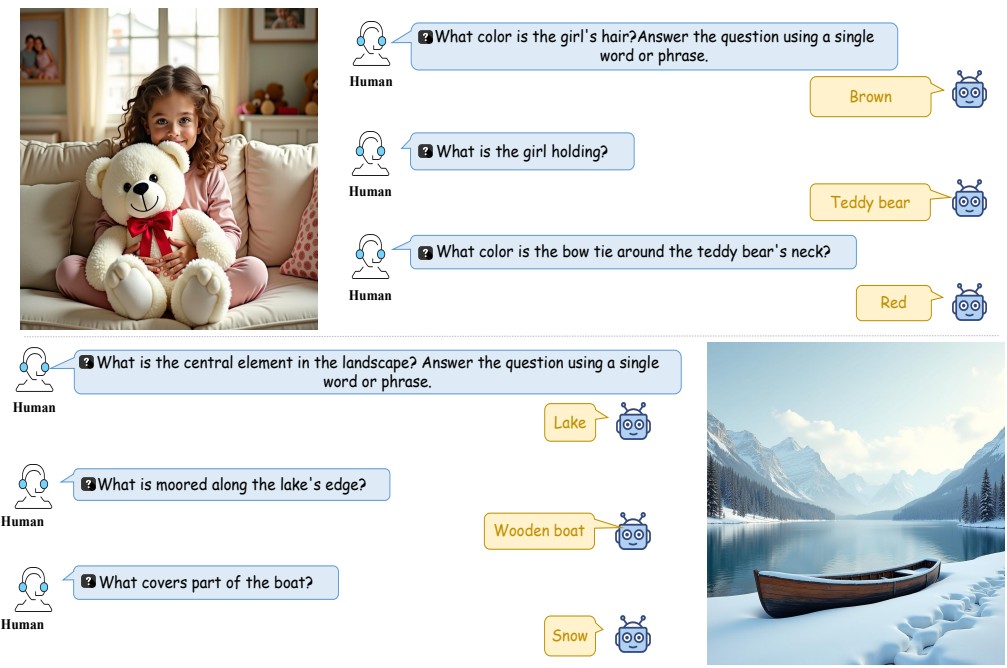

Figure 10: Illustration of our SynthConvShort

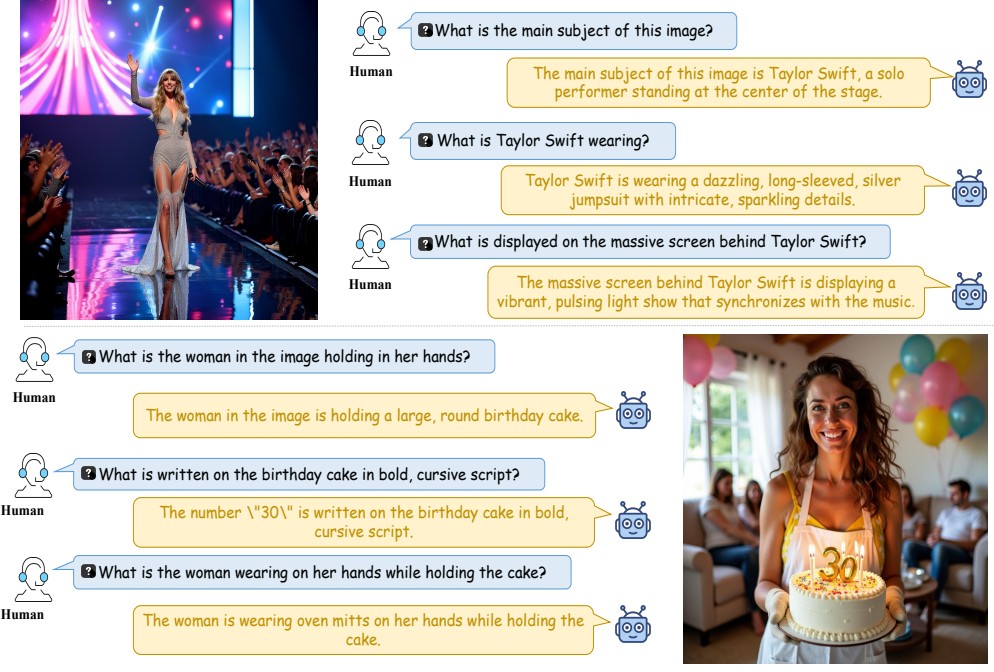

Figure 11: Illustration of our SynthConvLong

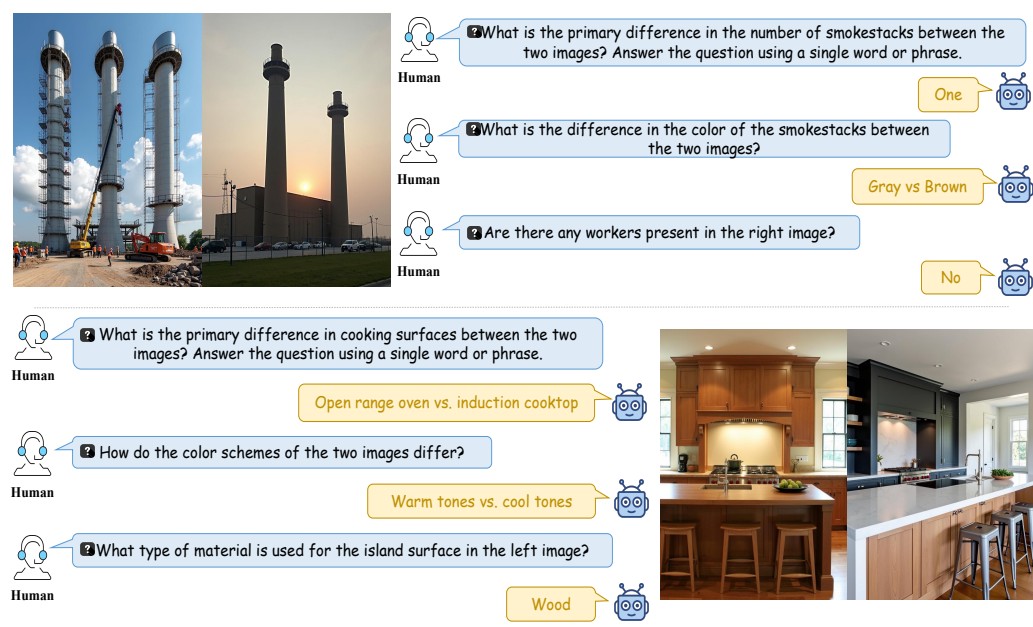

Figure 12: Illustration of our SynthContrastShort

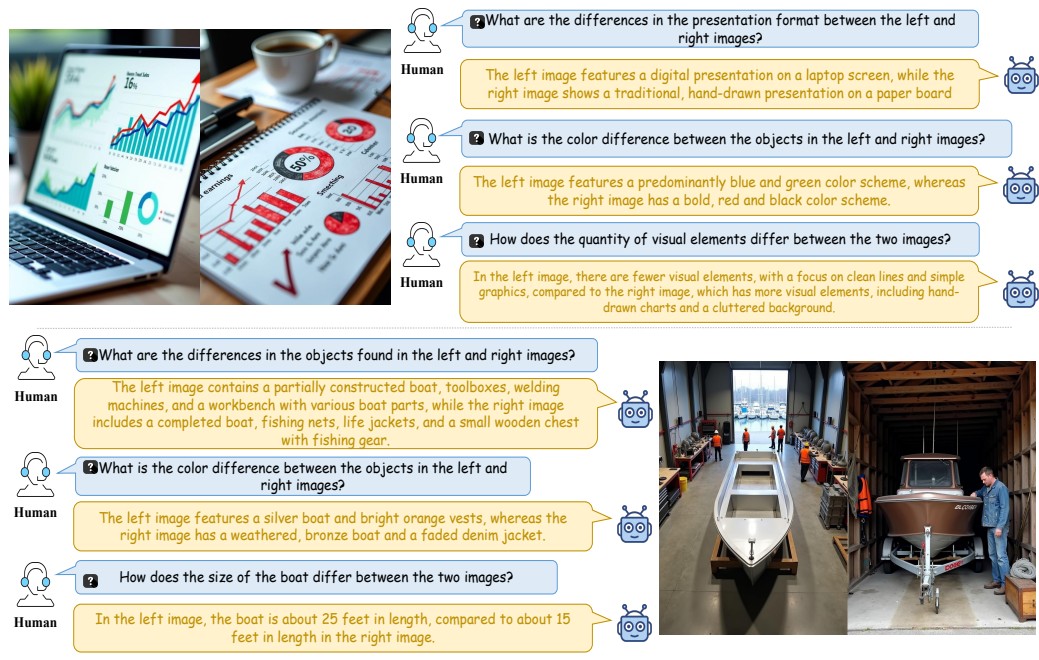

Figure 13: Illustration of our SynthContrastLong

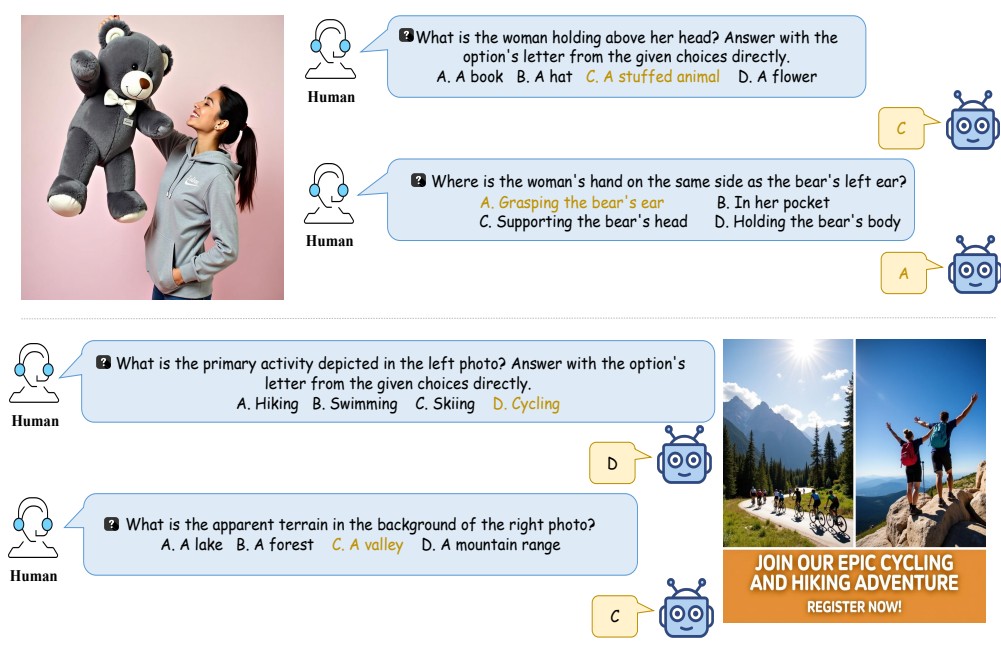

Figure 14: Illustration of our SynthMultiChoice

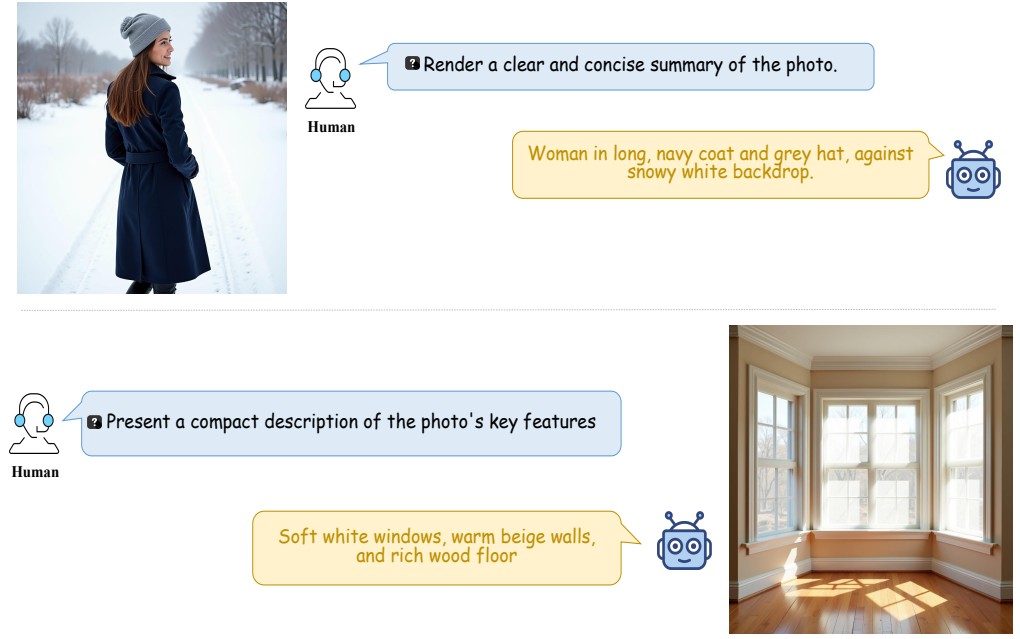

Figure 15: Illustration of our SynthColor

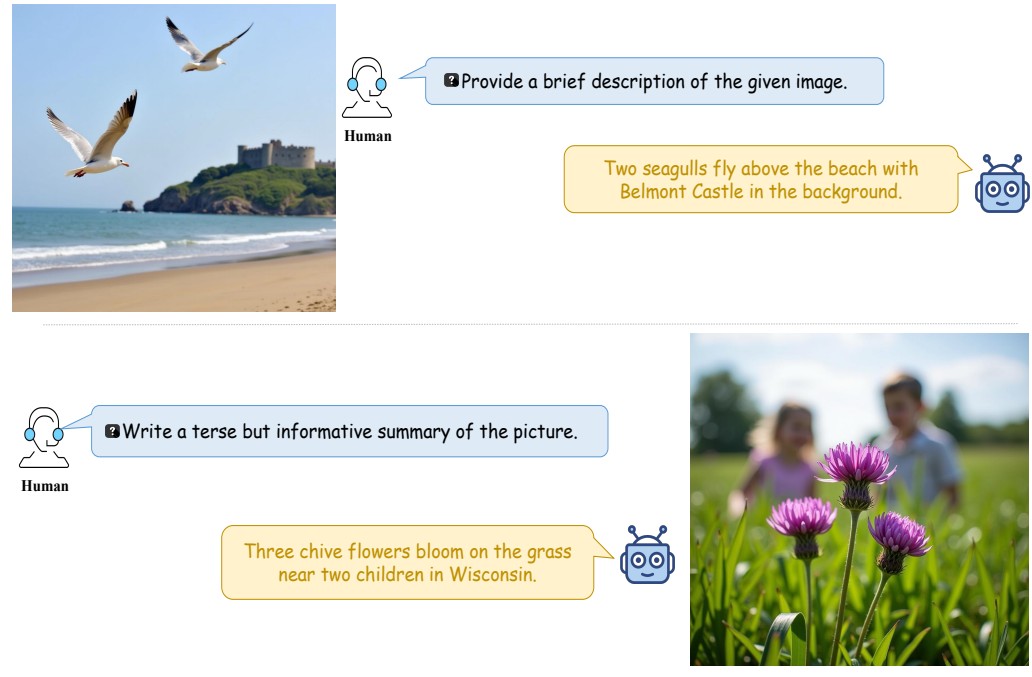

Figure 16: Illustration of our SynthCount

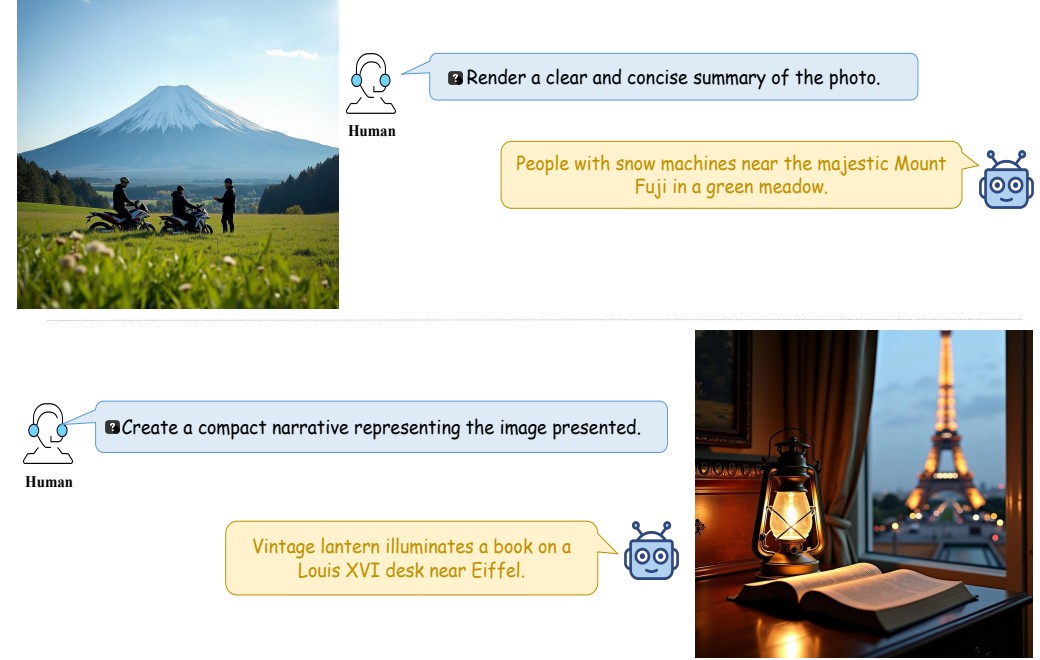

Figure 17: Illustration of our SynthScene

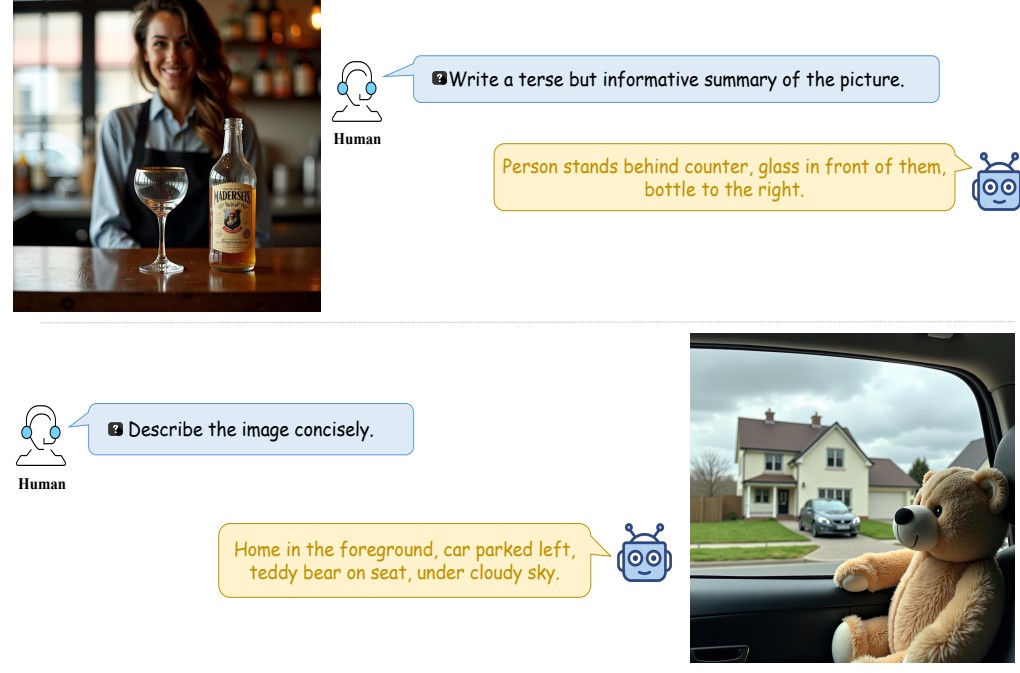

Figure 18: Illustration of our SynthSpatial

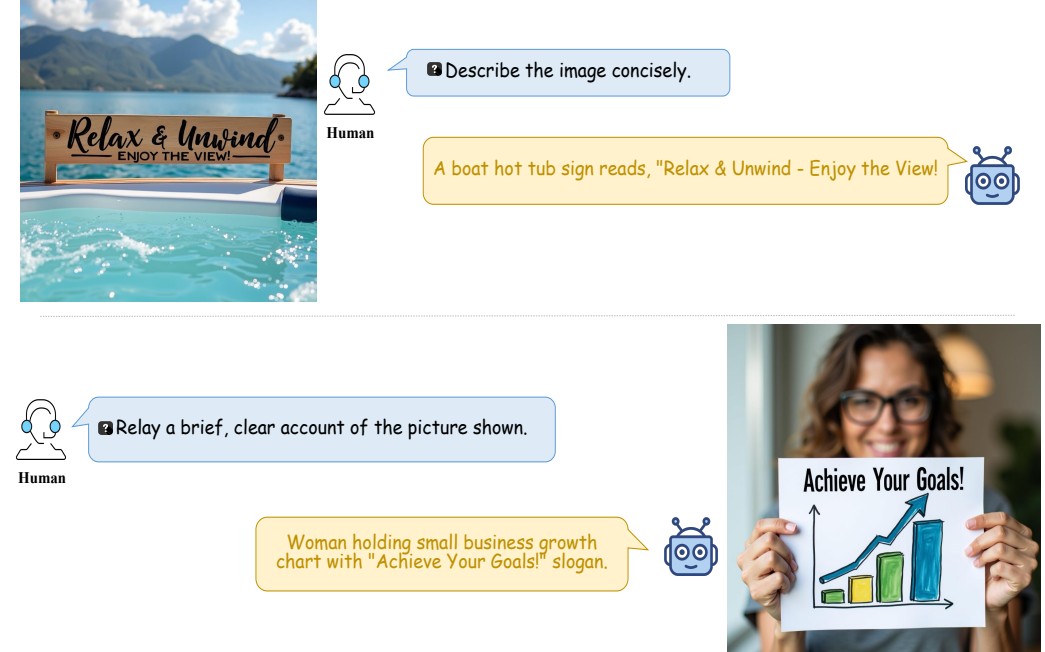

Figure 19: Illustration of our SynthText

