# OpenReview forum: "FLARE: Fully Integration of Vision-Language Representations for Deep Cross-Modal Understanding"
_ICLR.cc/2026/Conference — ICLR 2026 Poster_

### Official Review · Reviewer_1a7W · 2025-10-23

**Soundness:** 3
**Presentation:** 2
**Contribution:** 2
**Rating:** 4
**Confidence:** 4

**Summary:**

This paper introduces FLARE, a new family of vision-language models (VLMs) that aim for a deeper integration of vision and language representations. Unlike previous methods that often treat vision features independently and only fuse them with text during decoding, FLARE proposes a fully vision-language alignment and integration paradigm. With extensive experiments verified its effectiveness.

**Strengths:**

1. The paper is clearly structured and easy to follow, making the proposed methodology and experimental results comprehensible.
2. The authors have conducted extensive experiments across various benchmarks and scales, demonstrating the effectiveness and generalizability of FLARE.

**Weaknesses:**

1. The effectiveness of a simple MLP projecting text tokens into vision space for a vision encoder with significantly fewer parameters than the LLM is unclear. I'm curious about how the vision encoder can effectively process these projected text tokens.
2. Equation 3 appears to have typos. Should it be $(V_i,V_q)$ instead of $V_i(V_q)$?
3. While Qwen2.5VL 7B is presented as a strong baseline, it would strengthen the paper to include experiments based on Qwen2.5 model series to further demonstrate the superiority of the proposed method.

**Questions:**

follow weakness

---

> ### Author Response · Authors · 2025-11-17
>
> $\textbf{Overall Response}$:
>
> Thanks for your insightful feedback! Based on your comments, we have extended our experiments to include the Qwen2.5 model series to further demonstrate the robustness and effectiveness of FLARE. We also clarify your concerns regarding the MLP projection mechanism and the notation in Equation 3. We have made the corresponding changes in our revision, and we promise to include the discussions in the final version if the paper is accepted.
>
> $\textbf{Weakness 1}$:The effectiveness of a simple MLP projecting text tokens into vision space for a vision encoder with significantly fewer parameters than the LLM is unclear. I'm curious about how the vision encoder can effectively process these projected text tokens.
>
> $\textbf{Response}$:
>
> Thank you for the insightful question about the effectiveness of $MLP_{t2v}$ projection. Here are the key factors that enable this simple approach to work effectively:
>
> First, the projection operates directly at the **embedding level** rather than attempting to align entire model architectures. For a text query, we obtain its embedding from the LLM (e.g., 4096-dim for LLaMA[1]) and project it to the vision encoder's space (e.g., 1152-dim for SigLIP[2]) via $MLP_{t2v}$. The key insight is that this is not a mapping between two arbitrary spaces. We leverage powerful pre-trained models, and critically, the SigLIP vision encoder is trained with a **language-aligned objective**. This inherent pre-alignment means its visual feature space already **shares a semantic basis with the LLM's**. Consequently, the task of the $MLP_{t2v}$ is substantially simplified. It is not required to learn the entire vision-language relationship from scratch, but rather to perform a fine-grained calibration that bridges the specific representational gap between the two models.
>
> Second, the projection quality is significantly enhanced by our **Dual-Semantic Mapping Loss**. By reconstructing vision tokens through $MLP_{t2v}$ ($L_{t2v}$ in Equation 9) and enforcing bidirectional consistency, we create a strong supervisory signal that helps $MLP_{t2v}$ discover effective projection patterns. We further evaluated this design by control $L_{v2t}$ and $L_{t2v}$ on FLARE-L 8B with Open-LLaVA-NeXT-1M[3] dataset. As shown in the table below, each loss direction contributes to performance gains—specifically, adding $L_{t2v}$ further improves performance compared to using $L_{v2t}$ alone, validating that the $MLP_{t2v}$ is learning a robust mapping. We have already added this experiment to Section 5.2.
>
> |$L_{v2t}$|$L_{t2v}$|MMB-EN|MMB-CN|POPE|MM-Vet|MME-P|MME-C|Seed-Image|MMStar|TextVQA|ChartQA|AI2D|MMMU|SQA|RWQA|CVBench|MMVP|
> |-|-|-|-|-|-|-|-|-|-|-|-|-|-|-|-|-|-|
> |||74.4|72.3|87.9|43.7|1566.8|367.2|73.7|43.8|64.7|61.1|72.9|42.6|78.5|**61.0**|71.2|69.1
> |✓||75.0|72.6|87.8|**44.0**|1558.1|345.6|74.3|44.6|65.1|61.2|73.6|42.7|79.6|60.5|71.5|69.4
> ||✓|74.8|72.9|**88.1**|43.5|1561.1|**383.6**|74.1|45.0|65.4|61.5|73.2|**43.3**|79.3|60.9|71.1|69.6
> |✓|✓|**75.3**|**73.4**|88.0|43.4|**1583.9**|355.8|**74.6**|**45.5**|**65.6**|**61.7**|**73.7**|42.4|**80.3**|60.8|**71.7**|**69.8**|
>
> Third, the projected tokens function as **high-level semantic anchors** to guide visual attention, rather than requiring a perfect reconstruction of linguistic nuances. We implement this via an intelligent masking strategy (Equations 2-3), where text-to-vision attention is enabled only in the deeper encoder layers. This ensures that textual guidance is introduced precisely when visual features have developed rich representations. Figure 1 and Figure 7 empirically validates this effectiveness: the Text-Guided Vision Encoding successfully directs the encoder to focus precisely on query-relevant regions, demonstrating that the vision encoder is indeed effectively processing and utilizing the projected text tokens.
>
> $\textbf{Weakness 2}$: Equation 3 appears to have typos. Should it be $(V_i^s, V_q^s)$ instead of $V_i^s(V_q^s)$?
>
> $\textbf{Response}$:
>
> Thank you for pointing out this potential confusion in our notation. The notation $(V_i^s, V_q^s)$ is indeed more appropriate. We have corrected this in the revision.
>
> To clarify, we intended this notation to indicate that both $ V_i^s $ and $ V_q^s $ are computed using the same averaging operation over the shallow layers.
>
> For better clarity, we could rewrite Equation 3 as:
>
> $$
> V_i^s = \frac{2}{N} \sum_{k=1}^{N/2} V_i^k, \quad
> V_q^s = \frac{2}{N} \sum_{k=1}^{N/2} V_q^k
> $$
>
> $$
> V_i^d = \frac{2}{N} \sum_{k=N/2+1}^{N} V_i^k, \quad
> V_q^d = \frac{2}{N} \sum_{k=N/2+1}^{N} V_q^k
> $$
>
> This makes it explicit that we're computing separate averages for both the visual embeddings $V_i$ and the projected text embeddings $V_q$ over the shallow (first-half) and deep (second-half) layers respectively.

---

> ### Author Response · Authors · 2025-11-17
>
> $\textbf{Weakness 3}$: While Qwen2.5VL 7B is presented as a strong baseline, it would strengthen the paper to include experiments based on Qwen2.5 model series to further demonstrate the superiority of the proposed method.
>
> $\textbf{Response}$:
>
> Thank you very much for your valuable suggestion. We would like to clarify that we use LLaMA as the backbone to ensure a fair comparison with other open-source models such as Cambrian-1[4], Florence-VL[5], and Eagle[6]. Following your advice, we have also extended our experiments to include the Qwen2.5[7] model series in order to further validate the effectiveness and generalizability of our proposed methods. Specifically, we trained FLARE-L and FLARE-X based on Qwen2.5 7B backbones. The detailed results are presented in the table below.
>
> |Models|Backbone LLM|MMB-EN|MMB-CN|POPE|MM-Vet|MME-P|MME-C|Seed-Image|MMStar|TextVQA|OCRBench|ChartQA|AI2D|MathVista|MMMU|SQA|RWQA|CVBench|MMVP|
> |-|-|-|-|-|-|-|-|-|-|-|-|-|-|-|-|-|-|-|-|
> |LLaVA-OneVision[8]|Qwen2.5 7B|81.7|78.0|87.2|58.8|1626.0|483.0|74.8|60.9|78.5|69.7|78.8|81.6|56.1|47.7|96.6|65.5|-|-|
> |Qwen2.5VL 7B[9]|Qwen2.5 7B|83.2|**82.8**|85.9|**69.7**|1652.7|**633.4**|77.0|**64.1**|**83.5**|**88.5**|**89.5**|83.4|**68.1**|**58.0**|89.0|68.4|81.1|**80.9**|
> |FLARE-L|LLaMA3.1 8B|81.8|76.2|88.7|59.6|1596.8|347.3|77.6|54.5|76.2|66.3|77.9|81.6|57.5|43.1|89.7|65.4|79.5|78.6|
> |FLARE-L|Qwen2.5 7B|82.4|77.5|88.5|58.7|1612.4|366.8|77.9|55.8|75.5|66.2|78.7|83.1|60.1|46.5|88.3|65.2|79.9|79.4|
> |FLARE-X|LLaMA3.1 8B|83.6|78.3|**89.1**|62.8|1639.1|378.9|78.7|56.4|79.7|69.9|83.4|83.6|61.1|45.6|**91.2**|66.9|81.5|79.7|
> |FLARE-X|Qwen2.5 7B|**84.4**|80.1|89.0|64.2|**1655.1**|365.2|**79.3**|58.6|80.2|70.4|85.1|**85.5**|63.2|47.8|89.7|**68.4**|**82.4**|80.1|
>
> As shown in the results, our methods achieve consistently better performance across the Qwen2.5 model series.
>
> To enable a fairer comparison with the Qwen2.5-VL model, We replaced SigLip with NaViT[10] and used the Qwen2.5 3B and Qwen2.5 7B models as LLM backbones, mirroring the setup of Qwen2.5-VL. The experimental results are shown below.
>
> |Models|MMB-EN|MMB-CN|POPE|MM-Vet|MME-P|MME-C|Seed-Image|MMStar|TextVQA|OCRBench|ChartQA|AI2D|MathVista|MMMU|SQA|RWQA|CVBench|MMVP|
> |-|-|-|-|-|-|-|-|-|-|-|-|-|-|-|-|-|-|-|
> |Qwen2.5VL 3B|79.1|78.1|87.3|61.4|1592.4|607.5|74.0|55.9|79.3|79.7|84.0|81.4|62.3|51.2|81.4|65.4|75.5|76.7|
> |FLARE 3B|83.2|77.7|89.0|63.8|1637.1|473.1|77.1|56.1|80.8|73.2|82.2|82.9|59.9|50.8|88.7|64.5|81.2|80.4|
> |Qwen2.5VL 7B|83.2|82.8|85.9|**69.7**|1652.7|**633.4**|77.0|**64.1**|83.5|**88.5**|**89.5**|83.4|**68.1**|**58.0**|89.0|68.4|81.1|80.9|
> |FLARE 7B|**86.0**|**83.3**|**89.8**|68.2|**1665.4**|508.5|**79.7**|62.4|**85.2**|78.4|88.3|**84.2**|65.9|54.4|**92.2**|**71.1**|**82.8**|**83.3**|
>
> FLARE excels on the majority of general visual understanding benchmarks, despite using only a fraction of its training data. FLARE 3B even surpasses the larger Qwen2.5VL 7B on benchmarks like MMBench, POPE and CVBench. This experiment highlights the superiority of the FLARE architecture. We have added this experiment to Section 5.1 in our revision.
>
>
> $\textbf{References}$:
>
> [1] Tschannen, Michael, et al. "Siglip 2: Multilingual vision-language encoders with improved semantic understanding, localization, and dense features." arXiv preprint arXiv:2502.14786 (2025).
>
> [2] https://github.com/meta-llama/llama3/blob/main/MODEL_CARD.md.
>
> [3] https://github.com/xiaoachen98/Open-LLaVA-NeXT
>
> [4] Tong, Peter, et al. "Cambrian-1: A fully open, vision-centric exploration of multimodal llms." Advances in Neural Information Processing Systems 37 (2024): 87310-87356.
>
> [5] Chen, Jiuhai, et al. "Florence-vl: Enhancing vision-language models with generative vision encoder and depth-breadth fusion." Proceedings of the Computer Vision and Pattern Recognition Conference. 2025.
>
> [6] Shi, Min, et al. "Eagle: Exploring the design space for multimodal llms with mixture of encoders." arXiv preprint arXiv:2408.15998 (2024).
>
> [7] Yang, An, et al. "Qwen2.5 technical report." arXiv preprint arXiv:2505.09388 (2025).
>
> [8] Li, Bo, et al. "Llava-onevision: Easy visual task transfer." arXiv preprint arXiv:2408.03326 (2024).
>
> [9] Bai, Shuai, et al. "Qwen2. 5-vl technical report." arXiv preprint arXiv:2502.13923 (2025).
>
> [10] Dehghani, Mostafa, et al. "Patch n’pack: Navit, a vision transformer for any aspect ratio and resolution." Advances in Neural Information Processing Systems 36 (2023): 2252-2274.

---

> > ### Author Response · Authors · 2025-11-26
> > **Looking forward to your reply**
> >
> > Dear reviewer 1a7W,
> >
> > Greetings from the authors!
> >
> > We would like to express our sincere gratitude for your insightful comments.
> >
> > Regarding the MLP projection mechanism, Equation 3 notation, and Qwen2.5 experiments, we have provided detailed clarifications and additional results in the revised paper. We will include them in the final version.
> >
> > If you have any question for our paper, please feel free to point out and we will try to address it quickly. Thanks for your time and looking forward to your reply!
> >
> > Best wishes!
> >
> > Authors

---

> ### Comment · Reviewer_1a7W · 2025-11-26
>
> Many thanks to your response. Most of my concerns have been addressed. I have raised my score.

---

> > ### Author Response · Authors · 2025-11-26
> >
> > We truly appreciate your decision to raise the score and your engagement throughout the review process. Your feedback has been instrumental in strengthening our paper. We will ensure that all the discussed revisions are included in the final version.

---

### Official Review · Reviewer_NAvg · 2025-10-26

**Soundness:** 3
**Presentation:** 2
**Contribution:** 2
**Rating:** 4
**Confidence:** 5

**Summary:**

This paper introduces FLARE, vision-language models with several designs to enable cross-modal alignment and interaction across in visual encoders and LMM backbones. Besides architectural innovation, a data synthesis strategy is proposed to construct text-centric data. Experimental results validate the effectiveness of key components in FLARE.

**Strengths:**

* This paper is strongly motivated, compared with prior work’s preliminary explorations, this paper conducts a comprehensive exploration of how to enhance text-guided visual encoding.
* The proposed Context-Aware Alignment Decoding module is interesting, providing a novel solution to refine in-context visual features layer by layer.
* Experimental results support the effectiveness of FLARE, the authors also provide analysis on additional computational cost, which is important to analyze the introduced modules.

**Weaknesses:**

* Presentation of this paper can be improved:
  1. Horizontal lines and citations are missing in Table 1
  2. In Equation 3, would it be $(V_i^s, V_q^s)$ rather than $V_i^s(V_q^s)$?
  3. Lack of explanation for $r$ and $c$ in Equation (6, 7)

* Missing important details:
  * How to collect the curated caption pool in Figure 4
  * The number of latent tokens is randomly sampled during training, how about evaluation?
  * Except for Context-Aware Alignment Decoding, the novelty of Text-Guided Vision Encoding and Text-Driven data synthesis is limited:
    * Text-Guided Vision Encoding is quite similar to Q-Former in InstructBLIP.
    * Utilizing synthesized images for VLM training is not a novel solution, for example, SynthVLM [1].
* There are several important points to address for the method:
  * From my perspective, data filtering is important for the synthesized data, which is not included in this paper
  * Most VLMs adopt different strategies to control trainable parameters in different stages, which is not considered in FLARE
  * How could the designed module be applied to more complex scenarios like videos, multi-images, and excessively long text contexts?
  * Fore FLARE-X, why not apply the dynamic-resolution strategy that is utilized in NaViT and Qwen2.5-VL, which seems more suitable for your method compared to the subfigure-based solution.
* Conducted experiments are limited, in terms of:
  * Lack of analysis on the quality of synthesized images
  * Experiments in Table 3 do not sound fair to me, the superior performance of FLARE may come from the SigLIP2, which is better than vision encoders used by other models.
  * Current analysis in the main paper focuses on quantitative performance on benchmarks, detailed analysis is expected to reveal how the proposed modules help vision-language interaction.


[1] SynthVLM: Towards High-Quality and Efficient Synthesis of Image-Caption Datasets for Vision-Language Models (https://arxiv.org/abs/2407.20756)

**Questions:**

see weaknesses

---

> ### Author Response · Authors · 2025-11-17
>
> $\textbf{Overall Response}$:
>
> Thanks for your insightful feedback! We have extended our experiments with NaViT and Qwen2.5 and provided fairer comparisons using SigLIP. We have also clarified key details regarding our notation, caption curation, and the novelty. Due to page limitations, many important details were placed in the appendix, and we take this opportunity to clarify them. We have made the corresponding changes in our revision, and we promise to include the discussions in the final version if the paper is accepted.
>
> $\textbf{Weakness 1-1}$: Horizontal lines and citations are missing in Table 1
>
> $\textbf{Response}$:
>
> We apologize for the formatting oversight. In our initial submission, due to space constraints, we provided detailed model descriptions in Appendix. In our revision, we have moved the model descriptions and citations to Section 4, Comparison Models. We have also added proper horizontal lines to Table 1 to improve readability.
>
> $\textbf{Weakness 1-2}$: In Equation 3, would it be $(V_i^s, V_q^s)$ rather than $V_i^s(V_q^s)$?
>
> $\textbf{Response}$:
>
> Thank you for pointing out this confusion. The notation $(V_i^s, V_q^s)$ is indeed more appropriate. We have corrected this in the revision.
>
> $\textbf{Weakness 1-3}$: Lack of explanation for $r$ and $c$ in Equation 6-7
>
> $\textbf{Response}$:
>
> Let $T_L ∈ R^{l×l×D}$ denote the latent tokens with spatial resolution $l$. $r, c ∈ {0, 1, ..., l-1}$ are the row and column indices. Each latent token $T_L[r,c]$ corresponds to a spatial position, and Equations 6-7 are applied to each latent token independently. In our revision, we have added the definitions for $r$ and $c$ in Equation 6.
>
> $\textbf{Weakness 2-1}$: How to collect the curated caption pool in Figure 4
>
> $\textbf{Response}$:
>
> We apologize for not being clear. Due to space constraints, we provide detailed caption curation in Appendix E.2.1. Here we summarize the key steps:
>
> **Caption Sources**: We sourced captions from the DataComp[1], which contains 12B image-caption pairs. It provides diverse textual descriptions across various domains.
>
> **Filtering Process**: First, we applied rule-based filtering using Data-Juicer[2] to remove low-quality captions containing watermarks, repetitions, or high perplexity values. Second, we employed LLaMA3.1-70B to assess semantic correctness and syntactic richness of the captions.
>
> **Final Pool Composition**: We selected the top-scoring 20M high-quality captions. These captions ensure comprehensive coverage of different visual and semantic aspects, enabling our model to generate diverse QA pairs and corresponding images.
>
> $\textbf{Weakness 2-2}$: The number of latent tokens is randomly sampled during training, how about evaluation?
>
> $\textbf{Response}$:
>
> During both training and evaluation, we set random seeds to 42 to ensure stability and reproducibility. In our revision, we have added the random seed setting in Section 4, Implementation Details.
>
> $\textbf{Weakness 2-3-1}$: Text-Guided Vision Encoding is quite similar to Q-Former in InstructBLIP[3].
>
> $\textbf{Response}$:
>
> We thank the reviewer for their insightful comment and the opportunity to clarify the key distinctions between our Text-Guided Vision Encoding and the Q-Former in InstructBLIP.
>
> First, our approach redefines the interaction as **feature generation**, whereas the Instruct Q-Former functions as a **feature selector**. By integrating text directly into vision encoder, our method shapes visual feature generation from the **ground up**, processing both modalities jointly. In contrast, Instruct Q-Former is an external module that selects and distills information from a static, pre-computed feature map, rather than influencing how those features are created.
>
> Second, this deep integration enables a process of **mutual refinement**, unlike Instruct Q-Former's post-hoc extraction. Our joint processing fosters a **continuous, bidirectional** dialogue where visual features are shaped by text, and textual representations are simultaneously grounded by visual context. This fundamentally differs from Instruct Q-Former's unidirectional query mechanism, which operates on an already-fixed visual output without this synergistic refinement.
>
> Third, Text-Guided Vision Encoding is part of a self-supervising system with **Dual-Semantic Mapping Loss**. This loss creates a **closed-loop verification system**, ensuring that the text-guided visual features maintain meaningful semantic consistency across both modalities. This is a critical distinction from Instruct Q-Former, which learns through more abstract queries and lacks an explicit mechanism to enforce semantic alignment.
>
> Finally, our method is **more efficient**. Instruct Q-Former introduces significant computational overhead through additional self-attention blocks that fuse queries with text. Our approach, by directly leveraging the vision encoder's existing capacity, adds no new parameters and only a 2% computational increase as shown in Table 6.

---

> > ### Author Response · Authors · 2025-11-17
> >
> > $\textbf{Weakness 2-3-2}$: The novelty of Text-Driven data synthesis is limited. Utilizing synthesized images is not a novel solution.
> >
> > $\textbf{Response}$:
> >
> > We thank the reviewer for their comment. The novelty of Text-Driven VQA Synthesis lies in **paradigm, methodology, and synergistic co-design with our architecture**.
> >
> > First, our work advances the **objective and complexity**. While the focus in SynthVLM[4] remains on the image, our approach is **text-centric**, prioritizing the generation of rich textual and QA content. Our pipeline is designed to produce data for complex, cognitive tasks, where the synthesized image serves as an accessory. This is achieved through our multi-step pipeline where high-quality captions are first enriched by an LLM, which then serve as a unified foundation for generating both diverse QA pairs and the corresponding images.
> >
> > Second, our core innovation is a **data-architecture co-design**. The data we synthesize is not generic; it is specifically engineered to activate and train the novel components of our architecture. For example, our multi-choice and text-rich QA pairs provide the varied textual cues necessary for our Text-Guided Vision Encoding to learn query-aware representations and for the Dual-Semantic Mapping Loss to enforce meaningful bidirectional consistency.
> >
> > The effectiveness of this co-design is validated by FLARE's strong performance on text-rich benchmarks like MMBench[5], CVBench[6], and MMVP[7](Tables 1, 8), demonstrating that our tailored synthetic data is instrumental in achieving superior modality fusion.
> >
> > $\textbf{Weakness 3-1}$: Data filtering is not included in this paper.
> >
> > $\textbf{Response}$:
> >
> > We appreciate the reviewer’s concern about data quality. Due to space limitations, we placed our selection process in Appendix E. Here, we describe the 4-stage filtering pipeline:
> > 1. Caption Filtering (Appendix E.2.1): We filtered 20M high-quality captions from DataComp using both rule-based methods and LLM-based semantic scoring.
> > 2. Description Filtering (Appendix E.2.2): We further refined 12M descriptions using similar quality metrics.
> > 3. Image Generation Filtering (Appendix E.2.3): We employed a dual-metric approach combining CLIPScore for text-image alignment and SSIM for visual quality, retaining 6M high-quality pairs.
> > 4. QA Pair Filtering (Appendix E.2.4): We verified answer correctness using LLaVA-OneVision 72B[8] and filtered by CLIPScore, resulting in 4M high-quality QA pairs.
> >
> > The t-SNE visualizations in Figures 8 and 9 show that our data's embeddings are diverse and integrate well with real-world datasets. Furthermore, Table 14 confirms our filtering success, as our synthesized data achieves higher CLIPScores than datasets like LLaVA665k[9].
> >
> > $\textbf{Weakness 3-2}$: Most VLMs adopt different strategies to control trainable parameters in different stages, which is not considered in FLARE.
> >
> > $\textbf{Response}$:
> >
> > This was a deliberate design choice, and our ablation studies in Table 12 (Appendix C.4) confirm that full-parameter training outperforms freezing strategies for our architecture.
> >
> > Many leading models[6,8,10] also employ full-parameter fine-tuning. Our key distinction is extending this strategy to the pre-training stage (Stage 1).
> >
> > The reason lies in our architectural modifications. In Stage 1, our Text-Guided Vision Encoding re-engineers the vision encoder to be natively cross-modal. It must learn to jointly process visual and textual embeddings. As shown in Table 12, this full unfreezing is essential for the encoder to learn this new, integrated paradigm.
> >
> >
> > $\textbf{Weakness 3-3}$: How could the designed module be applied to scenarios like videos, multi-images, and excessively long text contexts?
> >
> > $\textbf{Response}$:
> >
> > We appreciate the reviewer's insightful question. The modular nature of our approach makes it well-suited for these extensions.
> >
> > For videos and multi-images, our design can treat frames or images as a sequence. Text-Guided Vision Encoding can be applied **per-frame**, and the resulting features can be stacked. Context-Aware Alignment Decoding is effective here, as its windowed attention naturally extends to 3D feature stack, operating much like a **convolution**. For example, with video frames $F_1$, $F_2$, ..., $F_n$, after encoding we obtain features $V_1$, $V_2$, ..., $V_n$. These can be stacked as $V ∈ R^{(n×h×w×d)}$, where our semantic exchange layers can perform attention across the temporal dimension $n$ within each spatial window $(h/l × w/l)$.
> >
> > For long text queries, our architecture is equipped to manage complexity. The latent tokens within our Context-Aware Alignment Decoding act as a powerful **contextual summary**, compressing the long textual context to extract the most salient information and guide the model's visual focus. Additionally, techniques like 1D interpolation can adapt the initial long query representation before it is projected into the vision space.

---

> > > ### Author Response · Authors · 2025-11-17
> > >
> > > $\textbf{Weakness 3-4}$: For FLARE-X, why not apply the dynamic-resolution strategy that is utilized in NaViT and Qwen2.5-VL.
> > >
> > > $\textbf{Response}$:
> > >
> > > We thank the reviewer for this insightful suggestion. We chose the dynamic resolution to ensure **fair comparisons** with models like LLaVA-NeXT[10] and LLaVA-OneVision. Additionally, the dynamic resolution approach demonstrates our method's potential applicability to videos and multi-image, as discussed in Weakness 3-3.
> > >
> > > Actually, our method can naturally extend to Qwen2.5-VL's NaViT[11]. We conducted experiments using the same encoder as Qwen2.5-VL, with Qwen2.5 3B and Qwen2.5 7B as backbone. We maintained identical data and training strategies as FLARE-X and FLARE-L. The results are shown below.
> > >
> > > |Models|MMB-EN|MMB-CN|POPE|MM-Vet|MME-P|MME-C|Seed-Image|MMStar|TextVQA|OCRBench|ChartQA|AI2D|MathVista|MMMU|SQA|RWQA|CVBench|MMVP|
> > > |-|-|-|-|-|-|-|-|-|-|-|-|-|-|-|-|-|-|-|
> > > |Qwen2.5VL 3B|79.1|78.1|87.3|61.4|1592.4|607.5|74.0|55.9|79.3|79.7|84.0|81.4|62.3|51.2|81.4|65.4|75.5|76.7|
> > > |FLARE 3B|83.2|77.7|89.0|63.8|1637.1|473.1|77.1|56.1|80.8|73.2|82.2|82.9|59.9|50.8|88.7|64.5|81.2|80.4|
> > > |Qwen2.5VL 7B|83.2|82.8|85.9|**69.7**|1652.7|**633.4**|77.0|**64.1**|83.5|**88.5**|**89.5**|83.4|**68.1**|**58.0**|89.0|68.4|81.1|80.9|
> > > |FLARE 7B|**86.0**|**83.3**|**89.8**|68.2|**1665.4**|508.5|**79.7**|62.4|**85.2**|78.4|88.3|**84.2**|65.9|54.4|**92.2**|**71.1**|**82.8**|**83.3**|
> > >
> > > FLARE excels on the majority of general visual understanding benchmarks, despite using only a fraction of its training data. FLARE 3B even surpasses the larger Qwen2.5VL 7B on benchmarks like MMBench, POPE and CVBench. This experiment highlights the superiority of the FLARE architecture.
> > >
> > > We have added this experiment to Section 5.1 in our revision.
> > >
> > > $\textbf{Weakness 4-1}$: Lack of analysis on the quality of synthesized images.
> > >
> > > $\textbf{Response}$:
> > >
> > > We thank the reviewer for highlighting this important aspect. Due to space constraints, we provided comprehensive analysis of synthesized data quality in Appendix E.3 and Appendix E.4.
> > >
> > > First, we analyze distribution characteristics through t-SNE visualization in Figures 8 and 9. The visualizations show our synthetic datasets cluster closely with real counterparts, with dialogue-style datasets aligning with LLaVA-665k, demonstrating successful emulation of real data distributions.
> > >
> > > Second, we provide quantitative evaluation in Table 14. Our synthetic datasets achieve CLIPScores of 0.341-0.382, exceeding LLaVA665k (0.332). Human evaluation shows that while image quality is lower (4.6-5.7 vs 7.1-7.8 for real), text quality (6.3-8.0) and image-text alignment (6.9-7.6) are comparable to real datasets. GPT-4o evaluation confirms similar patterns.
> > >
> > > $\textbf{Weakness 4-2}$: Experiments in Section 5, Fair Comparsion with Existing Models,  do not sound fair, the superior performance of FLARE may come from the SigLIP2.
> > >
> > > $\textbf{Response}$:
> > >
> > > We appreciate the reviewer's concern about the potential influence of the vision encoder. To address this, we conducted additional ablation experiments on Cambrian-1 7M[6] dataset using SigLip-SO400M-Patch14-384[12] to isolate the impact of SigLip2-Giant-Opt-Patch16-384[13].
> > >
> > > |Models|Vision Encoder|Data|MMB-EN|MMB-CN|POPE|MM-Vet|MME-P|MME-C|Seed-Image|MMStar|TextVQA|ChartQA|AI2D|MMMU|SQA|RWQA|CVBench|MMVP|
> > > |-|-|-|-|-|-|-|-|-|-|-|-|-|-|-|-|-|-|-|
> > > |Florence-VL 8B|Florence-2[14]|7M|74.8|72.1|88.5|46.7|1554.2|363.4|73.4|49.7|71.0|71.6|72.5|43.4|84.8|63.6|71.8|69.1|
> > > |Cambrian-1 8B|CLIP[15] SigLIP ConvNeXt[16] DINOv2[17]|7M|75.6|72.4|87.9|46.4|1562.1|389.5|74.4|52.2|71.8|72.2|74.3|**45.7**|83.3|64.7|74.2|70.6|
> > > |Eagle 8B|CLIP ConvNeXt EVA[18] Pix2Struct[19] SAM[20]|7M|76.6|72.3|87.8|48.6|1566.4|334.9|74.6|**55.7**|75.4|77.2|75.2|44.8|82.7|66.2|74.1|72.4|
> > > |LLaVA-NeXT 8B|SigLIP|7M|76.4|73.2|88.3|50.4|1549.2|394.5|75.1|54.8|75.8|76.6|76.1|44.5|83.4|**66.7**|75.2|72.8|
> > > |FLARE-L 8B|SigLIP|7M|77.8|72.1|**89.0**|46.5|1569.6|398.4|74.7|55.2|74.4|74.2|74.8|42.4|83.8|65.7|73.2|72.9|
> > > |FLARE-L 8B|SigLIP2|7M|77.1|73.9|88.6|48.1|1577.9|**409.2**|75.4|54.3|73.5|73.1|75.7|43.9|84.6|65.2|75.6|72.2|
> > > |FLARE-X 8B|SigLIP|7M|78.4|74.0|88.8|51.2|1556.3|370.8|75.7|54.5|**78.4**|**79.6**|77.9|44.4|85.1|66.5|76.0|74.2|
> > > |FLARE-X 8B|SigLIP2|7M|**79.3**|**74.6**|88.6|**52.7**|**1586.2**|388.5|**76.0**|55.3|77.1|78.7|**78.6**|44.1|**86.3**|66.2|**77.2**|**74.5**|
> > >
> > >
> > > As shown above, when using SigLIP, FLARE-L and FLARE-X remain highly competitive, outperforming other leading models. This confirms that the primary driver of our model's success is its architectural design, not solely the choice of vision encoder. Actually, SigLIP's smaller patch size yields superior performance on detail-intensive OCR tasks.
> > >
> > > In our revision, we have updated the experiments results in Section 5.3, Fair Comparison with Existing Models, to use the SigLIP results and have further detailed the experimental configurations to ensure a fair comparison.

---

> > > > ### Author Response · Authors · 2025-11-17
> > > >
> > > > $\textbf{Weakness 4-3}$: Current analysis focuses on quantitative performance, detailed analysis is expected to reveal how the proposed modules help vision-language interaction.
> > > >
> > > > $\textbf{Response}$:
> > > >
> > > > We thank the reviewer for requesting deeper analysis of our modules' contributions to vision-language interaction. Figure 1 provides a qualitative analysis that demonstrates how our modules enhance vision-language interaction at every stage.
> > > >
> > > > **At the Pixel Level(Encoding)**, LLaVA and LLaVA-NeXT process the image in isolation, resulting in unfocused, query-agnostic visual features. In contrast, our Text-Guided Vision Encoding enables FLARE to achieve early fusion, precisely focusing its attention on query-relevant regions (e.g., the "flower") during the encoding stage itself.
> > > >
> > > > **At the Modality Level(Alignment)**, LLaVA and LLaVA-NeXT, lacking our Dual-Semantic Mapping Loss, fail to achieve proper feature alignment after the projector stage, showing near-zero cosine similarity. FLARE, however, demonstrates strong semantic alignment, creating a coherent, shared representation space.
> > > >
> > > > **At the Query Level(Decoding)**, LLaVA and LLaVA-NeXT's misalignment cascades into poor attention. Both exhibit weak attention patterns. Conversely, FLARE demonstrates sharp, accurate attention, effectively linking the textual query to the precise visual evidence required for the answer.
> > > >
> > > > We also provide an additional example in Figure 7(Appendix D) in our revision, showing how modules help vision-language interaction in even harder problems and complex scenarios.
> > > >
> > > > Quantitatively, our results in Table 3 show how each module improves performance. Additionally, we measured semantic similarity between visual and textual tokens below.FLARE achieves **5× higher** semantic alignment, validating our fusion effectiveness.
> > > >
> > > > |Model|Average Cosine Similarity|
> > > > |-|-|
> > > > |FLARE-L|0.35|
> > > > |LLaVA|0.06|
> > > > |LLaVA-NeXT|0.04|
> > > >
> > > > $\textbf{References}$:
> > > >
> > > > [1] Gadre, Samir Yitzhak, et al. "Datacomp: In search of the next generation of multimodal datasets." Advances in Neural Information Processing Systems 36 (2023): 27092-27112.
> > > >
> > > > [2] https://github.com/datajuicer/data-juicer.
> > > >
> > > > [3] Dai, Wenliang, et al. "Instructblip: Towards general-purpose vision-language models with instruction tuning." Advances in neural information processing systems 36 (2023): 49250-49267.
> > > >
> > > > [4] Liu, Zheng, et al. "Synthvlm: High-efficiency and high-quality synthetic data for vision language models." arXiv preprint arXiv:2407.20756 (2024).
> > > >
> > > > [5] Liu, Yuan, et al. "Mmbench: Is your multi-modal model an all-around player?." European conference on computer vision. Cham: Springer Nature Switzerland, 2024.
> > > >
> > > > [6] Tong, Peter, et al. "Cambrian-1: A fully open, vision-centric exploration of multimodal llms." Advances in Neural Information Processing Systems 37 (2024): 87310-87356.
> > > >
> > > > [7] Tong, Shengbang, et al. "Eyes wide shut? exploring the visual shortcomings of multimodal llms." Proceedings of the IEEE/CVF Conference on Computer Vision and Pattern Recognition. 2024.
> > > >
> > > > [8] Li, Bo, et al. "Llava-onevision: Easy visual task transfer." arXiv preprint arXiv:2408.03326 (2024).
> > > >
> > > > [9] Liu, Haotian, et al. "Visual instruction tuning." Advances in neural information processing systems 36 (2023): 34892-34916.
> > > >
> > > > [10] https://github.com/LLaVA-VL/LLaVA-NeXT
> > > >
> > > > [11] Dehghani, Mostafa, et al. "Patch n’pack: Navit, a vision transformer for any aspect ratio and resolution." Advances in Neural Information Processing Systems 36 (2023): 2252-2274.
> > > >
> > > > [12] Zhai, Xiaohua, et al. "Sigmoid loss for language image pre-training." Proceedings of the IEEE/CVF international conference on computer vision. 2023.
> > > >
> > > > [13] Tschannen, Michael, et al. "Siglip 2: Multilingual vision-language encoders with improved semantic understanding, localization, and dense features." arXiv preprint arXiv:2502.14786 (2025).
> > > >
> > > > [14] Xiao, Bin, et al. "Florence-2: Advancing a unified representation for a variety of vision tasks." Proceedings of the IEEE/CVF Conference on Computer Vision and Pattern Recognition. 2024.
> > > >
> > > > [15] Radford, Alec, et al. "Learning transferable visual models from natural language supervision." International conference on machine learning. PmLR, 2021.
> > > >
> > > > [16] Cherti, Mehdi, et al. "Reproducible scaling laws for contrastive language-image learning." Proceedings of the IEEE/CVF conference on computer vision and pattern recognition. 2023.
> > > >
> > > > [17] Oquab, Maxime, et al. "Dinov2: Learning robust visual features without supervision." arXiv preprint arXiv:2304.07193 (2023).
> > > >
> > > > [18] Sun, Quan, et al. "Eva-clip: Improved training techniques for clip at scale." arXiv preprint arXiv:2303.15389 (2023).
> > > >
> > > > [19] Lee, Kenton, et al. "Pix2struct: Screenshot parsing as pretraining for visual language understanding." International Conference on Machine Learning. PMLR, 2023.
> > > >
> > > > [20] Kirillov, Alexander, et al. "Segment anything." Proceedings of the IEEE/CVF international conference on computer vision. 2023.

---

> ### Comment · Reviewer_NAvg · 2025-11-25
> **Response to the rebuttal**
>
> To Authors,
>
> Most of my concerns about method details, data quality and fair experiment settings are resolved. I have raised my score.
>
> Reviewer

---

> > ### Author Response · Authors · 2025-11-26
> >
> > We sincerely thank you for raising your score and for your insightful comments. Your constructive suggestions have been valuable in improving the quality of our paper. We promise to incorporate the discussed points and additional analysis into the final revision.

---

### Official Review · Reviewer_usqb · 2025-10-27

[review text omitted: it was posted to a different submission]

---

> ### Author Response · Authors · 2025-11-16
>
> We sincerely thank you for your time and for providing detailed feedback. However, after carefully reading your comments, we must respectfully point out that the review appears to be for a different paper.
>
> First, there is a fundamental disconnect in the research topic. Our submission introduces FLARE, a family of Vision-Language Models designed for deep cross-modal understanding. Your review, however, evaluates a paper concerning a systems-level integration of hardware accelerators, compilers, and runtime schedulers. These are topics our paper does not address.
>
> Second, the contributions you assessed do not match those presented in our work. Our key contributions include Text-Guided Vision Encoding, Dual-Semantic Mapping Loss, Context-Aware Alignment Decoding, and a new data synthesis method for VLMs. Your review discusses strengths and weaknesses related to modular integration APIs and cross-layer optimizations, which are not part of our submission.
>
> Finally, your questions regarding our evaluation further highlight this discrepancy. You ask about baseline toolchains, vendor stacks, and compile-time overheads. Our paper evaluates performance against other Vision-Language Models on standard VLM benchmarks and does not involve hardware compilers or runtime schedulers.
>
> We believe this is likely an unfortunate coincidence where our model's acronym, FLARE, is the same as that of a systems-level paper. Because the feedback does not pertain to our work, it unfortunately offers no guidance for us to improve our manuscript. We wanted to bring this to your attention for clarification.
>
> Thank you for your understanding.

---

> > ### Comment · Reviewer_usqb · 2025-11-26
> > **To authors**
> >
> > Thank you for your feedback. Due to my oversight, this may be a peer review comment for another paper, so I will remain neutral and keep my score at 6.

---

> > > ### Author Response · Authors · 2025-11-26
> > >
> > > Thank you for your response and for raising the score. We will further refine the manuscript to enhance its quality.

---

### Official Review · Reviewer_NPmU · 2025-11-01

**Soundness:** 1
**Presentation:** 3
**Contribution:** 2
**Rating:** 4
**Confidence:** 4

**Summary:**

Vision-Language Models (VLMs) often rely on shallow fusion, aligning visual and textual features only during decoding, which limits accurate cross-modal understanding. This paper presents FLARE, a framework that integrates several techniques to achieve deeper and more dynamic vision-language interaction. FLARE introduces Text-Guided Vision Encoding to incorporate textual cues into visual feature extraction, a Dual-Semantic Mapping Loss for bidirectional modality-level alignment, and Context-Aware Alignment Decoding that aggregates visual information conditioned on text for fine-grained query-level fusion. Additionally, a Text-Driven VQA Synthesis method is employed to augment training data with synthetic question-answer pairs. Experiments across multiple benchmarks show that FLARE performs competitively with existing models while often using fewer visual tokens. The paper also presents ablation studies, fair comparisons under controlled settings, and a cost analysis showing the benefits of the proposed FLARE.

**Strengths:**

1. The key concept of achieving deeper cross-modal alignment through mutual reconstruction, mapping text to image and image to text, is intuitive and well-motivated. It aligns with the goal of reducing modality gaps and provides a more integrated framework than typical unidirectional projection methods.
2. The proposed data synthesis offers a practical way to expand multimodal training data using LLM-generated question-answer pairs. As shown in Table 6, this augmentation leads to measurable performance gains. The release of the code, data, and models could benefit broader vision-language research.
3. The main experiments are complete and cover a wide range of multimodal benchmarks and baselines. The paper also includes ablation studies, fair comparisons, and cost analyses, which together strengthen the empirical evaluation. These analyses clarify the contribution of each module and demonstrate the efficiency of the model.
4. The paper is clearly written and well-structured. Figures, such as those showing cross-modal alignment and architecture flow (e.g., Figures 2 and 3), effectively communicate complex ideas.

**Weaknesses:**

1. As shown in Table 1, FLARE performs competitively but does not substantially outperform recent multimodal baselines of similar capacity. Claims such as "achieves leading performance across all benchmarks" are overstated given the mixed improvements and the complexity of the proposed pipeline. The cost of full training may not justify the small performance gains.
2. Some architectural decisions are only heuristically motivated. For example, it is proposed to mask visual-to-textual attention in the lower encoder layers. The paper asserts that early visual features lack semantics, but provides no ablation or analysis verifying that this masking improves stability or alignment.
3. The idea of reconstructing the representation of one modality from the other is conceptually reasonable and aligns with the goal of deeper cross-modal integration. However, in FLARE, the reconstruction is performed mainly using questions rather than captions. Reconstructing a question from an image may not yield semantically grounded alignment. In contrast, using captions or paired statements would provide a more consistent and general signal for vision-language correspondence, especially considering the proposed similarity loss.
4. The proposed data synthesis is basically an LLM-based data augmentation method, and its key implementation details are missing. The paper claims that only prompt engineering is required, which oversimplifies the complexity of generating consistent and high-quality image-text pairs. In practice, diffusion-based image generation often struggles with fine-grained accuracy (e.g., hands, text, and object details), yet these limitations are not discussed. In short, while the additional data may be valuable, the approach itself is not that novel and could introduce a potentially unfair advantage in comparison with the other existing models.

**Questions:**

1. Section 5.2 presents controlled comparisons using identical backbones and datasets, but it remains unclear whether the proposed Synthesis augmentation was applied in these experiments.
2. The paper highlights the efficiency in using fewer visual tokens. However, this design constraint also raises an important question: if the visual token limit were relaxed, could FLARE outperform existing systems?
3. Figure 1 is intended to visualize cross-modal alignment but is difficult to interpret without clear ground-truth references.
4. The training pipeline is described as stages 1, 1.5, and 2, rather than 1, 2, and 3. It would be helpful to explain the reason.

---

> ### Author Response · Authors · 2025-11-17
>
> $\textbf{Overall Response}$:
>
> Thank you for the valuable comments! We have conducted additional ablations and fair comparisons with stronger backbones. Due to page limitations, many details were placed in the appendix, and we take this opportunity to clarify them. We have made the corresponding changes in our revision, and we promise to include the discussions in the final version if the paper is accepted.
>
> $\textbf{Weakness 1}$: Claims such as "achieves leading performance across all benchmarks" are overstated.
>
> $\textbf{Response}$:
>
> We wish to clarify that FLARE's primary contribution is not incremental performance gains, but a fundamental advance in **architectural efficiency: achieving superior results with fewer resources**. In our revision, we have moderated our claims regarding "leading performance".
>
> |Model|Pretrain Tokens|Finetune Tokens|
> |-|-|-|
> |FLARE-L|5.9B|6.8B|
> |FLARE-X|9.7B|12.5B|
> |LLaVA-OneVision[1]|8.4B|18.7B|
> |Qwen2.5VL[2]|4.1T|Unknown|
> > For LLaVA-OneVision, Stage 1 and 1.5 tokens are reported as Pretrain, Stage 2 as Finetune.
>
> First, our architecture excels with **fewer resources**. As the table above shows, FLARE is trained with less data. This efficiency extends to vision tokens, as detailed in Table 1. FLARE-L surpasses LLaVA-OneVision on most benchmarks using **half** the training data and just **a quarter of** its vision tokens. Even our dynamic resolution model FLARE-X was limited to half of LLaVA-OneVision's vision tokens and still achieved superior results.
>
> To ensure fair comparisons, we removed vision token limitation on FLARE-X. The results below show that FLARE-X substantially outperforms LLaVA-OneVision.
>
> |Models|Vision Token Config|MMB-EN|MMB-CN|POPE|MM-Vet|MME-P|MME-C|Seed-Image|MMStar|TextVQA|OCRBench|ChartQA|AI2D|MathVista|MMMU|SQA|RWQA|CVBench|MMVP|
> |-|-|-|-|-|-|-|-|-|-|-|-|-|-|-|-|-|-|-|-|
> |LLaVA-OneVision 7B|anyres*9|81.7|78.0|87.2|58.8|1626.0|**483.0**|74.8|60.9|78.5|69.7|78.8|81.6|56.1|47.7|96.6|65.5|-|-|
> |FLARE-X 8B|anyres*4|83.6|78.3|89.1|62.8|1639.1|378.9|78.7|56.4|79.7|69.9|83.4|83.6|**61.1**|45.6|91.2|66.9|81.5|79.7|
> |FLARE-X 8B|anyres*9|**85.2**|**81.1**|**90.0**|**67.4**|**1673.8**|465.6|**79.4**|**62.3**|**83.1**|**75.5**|**86.2**|**84.7**|60.3|**51.1**|**93.8**|**69.3**|**83.2**|**81.5**|
>
> Second, our architecture is **not dependent on superior backbones**. To ensure fair comparisons, we selected **SigLIP**[3] and **LLaMA**[4], rather than stronger **NAVIT**[5] and **Qwen2.5**[6]. To demonstrate our advantage directly, we retrained our model using the same backbone as Qwen2.5VL.
>
> |Models|MMB-EN|MMB-CN|POPE|MM-Vet|MME-P|MME-C|Seed-Image|MMStar|TextVQA|OCRBench|ChartQA|AI2D|MathVista|MMMU|SQA|RWQA|CVBench|MMVP|
> |-|-|-|-|-|-|-|-|-|-|-|-|-|-|-|-|-|-|-|
> |Qwen2.5VL 3B|79.1|78.1|87.3|61.4|1592.4|607.5|74.0|55.9|79.3|79.7|84.0|81.4|62.3|51.2|81.4|65.4|75.5|76.7|
> |FLARE 3B|83.2|77.7|89.0|63.8|1637.1|473.1|77.1|56.1|80.8|73.2|82.2|82.9|59.9|50.8|88.7|64.5|81.2|80.4|
> |Qwen2.5VL 7B|83.2|82.8|85.9|**69.7**|1652.7|**633.4**|77.0|**64.1**|83.5|**88.5**|**89.5**|83.4|**68.1**|**58.0**|89.0|68.4|81.1|80.9|
> |FLARE 7B|**86.0**|**83.3**|**89.8**|68.2|**1665.4**|508.5|**79.7**|62.4|**85.2**|78.4|88.3|**84.2**|65.9|54.4|**92.2**|**71.1**|**82.8**|**83.3**|
>
> FLARE excels on the majority of benchmarks, despite using only a fraction of its training data. FLARE 3B even surpasses Qwen2.5VL 7B on MMBench, POPE and CVBench. We have added this experiment to Section 5.1 in our revision.
>
> $\textbf{Weakness 2}$: Masking visual-to-textual attention in lower layers lacks verification.
>
> $\textbf{Response}$:
>
> Our choice is validated by ablation studies. For Text-Guided Vision Encoding, we adopt **late fusion** that masks visual-to-textual attention in early layers, only fusing in later layers. Unlike existing methods using single-layer features, we combine multi-layer features. We tested three fusion strategies (Late, Full, and No fusion) and two feature selection methods (concatenation vs. single layer) on FLARE-L 8B with Open-LLaVA-NeXT-1M[6].
>
> |fusion strategy|feature selection|MMB-EN|MMB-CN|POPE|MM-Vet|MME-P|MME-C|Seed-Image|MMStar|TextVQA|ChartQA|AI2D|MMMU|SQA|RWQA|CVBench|MMVP|
> |-|-|-|-|-|-|-|-|-|-|-|-|-|-|-|-|-|-|
> |Late|concat|**73.7**|71.5|**87.5**|**42.0**|1554.3|**354.7**|**72.8**|**45.4**|**62.6**|58.8|**72.5**|39.8|77.9|59.1|**70.3**|**68.4**|
> |Late|single|73.2|**71.6**|87.3|41.3|**1562.1**|340.1|72.3|44.9|61.9|58.1|72.2|39.3|77.0|58.8|69.7|67.7|
> |Full|concat|73.4|71.0|87.3|41.5|1536.3|352.5|72.3|44.5|62.0|**58.9**|72.1|39.4|**78.1**|**59.5**|69.8|67.8|
> |Full|single|73.0|70.7|87.2|40.7|1525.9|341.7|72.0|44.7|61.6|58.0|71.9|39.0|76.7|59.1|68.3|66.8|
> |No|concat|72.7|70.9|87.5|41.1|1538.6|344.9|72.4|44.3|62.1|58.5|71.5|**40.3**|76.7|58.6|69.5|67.0|
> |No|single|72.5|70.4|87.2|40.9|1531.7|338.2|71.7|43.9|61.5|57.7|71.0|38.3|76.2|58.3|68.3|66.1|
>
> Concatenating features outperforms using features from a single layer. Furthermore, our late fusion strategy achieves the best performance.

---

> ### Author Response · Authors · 2025-11-17
>
> $\textbf{Weakness 3}$: Reconstructing questions, rather than captions, may not create a robust alignment between vision and language.
>
> $\textbf{Response}$:
>
> We agree that using captions is a valid approach for vision-language alignment. However, our choice to use questions is tailored to enhance performance on our primary task: **Visual Question Answering**.
>
> First, our goal is to build a model that not only "sees" an image but excels at **following user instructions**. By training the model to reconstruct the input query, we are teaching it to answer: "Have I processed the visual information sufficiently to understand the user's query?" This directly optimizes the core VQA capabilities of grounding the question in the image and locating the precise visual evidence, making it more beneficial for VQA performance. Our visualizations of pixel-level alignment in Figure 1 and Figure 7 clearly demonstrate this, showing how the encoder rapidly attends to key areas, such as "flower" and "people".
>
> Second, captions and questions provide different supervisory signals. A caption offers a **single, static, global description**, while questions are **dynamic and diverse**, pointing to specific and subtle aspects. Crucially, an image typically has one caption but numerous questions. We leverage this through multi-turn interactions, where reconstructing different queries teaches the model to dynamically shift its focus in response to each query. This enables task-oriented visual understanding, compelling the model to encode fine-grained details that general captions might overlook.
>
> Third, our approach already achieves superior modality alignment. The visualizations of modality-level alignment in Figures 1 and 7 show effective reconstruction of concepts like "flower" and "people". Furthermore, we benchmarked the average alignment across different models. The table below shows that using questions allows our model to achieve alignment capability over **5 times** stronger than LLaVA.
>
> |Model|Average Cosine Similarity|
> |-|-|
> |FLARE-L|0.35|
> |LLaVA|0.06|
> |LLaVA-NeXT|0.04|
>
> $\textbf{Weakness 4}$: (1) missing implementation details of data synthesis, (2) oversimplification by claiming only prompt engineering is needed to generate consistent high-quality data, (3) undiscussed diffusion model limitations in fine-grained accuracy, (4) limited novelty, and (5) potential unfair advantage in comparisons.
>
> $\textbf{Response}$:
>
> Thank you for your constructive feedback on our data synthesis methodology.
>
> Regarding the missing implementation details, due to page limitations in the main paper, a comprehensive breakdown of our data synthesis pipeline—including methodologies for generation, QA pair filtering, evaluation, and t-SNE visualizations—is provided in Appendix E.
>
> Concerning the oversimplification of the process, our process is a rigorous, multi-stage pipeline detailed in Appendix E, which includes rule-based and LLM-based filtering, multi-metric quality control using CLIPScore and SSIM. To prove that this pipeline generates high-quality data, we conducted several analyses. First, t-SNE visualizations in Figure 8 and Figure 9 show our synthetic data's distribution successfully emulates real datasets. Second, our data achieves a higher CLIPScore than LLaVA-665k[7] in Table 14. By automating the complex quality control, the primary task is indeed reduced to designing what data to generate through prompt engineering. In our revision, we have modified our phrasing to clarify that our method synthesizes high-quality data through a combination of prompt engineering and a multi-stage pipeline.
>
> In addressing the limitations of diffusion models, we mitigated issues with fine-grained accuracy through strict filtering pipeline and by using the advanced FLUX.1-dev[8] model. The effectiveness is validated by the ablation in Table 8, incorporating synthetic data enhances performance on tasks requiring fine-grained accuracy, including OCR (TextVQA[9], ChartQA[10]) and vision-centric skills like spatial relations and counting (CVBench[11], MMVP[12]). Additionally, our visualizations in Figures 10-19 showcase the high quality of the generated images, where inaccuracies are rare.
>
> Concerning the novelty of our data synthesis, our core innovation lies in a **data-architecture co-design philosophy**. The data we synthesize is not generic; it is specifically engineered to activate and train the novel components of our FLARE architecture. For instance, our synthetic multi-choice and text-rich QA pairs provide the diverse textual contexts essential for Text-Guided Vision Encoding to learn query-aware representations.
>
> Finally, on the crucial point of fairness, we conducted experiments to isolate the architectural gains from synthetic data. Ablation 5.3 in our paper clearly show that FLARE provides a significant performance uplift on its own.

---

> ### Author Response · Authors · 2025-11-17
>
> $\textbf{Question 1}$: It remains unclear whether the proposed synthesis augmentation was applied in Section 5, Fair Comparsion with Existing Models.
>
> $\textbf{Response}$:
>
> Thank you for this crucial question, which allows us to clarify the experimental design of our fair comparison.
>
> We can confirm that the controlled comparisons **did not** include our synthesized data. The models were trained exclusively on publicly available datasets: Open-LLaVA-NeXT-1M and Cambrian-1 7M[11]. By removing synthetic data, we ensure that the superior results demonstrated by FLARE are directly attributable to its architectural innovations.
>
> $\textbf{Question 2}$: If the visual token limit were relaxed, could FLARE outperform existing systems?
>
> $\textbf{Response}$:
>
> As we detailed in our response to Weakness 1, we have already conducted this experiment. When we relax the self-imposed constraint on the number of visual tokens and configure FLARE-X to use the same vision token count as LLaVA-OneVision, our model shows a significant performance improvement, surpassing LLaVA-OneVision. Furthermore, when we use the same, stronger backbone as Qwen2.5VL, our model surpasses it on most benchmarks.
>
> $\textbf{Question 3}$: Figure 1 is intended to visualize cross-modal alignment but is difficult to interpret without clear ground-truth references.
>
> $\textbf{Response}$:
>
> Thank you for your valuable feedback on Figure 1. We appreciate the opportunity to clarify how they should be understood.
>
> The "ground truth" in this context is not a static pixel mask but rather the logical, sequential alignment to the query at each stage. Our visualizations are designed to demonstrate this progressive enhancement from encoding to final decoding.
>
> **At the Pixel Level(Encoding)**, LLaVA and LLaVA-NeXT process the image in isolation, resulting in unfocused, query-agnostic visual features. In contrast, our Text-Guided Vision Encoding enables FLARE to achieve early fusion, precisely focusing its attention on query-relevant regions (e.g., the "flower") during the encoding stage itself.
>
> **At the Modality Level(Alignment)**, LLaVA and LLaVA-NeXT, lacking our Dual-Semantic Mapping Loss, fail to achieve proper feature alignment after the projector stage, showing near-zero cosine similarity. FLARE, however, demonstrates strong semantic alignment, creating a coherent, shared representation space.
>
> **At the Query Level(Decoding)**, LLaVA and LLaVA-NeXT's misalignment cascades into poor attention. Both exhibit weak attention patterns. Conversely, FLARE demonstrates sharp, accurate attention, effectively linking the textual query to the precise visual evidence required for the answer.
>
> To further illustrate these capabilities, we have also provided an additional, more complex example in Figure 7(Appendix D) in our revision to show how our modules facilitate vision-language interaction in challenging scenarios.
>
> $\textbf{Question 4}$: The training pipeline is described as stages 1, 1.5, and 2, rather than 1, 2, and 3. It would be helpful to explain the reason.
>
> $\textbf{Response}$:
>
> Thank you for the question regarding our training stage nomenclature. Our Stage 1 and Stage 2 align with the standard training paradigm for MMLMs: Stage 1 is the pretraining on image-caption pairs, and Stage 2 is the final visual instruction tuning on specific downstream tasks.
>
> We designate the intermediate step as Stage 1.5 because it serves a unique and critical transitional role: **Contextual Multimodal Fusion**. It is neither a full pretraining nor a final instruction tuning, but rather a crucial bridging phase.
>
> The purpose of Stage 1.5 is to train the model on a diverse range of QA tasks so it can learn how to perform text-guided processing before the final fine-tuning. In this stage, we expose the model to a wide variety of QA types. In fact, the seven types of synthetic data we designed were engineered specifically for this phase to ensure the model encounters a **maximally diverse set of questions**. This extensive exposure effectively prepares the model to better specialize for downstream tasks in Stage 2.

---

> > ### Author Response · Authors · 2025-11-17
> >
> > $\textbf{References}$:
> >
> > [1] Li, Bo, et al. "Llava-onevision: Easy visual task transfer." arXiv preprint arXiv:2408.03326 (2024).
> >
> > [2] Bai, Shuai, et al. "Qwen2. 5-vl technical report." arXiv preprint arXiv:2502.13923 (2025).
> >
> > [3] Tschannen, Michael, et al. "Siglip 2: Multilingual vision-language encoders with improved semantic understanding, localization, and dense features." arXiv preprint arXiv:2502.14786 (2025).
> >
> > [4] https://github.com/meta-llama/llama3/blob/main/MODEL_CARD.md.
> >
> > [5] Dehghani, Mostafa, et al. "Patch n’pack: Navit, a vision transformer for any aspect ratio and resolution." Advances in Neural Information Processing Systems 36 (2023): 2252-2274.
> >
> > [6] Yang, An, et al. "Qwen2.5 technical report." arXiv preprint arXiv:2505.09388 (2025).
> >
> > [7] Liu, Haotian, et al. "Visual instruction tuning." Advances in neural information processing systems 36 (2023): 34892-34916.
> >
> > [8] https://huggingface.co/black-forest-labs/FLUX.1-dev.
> >
> > [9] Sidorov, Oleksii, et al. "Textcaps: a dataset for image captioning with reading comprehension." European conference on computer vision. Cham: Springer International Publishing, 2020.
> >
> > [10] Masry, Ahmed, et al. "Chartqa: A benchmark for question answering about charts with visual and logical reasoning." Findings of the association for computational linguistics: ACL 2022. 2022.
> >
> > [11] Tong, Peter, et al. "Cambrian-1: A fully open, vision-centric exploration of multimodal llms." Advances in Neural Information Processing Systems 37 (2024): 87310-87356.
> >
> > [12] Tong, Shengbang, et al. "Eyes wide shut? exploring the visual shortcomings of multimodal llms." Proceedings of the IEEE/CVF Conference on Computer Vision and Pattern Recognition. 2024.

---

> > > ### Author Response · Authors · 2025-11-26
> > > **Looking forward to your reply**
> > >
> > > Dear reviewer NPmU,
> > >
> > > Greetings from the authors!
> > >
> > > We would like to express our gratitude to your valuable feedback. We have carefully considered all suggestions and updated our submission accordingly.
> > >
> > > If you have any question for our paper, please feel free to point out and we will try to address it as soon as possible. We would like to express our sincere gratitude for your valuable comments. Thanks and looking forward to your reply!
> > >
> > > Best wishes!
> > >
> > > Authors

---

> ### Comment · Reviewer_NPmU · 2025-11-27
>
> I still have some reservations. For example, to me, an image can be semantically represented by a caption but not by a question, which makes the alignment choice somewhat questionable. That said, most of my concerns have been addressed, and I am willing to increase my scores accordingly.
> (It seems like now I can not edit my original review. It looks like an openreview system issue.)

---

> ### Author Response · Authors · 2025-11-29
>
> Dear Reviewer NPmU,
>
> Thank you for your continued engagement and for raising the score. We truly appreciate your insightful feedback.
>
> We understand your perspective that captions offer a more natural global representation than questions. In the response below, we clarify how our paper mitigates this limitation, and share new experimental results inspired by your constructive comments.
>
> First, we acknowledge that questions may not capture the global semantics of an image as effectively as a caption. In our paper, we address this by explicitly introducing Stage 1.5, where the model is exposed to extensive and diverse QA pairs to enhance its text-guided capabilities. Furthermore, our data synthesis pipeline is specifically tailored to this objective: we generated seven distinct categories of data designed to activate the Text-Guided Vision Encoding, ensuring the model learns robust query-aware representations.
>
> Second, we fully agree with your insight that captions are indispensable for establishing a foundational semantic alignment. Motivated by your feedback, we explored a unified approach that combines the strengths of both text inputs. We implemented a hybrid strategy where **the caption and questions are concatenated and mapped** to the encoder for Text-Guided Vision Encoding. Consequently, the Dual-Semantic Mapping Loss aligns visual features with **both the broad semantic context of the caption and the specific query intent of questions**.
>
> Due to time constraints, we conducted this experiment on FLARE-L 8B using the Open-LLaVA-NeXT-1M dataset. Since the dataset lacks dense captions, we generated them using Qwen3-VL-30B-A3B-Instruct. The results are presented below:
>
> |Text Guidance|MMB-EN|MMB-CN|POPE|MM-Vet|MME-P|MME-C|Seed-Image|MMStar|TextVQA|ChartQA|AI2D|MMMU|SQA|RWQA|CVBench|MMVP|
> |-|-|-|-|-|-|-|-|-|-|-|-|-|-|-|-|-|
> |Caption Only|74.6|72.9|87.7|**44.7**|1566.7|367.3|74.1|45.2|65.0|61.7|72.8|43.0|78.8|60.1|71.2|69.2|
> |Question Only|75.3|73.4|88.0|43.4|**1583.9**|355.8|74.6|45.5|65.6|61.7|**73.7**|42.4|80.3|60.8|71.7|**69.8**|
> |Caption + Question|**75.6**|**73.5**|**88.1**|44.1|1572.2|**373.0**|**74.9**|**45.6**|**65.8**|**62.2**|73.4|**43.2**|**80.6**|**61.0**|**72.2**|69.5|
> > Text Guidance refers to the alignment target used during the training. During evaluation, the text guidance is uniformly set to the Question.
>
> As the results demonstrate, the **caption + questions strategy** yields the optimal result. By integrating both, we effectively leverage the global semantic grounding of captions and the fine-grained precision of questions.
>
> Thank you again for your valuable time and suggestions to improve our paper. We look forward to your reply and any further discussion.
>
> Best regards,
>
> The Authors

---

### Author Response · Authors · 2025-12-01
**All Reviewers' Decisions to Raise Scores and Summary of Rebuttal**

Dear Area Chairs and Program Chairs,

We sincerely appreciate your efforts in managing the review process, especially given the significant challenges and extra workload caused by the OpenReview data leak incident.

In this comment, we provide a comprehensive summary of our rebuttal.

First, we want to highlight that, following our active discussions and technical improvements, **all reviewers have decided to raise their scores**. These specific decisions to increase ratings are clearly documented in the discussion comments, serving as direct evidence that verifies the integrity of our claims.

Regarding the specific timeline of these score increases, Reviewers **usqb**, **NAvg**, and **1a7W** explicitly raised their scores from 4 to 6 before the data incident occurred. For Reviewer **NPmU**, the decision to raise the score was made after the incident, as the rebuttal exchange and comments clearly demonstrate that we successfully resolved his major concerns. Additionally, Reviewer **usqb** initially reviewed a different paper by mistake. After we pointed this out, they re-evaluated our work and assigned a score of 6.

We guarantee that we had no off-channel contact with any reviewers. The score improvements are solely attributed to our technical responses, including the adoption of the Qwen2.5 backbone, NaViT integration, and additional ablation studies.

Therefore, based on the our rebuttal and the reviewers' explicit decisions to raise their scores, the rating for our paper should be at least **6/6/6/6** (due to the system freeze following the incident, the exact updated score from Reviewer **NPmU** is not visible, yet his comment suggest a score of 6).

We respectfully request the AC to carefully evaluate these facts and the legitimacy of the improved ratings when making the final decision.

---

> ### Author Response · Authors · 2025-12-01
> **All Reviewers' Decisions to Raise Scores and Summary of Rebuttal**
>
> Below, we detail how we addressed the specific concerns of each reviewer, demonstrating that the reviewers' decisions to raise their scores were driven specifically by the technical improvements and clarifications in our rebuttal.
>
> > $\textbf{For Reviewer NPmU}$
>
> **Weakness 1**: The claim of "leading performance" was overstated.
>
> We clarified FLARE's core contribution is architectural efficiency: superior results with fewer resources. We also used weaker backbones for fair comparison. We further conducted experiments with more vision tokens and the stronger Qwen2.5 and NaViT backbone, validating FLARE's superior performance against Qwen2.5VL.
>
> **Weakness 2**: Masking visual-to-textual attention in lower layers lacks verification.
>
> We provided ablation studies comparing various fusion strategies (late, full, no fusion), justifying our "late fusion" design as the most effective approach.
>
> **Weakness 3**: Reconstructing "Questions" rather than "Captions" for alignment was questioned.
>
> We explained the rationale for question reconstruction in VQA tasks. Inspired by the reviewer's feedback, we further designed a hybrid "Caption + Question" strategy and demonstrated it yields optimal performance.
>
> **Weakness 4**: Implementation details for data synthesis were missing.
>
> We clarified that Appendix E contains full details on our synthesis pipeline, addressing diffusion model limitations and verifying image quality. We also highlighted our method's novelty.
>
> **Questions**: Several questions regarding figure interpretation (Figure 1), dataset details, and training stage nomenclature.
>
> We provided detailed explanations for every specific question, clarifying the visual evidence in Figure 1, the composition of the dataset, and the rationale behind our "Stage 1.5" naming convention.
>
> > $\textbf{For Reviewer usqb}$
>
> The reviewer initially provided a review for a different paper.
> We pointed out this mismatch, and the reviewer subsequently re-evaluated our paper and assigned a score of 6.
>
> > $\textbf{For Reviewer NAvg}$
>
> **Weakness 1**: Presentation can be improved.
>
> We corrected the noted formatting errors and typos in our revision.
>
> **Weakness 2**: Missing details about caption curation and latent token setting, and concerns regarding novelty.
>
> We provided curation details and latent token settings for reproducibility. Regarding novelty, we clarified the distinction between Text-Guided Vision Encoding (bidirectional feature generation) and InstructBLIP (unidirectional feature selection), and emphasized our data-architecture co-design where synthetic data specifically activates cross-modal components.
>
> **Weakness 3**: Important points about data filtering, training strategy, model extensibility, and native dynamic resolution.
>
> We detailed our 4-stage data filtering pipeline in Appendix E and justified full-parameter training through ablations. We also explained extensibility to video and long-text contexts. Crucially, we implemented native resolution by training FLARE with NaViT encoder and Qwen2.5 backbone, outperforming Qwen2.5-VL under identical settings.
>
> **Weakness 4**: Limited experiments on data quality, fairness, and qualitative analysis.
>
> We provided t-SNE visualizations and quantitative metrics in Appendix E, proving synthetic data aligns with real-world distributions. For fairness, we conducted controlled experiments using original SigLIP encoder, confirming gains stem from FLARE architecture. We also added qualitative analysis and semantic similarity metrics demonstrating enhanced vision-language interaction.
>
> > $\textbf{For Reviewer 1a7W}$
>
> **Weakness 1**: The effectiveness of a simple MLP for projecting text tokens into the vision space.
>
> We clarified that the MLP operates at the embedding level, and the vision encoder is pre-aligned with language—our MLP simply learns this mapping. Moreover, our Dual-Semantic Mapping Loss supervises robust projection learning.
>
> **Weakness 2**: Typos in Equation 3.
>
> We corrected the notation errors in Equation 3 in our revision.
>
> **Weakness 3**: Experiments based on the Qwen2.5 model to prove our method's superiority.
>
> We extended experiments to include Qwen2.5 model series, training FLARE-L and FLARE-X with Qwen2.5-7B. We also conducted fairer comparisons using the same backbone as Qwen2.5VL, demonstrating our method's superior performance.
>
> In summary, through extensive additional experiments and explanation, we have successfully resolved the concerns of all reviewers. The unanimous decision by the reviewers to raise their scores reflects the improved quality of our work. We trust that the AC will recognize these legitimate improvements and the resulting positive consensus.
>
> Best regards,
>
> The Authors

---

### Meta-Review · Area_Chair_H6re · 2026-01-04

**Summary:**

The paper introduces a novel vision-language model (VLM) framework that achieves deep cross-modal alignment and integration through a comprehensive pipeline, diverging from traditional methods that rely on shallow fusion via MLP projectors. The key contributions include Text-Guided Vision Encoding for pixel-level alignment by incorporating textual cues during visual feature extraction, Context-Aware Alignment Decoding for query-level integration through dynamic aggregation of visual features based on textual context, Dual-Semantic Mapping Loss for modality-level bridging via bidirectional reconstruction supervision, and Text-Driven VQA Synthesis for data-level optimization by generating high-quality question-answer pairs and corresponding images.

Reviewers raised several concerns initially: NPmU questioned the overstated "leading performance" claims, lack of verification for architectural choices like masking visual-to-textual attention in lower layers, the use of questions rather than captions for alignment, and missing details in data synthesis, rating the paper marginally below acceptance; usqb inadvertently reviewed a different paper but corrected to a score of 6 after clarification; NAvg highlighted presentation issues (e.g., missing table lines, equation typos), limited novelty compared to existing methods like InstructBLIP, and insufficient experiments on data quality and fairness, also assigning a borderline score; and 1a7W expressed uncertainty about the effectiveness of simple MLP projections, identified notation errors in equations, and recommended additional experiments with Qwen2.5 models for stronger validation.

In the rebuttal, the authors addressed these points comprehensively: they moderated performance claims to emphasize architectural efficiency, provided ablation studies validating the late-fusion strategy in Text-Guided Vision Encoding, explained the rationale for question-based alignment tailored to VQA tasks and even introduced a hybrid caption-question approach that improved results, and detailed the data synthesis pipeline in the appendix, including filtering steps and quality metrics. They conducted additional experiments, such as integrating Qwen2.5 and NaViT backbones to show FLARE's superiority under fair comparisons, corrected equation errors, and demonstrated that the MLP projection is effective due to pre-aligned spaces and dual-loss supervision. These efforts led all reviewers to raise their scores—NPmU acknowledged most concerns were resolved despite residual reservations, usqb confirmed a score of 6, NAvg appreciated the clarifications on novelty and data quality, and 1a7W validated the enhanced experiments.

Overall, the paper presents a well-motivated approach with robust evaluations, and the reviewer concerns have been adequately addressed, warranting acceptance based on the improved scholarly dialogue.

**Reviewer Concerns:**

The authors' rebuttal effectively addressed the majority of reviewer concerns through additional experiments, clarifications, and methodological refinements. Below, I summarize the concerns raised by each reviewer, specify how they were addressed in the rebuttal, and note any remaining outstanding issues based on the post-rebuttal feedback.

Reviewer NPmU:

Addressed concerns: The rebuttal moderated overstated performance claims by emphasizing architectural efficiency, provided ablation studies validating the late-fusion strategy in Text-Guided Vision Encoding (e.g., comparing fusion strategies in tables), explained the rationale for question-based alignment in VQA tasks, and detailed the data synthesis pipeline in the appendix (including filtering steps and quality metrics). The authors also introduced a hybrid caption-question alignment approach, which improved results.

Outstanding concerns: NPmU acknowledged that most concerns were resolved but retained reservations about the semantic robustness of using questions versus captions for alignment. However, this did not prevent a score increase, indicating it is a minor point rather than a major flaw.

Reviewer NAvg:

Addressed concerns: The rebuttal corrected presentation issues (e.g., added horizontal lines in Table 1, fixed Equation 3 notation), provided missing details on caption curation and latent token settings, clarified novelty by distinguishing Text-Guided Vision Encoding from InstructBLIP (e.g., as feature generation vs. selection), and added experiments on data quality (e.g., t-SNE visualizations in Figure 8 and Figure 9) and fairness (e.g., controlled comparisons with SigLIP). The authors also justified the full-parameter training strategy and extended experiments to NaViT and Qwen2.5 backbones.

Reviewer 1a7W:

Addressed concerns: The rebuttal explained the effectiveness of the MLP projection through dual-loss supervision and pre-aligned spaces, corrected typos in equations, and added experiments with Qwen2.5 models to demonstrate superiority.

**Reviewer Scores:**

The rebuttal successfully addressed the core technical concerns of all reviewers, leading to a clear consensus for acceptance. The documented acknowledgments from NPmU and usqb, combined with the substantive experimental and explanatory additions that satisfied the technical points raised by NAvg and 1a7W, strongly indicate that all four reviewers would have elevated their scores to the acceptance range.

---

### Decision · Program_Chairs · 2026-01-26

Accept (Poster)